



# Seismo-tectonics of Greater Iberia: An updated review

Antonio Olaiz[1], José A. Álvarez Gómez[2], Gerardo de Vicente[2,3], Alfonso Muñoz-Martín[2,3], Juan V. Cantavella[4], Susana Custódio[5], Dina Vales[6] and Oliver Heidbach[7,8].

[1]Repsol E&P, C7 Méndez Álvaro 44, 28045, Madrid, Spain
[2]GEODESPAL, Faculta de C.C. Geológicas, Universidad Complutense de Madrid, Spain
[3]Instituto de Geociencias IGEO, CSIC-UCM, Madrid, Spain
[4]Instituto Geográfico Nacional, C/General Ibáñez Íbero, 3, 28003, Madrid, Spain
[5]University of Lisbon
[6]Instituto Português do Mar e a Atmosfera, Lisboa, Portugal
[7]Helmholtz Centre Potsdam, GFZ German Research Centre for Geosciences. Germany, 14473 Potsdam, Germany
[8]Institute for Applied Geosciences, TU Berlin, 10587 Berlin, Germany

*Correspondence to*: Antonio Olaiz (antoniojose.olaiz@repsol.com)

**Abstract.** From the analysis of 542 moment tensor focal mechanisms in Iberia, active tectonic deformations and stresses were inferred by implementing different and complementary methodologies: FMC classification of the rupture type; composed focal mechanism based on the average seismic moment tensor; rotation angle between tensors estimates; Right Dihedra composed focal mechanisms; Slip Model analysis to determine the strain conditions and classical stress inversion methodology. By using the Slip Model results and considering the tectonic constraints of the Cenozoic deformation in Iberia, 20 the study region was subdivided into a series of zones where the different methods were individually applied. The results indicate that thrust faulting stress regimes are active in the Gorringe-Horseshoe area and the easternmost Tell Atlas. In the south, most of the zones are transpressive, as well as in the southwestern corner of Iberia, south of Lisbon. The exception is the Granada Basin, which displays an almost radial normal faulting stress regime. Normal faulting stresses are dominant in the Pyrenees and in the Mediterranean rim, north of the Betics. In the central part of the Pyrenees, we find a maximum 25 horizontal extension perpendicular to the range, indicating that local stresses related to post-orogenic collapse or isostatic rebound dominate over regional ones. The maximum horizontal compression along the Eurasia-Africa plate limit is very homogeneously close to N154°E, except in some parts of the Betics that are probably influenced by a remanent effect of the Alboran Slab. In the Central Ranges and offshore Atlantic, the maximum horizontal compression is slightly rotated anticlockwise to N140°E.



# 1 Introduction and objectives

The Iberian Peninsula shows evidence of an intense and distributed Alpine deformation that occurred over geologic time scales (de Vicente and Vegas, 2009) (Fig. 1). After the Variscan orogeny, and during the Mesozoic, numerous extensional structures developed, in which thick sedimentary deposits accumulated, with one exception, on the Iberian Massif to the

west. At the northern edge of Iberia, this extension even reached the stage of oceanic crust generation (Montadert et al., 1979; Nirrengarten et al., 2018; Sibuet et al., 2004), albeit during a very short time (Aptian-Albian) (Srivastava et al., 1990). According to tectonic reconstructions, the Iberian block moved independently relative to Africa and Eurasia until it collided with Eurasia to form the Cantabrian-Pyrenean Orogen. From the beginning of the Eocene, Iberia was significantly compressed, not only at its northern border, where an incipient subduction zone was located (Gallastegui and Pulgar, 2002;

Fernandez-Viejo et al., 2012), but also in its interior.

The result of Alpine compression in the interior of Iberia was the inversion of the Mesozoic aulacogen of the Iberian Basin (Iberian Chain, IC), and the development of a series of ranges with crustal thickening along the Iberian Massif (i.e. the Spanish-Portuguese Central System, SPCS). This set of intra-plate ranges can also be considered as an incipient and aborted orogen (de Vicente et al., 2022). It has also been suggested that the Iberian block accommodated shortening by forming

lithospheric folds (Cloetingh et al., 2002). Accompanying these large thrusts, major strike-slip faults and deformation belts were activated at the crustal scale, such as the South ("Castilian") and North ("Aragonese") Branches of the IC, and the Messejana-Plasencia fault (more than 500 km long), which nucleated on an end-Triassic basic dyke related to the Central Atlantic Magmatic Province (Cebriá et al., 2003; Villamor, 2002; de Vicente et al., 2021). The age of the main deformation event for these fault systems is Oligocene - Lower Miocene, although, in the westernmost sector, the SPCS and the left-

lateral strike-slip faults of Regua and Vilariça display significant deformation during the Middle-Upper Miocene, and are still considered as active structures (Cabral, 2012).

Today, extensional structures dominate the easternmost part of Iberia (since the Upper Miocene), due to back-arc extension related to a subduction zone below Corsica and Sardinia, which were initially a part of the Iberian Plate (van Hinsbergen et al., 2014). A normal faulting stress regime also affects the Pyrenees, where a post-orogenic collapse process has been

suggested (Asensio et al., 2012). Thus, the active plate boundary would have migrated from the north to the south of the Iberian Peninsula (Terceira Ridge - Gloria Fault – Alboran - Tell Atlas), where the emplacement of the Alboran Domain and the subduction of the southern edge of the Iberian Plate, have produced a diffuse plate boundary that encompasses the Betics, where shortenings and extensions occur almost simultaneously. In this Cenozoic and neotectonic complex deformation setting, it is not surprising that the present tectonic stresses in Iberia display large variations in both the stress regime and

orientation of the principal stress axes (de Vicente et al., 2008) over relatively small areas.

The estimation of earthquake focal mechanisms in recent years, performed by seismic institutions in Spain and Portugal (IGN, IAG and IPMA), has generated a large amount of information that adds to scientific publications resulting from



different projects, such as TopoIberia (e.g. Matos et al., 2018; Martín et al., 2015), or significant earthquake crisis (e.g. Cesca et al., 2021; Villaseñor et al., 2020).

In this study, we will exclusively use well-fitted moment tensor focal mechanisms to study the contemporary deformation pattern in Iberia. We analyse the rupture characteristics of focal mechanisms populations for defined tectonic subareas and use the Slip Model described by Reches (1983) and de Vicente (1988) to assess which of the two nodal planes was the rupture plane. This information and the focal mechanism populations are then used to perform a stress inversion to determine the orientation the maximum horizontal stress sHmax and the tectonic stress regime. We also derive from the individual

focal mechanism the sHmax orientation and integrate these results and the ones from the stress inversion into a revised dataset from the World Stress Map project based on borehole logs, overcoring measurements and geological stress indicators.





**Figure 1: Alpine-Cenozoic tectonic map of Iberia, including continental and offshore domains, and showing areas that have experienced intense and distributed deformation. More recent normal faulting is shown by red lines. Q: Quaternary.**



## 2 Data from earthquake focal mechanism

In this study we establish a new and comprehensive compilation of robust focal mechanism solutions of Greater Iberia, inferred from waveform moment tensor inversions, using the following catalogues:

- Global Centroid Moment Tensor (former Harvard Centroid Moment Tensor https://www.globalcmt.org/, Dziewonski et al., 1981; Ekström et al., 2012)

- Instituto Geográfico Nacional de España (https://www.ign.es/web/ign/portal/tensor-momento-sismico/-/tensor-momento-sismico/getExplotacion, Rueda and Mezcua, 2005)

- Instituto Andaluz de Geofísica (https://iagpds.ugr.es/investigacion/informacion-general, Stich et al., 2003, 2006, 2010).

- Istituto Nazionale di Geofisica e Vulcanologia (http://terremoti.ingv.it/en/tdmt; Scognamiglio et al., 2006; Pondrelli et al., 2002, 2004)

- GFZ-Postdam (https://geofon.gfz-potsdam.de/old/eqinfo/list.php?mode=mt)

- ETH- Swiss Seismological Service (https://geophysics.ethz.ch/research/groups/sed.html) (Braunmiller et al., 2002).

- IPMA Portuguese Institute for Sea and Atmosphere (https://www.ipma.pt/en/geofisica/tensor)

This dataset was then expanded with different regional publications (e.g. Carreño et al., 2008; Chevrot et al., 2011; Domingues et al., 2013; Custodio et al., 2016), datasets associated with projects such as TopoIberia (Martín et al., 2015),
datasets related to earthquake clusters (Morales et al., 2015; Matos et al., 2018) and specific publications concerning seismic crises (e.g. Cesca et al., 2021; Villaseñor et al., 2020). In addition, nine unpublished moment tensor focal mechanisms were determined for events occurred between 2003 and 2019 (Table 1) and focal mechanisms for two historical events were also included in our dataset (Stich et al., 2005). The events are shallower than the Moho proposed by Diaz et al. (2016), with exception of some events located in oceanic crust (depth < 30 km), where the rheology of the upper mantle might assume as
similar to the crust. In those cases where an event is duplicated, having solutions in more than one catalogue, the double couple percentages (%DC) were compared and the solution with the highest %DC was selected.

| Long | Lat | Depth | StrikeA | DipA | RakeA | StrikeB | DipB | RakeB | Mw | Date |
|---|---|---|---|---|---|---|---|---|---|---|
| -5.98 | 41.5 | 9 | 7 | 71 | 25 | 268 | 66 | 160 | 3.8 | 20030112 |
| -2.34 | 40.21 | 11 | 228 | 71 | -157 | 130 | 69 | -20 | 3.3 | 20091014 |
| -7.62 | 38.96 | 15 | 288 | 75 | 180 | 18 | 90 | 15 | 3.5 | 20100327 |
| -8.546 | 37.35 | 8.5 | 84 | 85 | 168 | 175 | 78 | 6 | 3.3 | 20150722 |
| -2.29 | 41.61 | 12 | 179 | 46 | -87 | 355 | 44 | -93 | 3.6 | 20150805 |
| -8.524 | 37.237 | 20 | 216 | 50 | 88 | 39 | 40 | 92 | 3.2 | 20151021 |
| -4.62 | 42.86 | 14 | 134 | 75 | -86 | 297 | 16 | -106 | 3.6 | 20180519 |
| -9.552 | 37.73 | 28 | 35 | 62 | 52 | 275 | 62 | 52 | 3.6 | 20181001 |
| -8.013 | 36.37 | 30 | 175 | 42 | -165 | 74 | 80 | -49 | 3.8 | 20190716 |

**Table 1: New focal mechanisms calculated in this study**





Thus, the final database consists of 542 events. In terms of temporal coverage, the first earthquake in our database is that of Benavente (Portugal) in 1909 and the most recent one occurred on 30 September 2023 (Fig. 2).

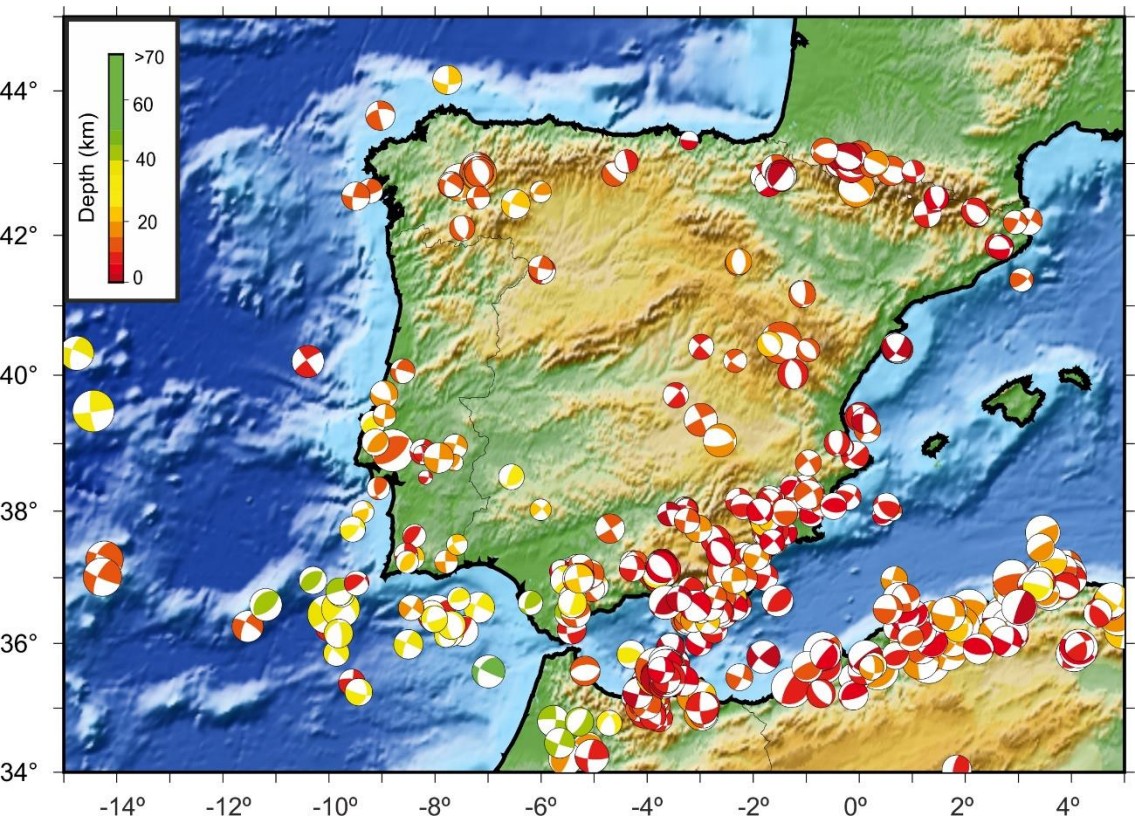

**Figure 2: Distribution of the 542 focal mechanisms used in this study. The color of focal mechanisms indicates hypocentral depth.**

## 3 Methodology

To describe the tensor characteristics of seismicity, either a stress-type or a deformation-type approach can be adopted. Both types of analysis are complementary, therefore we will use both. In the case of the deformation type analysis, we will use

two approaches: A kinematic analysis that will classify the type of rupture and obtain a combine focal mechanism (Álvarez-Gómez, 2019; Kiratzi and Papazachos, 1995), using also, the right dihedral diagram (Angelier and Mechler, 1977) and the slip model (Reches, 1983). To determine the stress tensor using classical inversion procedures, we will use the methodology proposed by Reches et al., (1992).

Thus, we will use the appropriate terminology for each type of analysis. In the case of individual focal mechanisms, we will

differentiate between reverse, strike-slip, and normal ruptures. When we refer to deformation, we will use shortening, shear, and extension; while when we deal with stresses, we will use normal faulting stress regime, strike-slip stress regime, and thrusting stress regime.



### 3.1 Kinematic analysis

To gain insight into the kinematics related to the brittle behaviour of the lithosphere, we binned the focal mechanisms
according to their rupture characteristics in tectonic sub-areas. We refer to these from here onwards to as tectonic zones.
These zones were delimited considering the tectonic regimes, using a methodology that is explained in more detail in section
4 as is a fundamental step on the stress inversion analysis.

For each tectonic zone, we classified the focal mechanisms by rupture type (reverse, strike-slip, normal) using the Focal
Mechanisms Classification (FMC) diagram (Álvarez-Gómez, 2019). Then, for each rupture type population, we obtained a
combined focal mechanism by averaging the moment tensor components following the approximation of Kiratzi and
Papazachos (1995):

$$\bar{F}_{ik} = \frac{\sum_{x=1}^{n} M_o^x F_{ik}^x}{\sum_{x=1}^{n} M_o^x}, \tag{1}$$

where Fik is the normalized moment tensor, M0 is the seismic moment, and the sums are performed over the number of
events in each rupture population. If most of the released seismic moment is controlled by one of the earthquakes, the
composed focal mechanism is very close to that larger event (Kiratzi and Papazachos, 1995). To avoid this effect in areas
with moderate seismicity, an adequate approximation is to use only the sum of the seismic moment tensor components,
disregarding the respective seismic moments. In this case, all earthquakes will have the same weight in the sum.

$$\bar{F}_{ik}^{RT} = \sum_{x=1}^{n} \bar{F}_{ik}^x, \tag{2}$$

where RT is each of the rupture types, encompassing n events, and i and k are the indices of the moment tensor components.
As a result, we obtain a combined focal mechanism for each rupture type reflecting the geometry of the corresponding
moment tensors and related deformation.

Following this reasoning we can obtain also a composed focal mechanism considering all the events in a tectonic zone:

$$\bar{F}_{ik}^{zone} = \sum_{y=1}^{n} \bar{F}_{ik}^{(y)} = \bar{F}_{ik}^{Reverse} + \bar{F}_{ik}^{Strike-slip} + \bar{F}_{ik}^{Normal}, \tag{3}$$

As can be seen in equation 3, the summation of the combined rupture types is equivalent to the sum of all tensors. This step
allows us to estimate the degree of strain partitioning by means of the computation of the minimum rotation angle (Kagan,
1991) between the different rupture types as is described below.

A way to assess the character of a moment tensor is to use its compensated linear vector dipole (clvd) component, which
quantifies the extent to which the deviatoric part of the seismic moment tensor differs from a pure double-couple. We used
the clvd ratio (fclvd) from Frohlich and Apperson (1992):

$$fclvd = \frac{|m_B|}{max[|m_T|, |m_P|]}, \tag{4}$$



where mT, mB and mP are the largest, intermediate, and smallest principal components of the summed moment tensor. When the double couple component is dominant, the fclvd tends to be 0 while when the value approaches 0.5 the tensor is far from a double couple. The clvd proportion of the summed moment tensor can be used also to analyse the seismotectonics of a zone (e.g. Frohlich and Apperson; 1992; Jost et al., 1998; Buforn et al., 2004; Borges et al., 2007; Bailey et al., 2012).

If we consider the composed moment tensor to be a representation of the seismic strain in a zone, we can use its orientation and characteristics to get insight into the deformation pattern of the studied tectonic zones. A first approximation is to consider the orientation of the principal strain axes. The T axis is equivalent to the extension axis, the P axis is equivalent to the shortening axis, and the B axis is the intermediate axis, which can be neutral for a plane strain deformation (a pure double couple moment tensor) or it can be an extension or a shortening axis depending on the tectonics of the zone.

It is of interest also to quantify the amount of seismic deformation taking place in each zone using the different rupture types. These different rupture processes on a zone cannot be considered to reflect temporal changes in the regional deformation field as the time interval of the catalogue is very short, but rather local strain axes permutations. To quantify these changes in the orientation of the axes between rupture types, we resort to the minimum rotation angle between tensors (or Kagan angle; Kagan, 1991). The angle for a pure axis permutation maintaining the orientation of all the axes would be 90°. In practice, if

we consider the angle between the focal mechanisms of different types of rupture and given that they nucleate in faults with different orientations, this angle may depart slightly from 90º.

Finally, to analyse the shape of the seismic deformation tensor, we can adapt the Flinn diagram for 3D strain tensor shapes used in classic structural geology analysis. The Flinn diagram represents the relation between the principal strain axes, or principal extension (Flinn, 1958), where the abscissa is the relation ŝ2/ŝ3 and the ordinate the relation ŝ1/ŝ2. An alternative to

these values was proposed by Ramsay (1967) who suggest the use of natural logarithm of these relations so that:

$$ln[\hat{s}_1/\hat{s}_2] = \varepsilon_1 - \varepsilon_2 \, , \tag{5}$$

$$ln[\hat{s}_2/\hat{s}_3] = \varepsilon_2 - \varepsilon_3, \tag{6}$$

where $\varepsilon 1$, $\varepsilon 2$ and $\varepsilon 3$ are the natural strains of the largest, intermediate, and smallest principal strain axes of the deformation ellipsoid; i.e. the magnitude of the changes in the length of the axes. Similarly, the principal strain axes of the combined

seismic moment tensors can be considered the amount of seismic strain change induced by the earthquakes in a volume. Consequently the logarithmic Flinn diagram can be adapted to this purpose. Additionally, the shape of the ellipsoid can also be defined by the k-value, which is defined as

$$k = \frac{\varepsilon_1 - \varepsilon_2}{\varepsilon_2 - \varepsilon_3}, \tag{7}$$

This parameter has values of 0 for oblate strain shape, 1 for plane strain and ∞ for prolate strain shapes. In the case of

seismic moment tensors, a pure double couple has the form



$$\begin{bmatrix} M_0 & 0 & 0 \\ 0 & 0 & 0 \\ 0 & 0 & -M_0 \end{bmatrix}, \tag{8}$$

having the values of the principal axes the same magnitude although of opposite sign to conserve the volume. Strictly speaking, a double couple corresponds to a plane strain ellipsoid with no strain on the orthogonal plane to the maximum and minimum moment axes mT=-mP, mB=0. If the intermediate moment axis is different from 0, then the compensated linear vector dipole component appears as shown in equation 3.

To represent the combine seismic moment tensors in the Flinn diagram we defined the ordinate and abscissa as M1 – M2 (or mT – mB) and M2 – M3 (or mB – mP) respectively, where M1 ≥ M2 ≥ M3 (we used the logarithm of these values to improve the data presentation). The shape of the tensor can then be defined in an equivalent way as in equations (5 and 6):

$$k = \frac{M_1 - M_2}{M_2 - M_3}, \tag{9}$$

## 3.2 Kinematic analysis

Focal mechanisms provide valuable insights into earthquake rupture kinematics, including the strike, dip and rake of the two nodal planes. However, most of the time, the selection of the true fault plane among the two possible ones is not straightforward. The strain orientation derived from thrust or normal faulting focal mechanisms may remain the same irrespective of the true fault plane, but the dip direction of the fault would not be constrained. On the other hand, for strike-slip faulting focal mechanisms, the strike of the fault plane is crucial to properly define the strain field.

The Slip Model (Reches, 1983; de Vicente et al., 1988) identifies which of the nodal planes is more prone to slip from a mechanical point of view (the neoformed plane). Additionally, it is helpful to identify areas under similar strain conditions. In this study, we apply the workflow suggested by de Vicente et al. (2008), Olaiz et al., (2009) and Arcila and Muñoz-Martín (2020). The Slip Model proposed by Capote et al. (1991), based on de Vicente (1988) and Reches (1983), defines the maximum shortening trend (Dey, EHmax) and the shape factor (k´) of the deformation ellipsoid for each individual focal mechanism. This method, based on the Navier–Coulomb fracture criterion, assumes that the brittle strain and stress axes are parallel and that one of the axes is close to vertical. According to the Slip Model, under the triaxial strain conditions of brittle strain, fractures are arranged in orthorhombic symmetry concerning the fundamental axes of the strain ellipsoid.

$$k´ = e_y/e_z, \tag{10}$$

where ez is the axis of the vertical strain and ey is the axis of the maximum horizontal shortening

Accordingly, replacing ey and ez in equation (10):

$$k´ = (sin^2 D cos^2 B)/(1 - sin^2 D cos^2 B) , \tag{11}$$

$$B = sin^2 D cos^2 P , \tag{12}$$





Where D is the dip and P the pitch of the slip vector on the fault plane.

Two sequences of strain are established as a function of k´, from reverse to normal trough strike-slip faulting, and k´ is rescaled to plot values continuously (Table 2).

| k´ = ∞ | Plane strain | Pure strike-slip (Pitch =0) |
|---|---|---|
| ∞ > k´> 1 | Shear with extension | Strike-slip normal |
| k´= 1 | | |
| 1 > k´> 0 | Extension with shear | Normal strike-slip |
| k´= 0 | Plane strain | |
| 0 > k´> -0.5 | Radial extension | Pure normal (Pitch = 90) |
| k´= -0.5 | Pure radial extension | |
| k´= -0.5 | Pure radial shortening | |
| -1 > k´> -0.5 | Radial shortening | Pure reverse (Pitch = 90) |
| k´= -1 | Plane strain | |
| -2 < k´< -1 | Shortening with shear | Reverse strike-slip |
| k´= -2 | | |
| -∞ < k´< -2 | Shear with shortening | Strike-slip reverse |
| k´ = -∞ | Plane strain | Pure strike-slip (Pitch = 0) |

**Table 2 k´ values obtained from the Slip Model**

Additionally, based on the relationship between dip and pitch proposed by de Vicente (1988), one the nodal plane coincides with the character of the focal and will dissipate frictional energy more efficiently. Thereby, the selected plane can be used in
stress inversion methods, based on striation-fault pair orientations (Angelier and Mechler, 1977). Hence, the quality of the stress inversion results is improved in comparison to that obtained using both planes (Michael, 1987; de Vicente, 1988; Giner-Robles et al., 2006).

**3.3 Dynamic analysis, stress inversions**

For the stress inversion we apply the method proposed by Reches et al. (1992). The method incorporates two constraints:
first, the stresses in the slip direction satisfy the Coulomb yield criterion; and second, the slip occurs in the direction of maximum shear stress along the fault. The computations yield the complete stress tensor, normalized by the vertical stress, and evaluate the mean coefficient of friction (μ) and the mean cohesion (C) of the faults during the time of faulting (Reches, 1987). Thus, for every selected population, two angular quality criteria are obtained: the slip misfit (SLIP), which is the mean angle between the observed and calculated slip axes of all faults in the cluster, and the principal angles misfit (PAM),
which is the angle between the ideal stress axes of each nodal plane and general stress axes of the entire group according to the optimal mechanical condition for faulting (Reches, et al., 1992). In addition, the stress ratio (R) is stablished as is proposed by McKenzie (1969), Etchecopar et al. (1981), Gephart and Forsyth (1984), Delvaux et al. (1997) among others (Equation 13, Table 3).

$$R = (\sigma_2 - \sigma_3)/(\sigma_1 - \sigma_3) \,, \tag{13}$$



To check the statistical representativeness of the population of focal mechanisms within in each tectonic zone, a Monte Carlo bootstrapping approach was applied to find the value for the friction coefficient with the smallest errors. This technique allows us to determine the possible variability of the principal stress axes, especially potential permutations between two principal stress axes when they have similar magnitudes.

| R | | Stress Regime | Vertical axis |
|---|---|---|---|
| R=1 | $\sigma_1 = \sigma_2 > \sigma_3$ | Radial Thrusting | $\sigma_3$ |
| $1 > R > 0$ | $\sigma_1 > \sigma_2 > \sigma_3$ | Triaxial Thrusting | $\sigma_3$ |
| R = 0 | $\sigma_1 > \sigma_2 = \sigma_3$ | Uniaxial Thrusting | $\sigma_3$ |
| $0.5 > R > 0$ | $\sigma_1 > \sigma_2 > \sigma_3$ | Strike-slip Thrusting | $\sigma_2$ |
| R = 0.5 | $\sigma_1 = \sigma_2 > \sigma_3$ | Pure Strike-slip | $\sigma_2$ |
| $1 > R > 0.5$ | $\sigma_1 > \sigma_2 > \sigma_3$ | Strike-slip Normal | $\sigma_2$ |
| R =1 | $\sigma_1 = \sigma_2 > \sigma_3$ | Uniaxial Normal | $\sigma_1$ |
| $1 > R > 0$ | $\sigma_1 > \sigma_2 > \sigma_3$ | Triaxial Normal | $\sigma_1$ |
| R =0 | $\sigma_1 > \sigma_2 > \sigma_3$ | Radial Normal | $\sigma_1$ |

**Table 3: Relation between stress tensor (R shape factor ratio) and the stress regime.**

## 230 4 Tectonic zonation for the stress-strain analysis

To define the tectonic zones needed for the stress inversion and the strain analysis we use the Slip Model (De Vicente et al., 1988). This model provides unique values of the shape factor (k´) and the orientation of the shortening (or minimum extension) axis (Dey) for each individual focal mechanism. Interpolation of these values allows generation of continuous maps showing the variation of both parameters (e.g. de Vicente et al., 2008; Olaiz et al., 2009; Arcila and Muñoz-Martín,

2021). In this study, we built interpolated maps of the shape factor (k´) and the value for Dey using the blockmean module of the Generic Mapping Tools (Wessel and Smith, 1991; Wessel et al., 2013). The shape factor (k´) is a scalar that varies between 0 and 300. Therefore, the values are average normalized for each node and subsequently interpolated on a continuous surface. These maps are a powerful tool to better define different strain regions based on homogenous shape factor values and similar Dey trends. Thus, the grouping of the focal mechanisms is straightforward, allowing an

optimization of the results for the techniques designed for populations.

The three fundamental pieces of information that we considered for the delineation of the tectonic zones in the Iberian Peninsula are (a) the available information on the neotectonic (in this case, Alpine) structural deformational style (Fig. 1), (b) the density of the available data (Fig. 2), and (c) the dominant type of focal mechanism. The global and interpolated analysis of the data using the Slip Model allows us to consider, simultaneously, these parameters (Figs. 3 and 4).



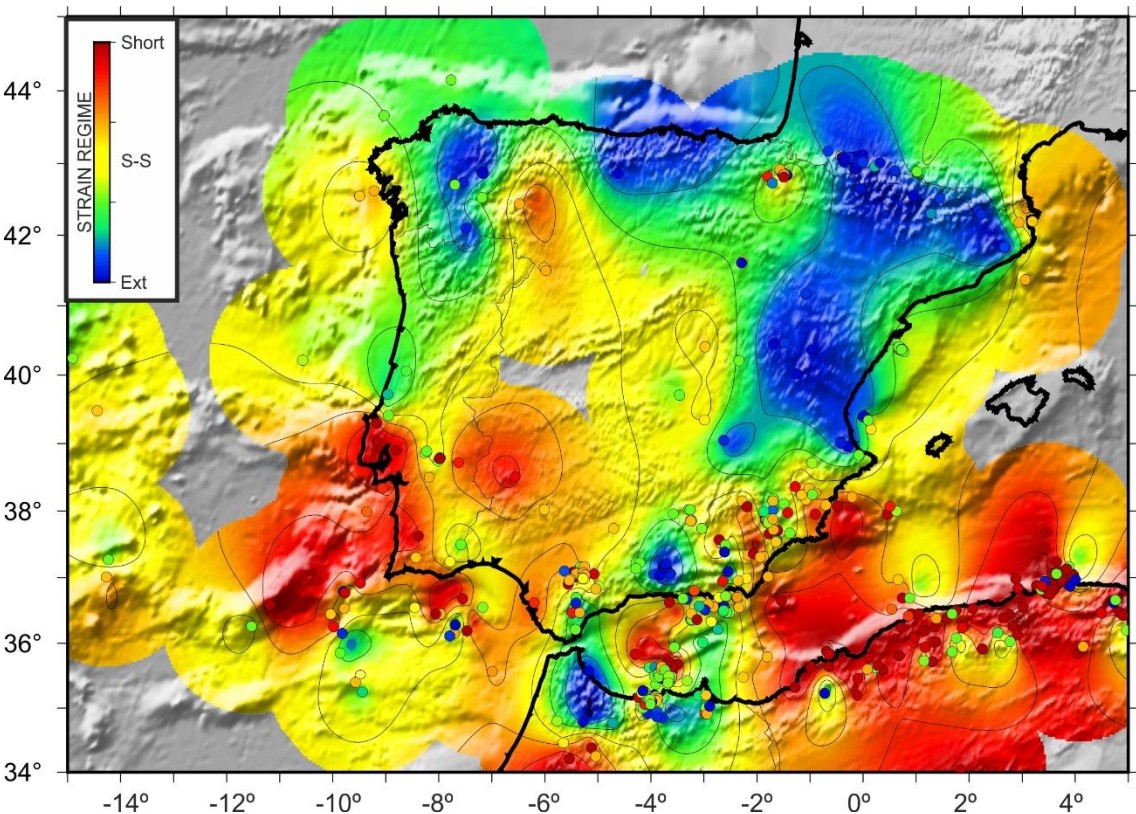


**Figure 3: Style of active deformation (strain regime) for Iberia determined from focal mechanisms. The map shows the shape factor (k´) determined using the Slip Model (Reches, 1983 and de Vicente, 1988). Each dot represents a focal mechanism. Short.: shortening. S-S: shear strain. Ext.: extension.**





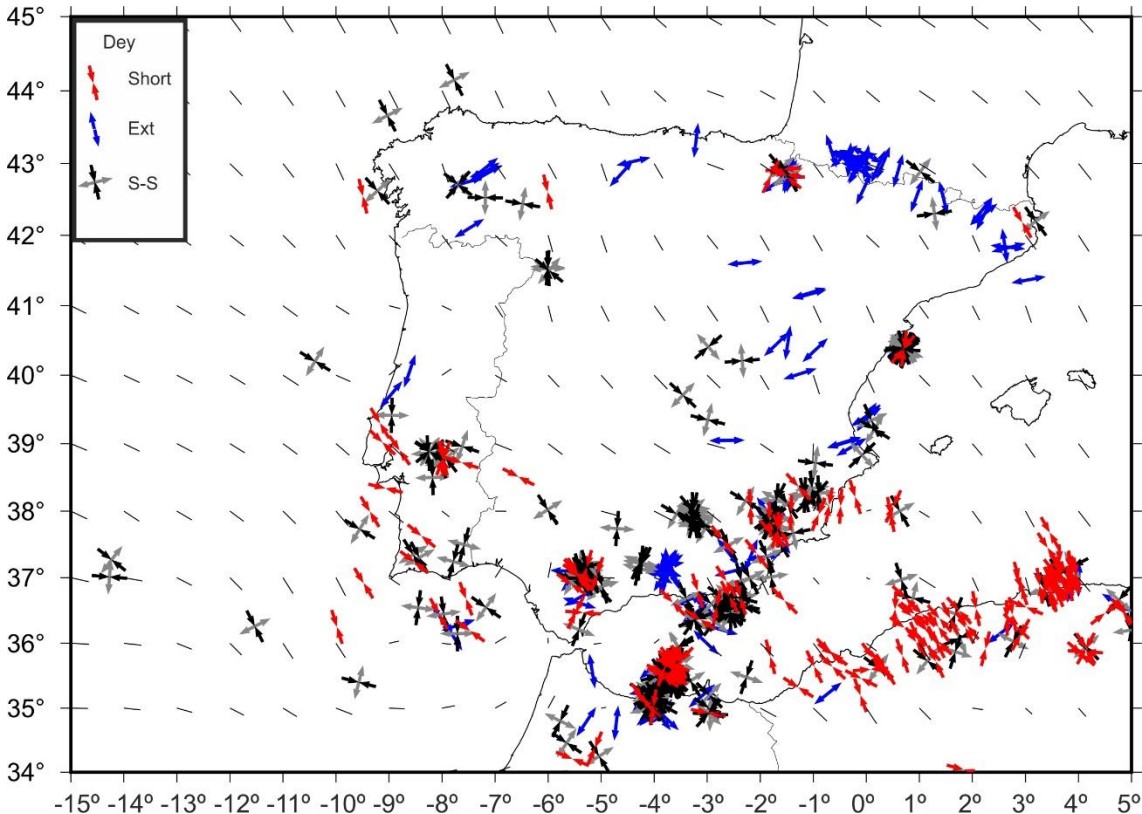

**Figure 4: Solutions of the Slip Model (by assuming that one principal strain axis is vertical) for all analysed focal mechanisms. Short, and Ext denote the principal horizontal shortening and extension axes, respectively, and S-S the shear strains. Black bars are Dey trend interpolations.**

Considering these criteria, we sub-divided Iberia into the following zones (Fig. 5):

In the Pyrenees we differentiated three zones: the Central Pyrenees (CP), with mostly normal faulting focal mechanisms, and two additional zones at the eastern and western ends, where there are more focal mechanisms: the Western Pyrenees (WP) and the Eastern Pyrenees - Northern Catalan Coastal Range (EPCE). Further west, on the N edge, we grouped together the earthquakes in North-Western Galicia (NWG), with numerous focal mechanisms in the central area, but also offshore. In the east of the peninsula, offshore near the coast, we differentiated two tectonic zones, North and South Valencia Trough (NVT and SVT, respectively). The former is from a local seismic network. Onshore, we have separated the focal mechanisms in the IC and south of the eastern SPCS (Central Basins, CB), as well as those in the tectonic zone near Lisbon, which we named the Western Central System (WCS). Further south but not yet at the active plate boundary, we defined Algarve (AL) the tectonic zone of. In the offshore Atlantic to the west, although the data are very scattered, focal mechanisms are similar and several correspond to earthquakes that occurred at mantle depths, but are not well constrained. Therefore, we grouped them in the tectonic zone Offshore Atlantic (OA). Further south, mainly offshore and following the plate boundary, we considered the Gorringe-Horseshoe (GH) and Gulf of Cadiz (GC) tectonic zones. The large number and variability of focal mechanisms in the Betics require smaller tectonic zones. Thus, and considering the predominance of normal faulting in the



Granada Basin (GB), we differentiated the Western Betics (WB) to its west, the Betics Antequera (BA) to its north, and the Eastern Betics (EB) to its east. In the Alboran Sea, we considered the North Alboran (NA), also with onshore focal mechanisms, and the Alboran Ridge (AR), at the probable plate boundary. Further south, on the African plate, in the Rif, we

considered the tectonic zones of Al Hoceima (ALH) and Rif (RF). Finally, in the Algerian Atlas, we analysed two populations, one further east, the Eastern Tell Atlas (ETA) and the other further to the west, the Western Tell Atlas (WTA). Fig. 5 shows the focal mechanisms in different colours for each area. We used the Rif (RF) focal mechanisms in the Slip Model analysis (Figs. 3 and 4), but any stress inversion had a good quality solution and, therefore, we are not going to comment on RF active tectonics (Fig. 5).

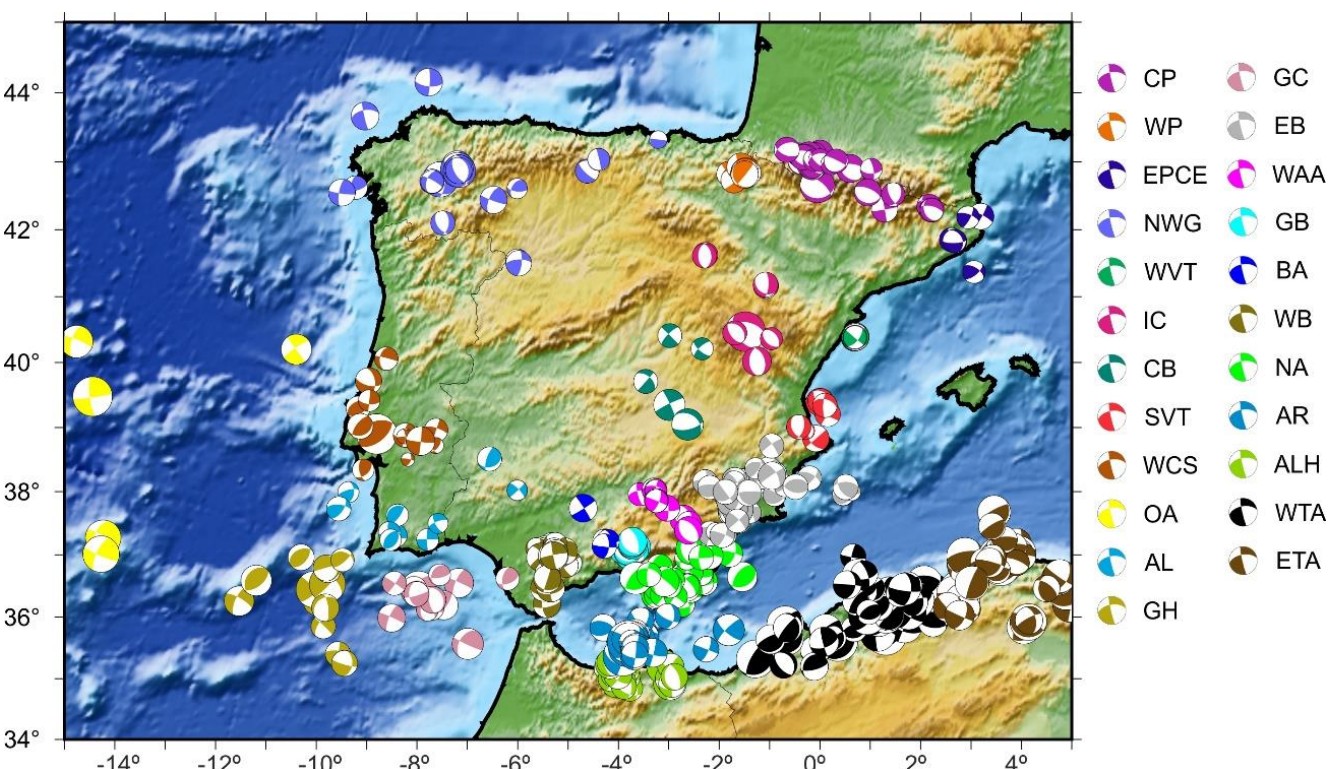


**Figure 5: Color-coded focal mechanisms for every considered tectonic zone: Central Pyrenees (CP); Western Pyrenees (WP); Eastern Pyrenees-Northern Catalan Coastal Range (EPCE); North-Western Galicia (NWG); Western Valencia Trough (WVT); Iberian Chain (IC), Central Basins (CB); Western Spanish Portuguese Central System (WCS); Offshore Atlantic (OA); Gorringe-Horseshoe (GH); Southern Valencia Trough (SVT); Algarve (AL); Eastern Betics (EB); Western Alcaraz Arch (WAA); Granada**
**Basin (GB); Northern Alboran (NA); Alboran Ridge (AR); Al Hoceima (ALH); Western Tell Atlas (WTA); Eastern Tell Atlas (ETA); Gulf of Cadiz (GC); Western Betics (WB); Betics Antequera (BA).**

## 5 Results

The results obtained following the methodologies described above are summarized in Tables 4 and 5 for all considered tectonic zones. For additional information see Appendix A and Appendix B.




| Label | Tectonic Zone | P trend | P plunge | B trend | B plunge | T trend | T plunge | Rupture type | fclvd | k value | N-SS | SS-R | R-N |
|---|---|---|---|---|---|---|---|---|---|---|---|---|---|
| | | | | | | | | | | | Minimum Rotation Angle (°) | | |
| CP | Central Pyrenees | 163 | 76 | **296** | 10 | 28 | 10 | N | -0.10 | 1.39 | 88.6 | - | - |
| WP | Western Pyrenees | **99** | 21 | 287 | 69 | 190 | 3 | SS | -0.31 | 3.52 | 82.1 | 74 | 72.7 |
| EPCE | Eastern Pyrenees | **344** | 17 | 129 | 70 | 251 | 11 | SS | -0.28 | 2.98 | 76.1 | - | - |
| NWG | NW Galicia | **328** | 14 | 138 | 76 | 237 | 2 | SS | -0.34 | 4.09 | 78.5 | 43.8 | 98.4 |
| WVT | Western Valencia Trough | **2** | 29 | 143 | 54 | 262 | 19 | SS-N | 0.06 | 0.84 | 16.3 | 25.6 | 25.5 |
| IC | Iberian Chain | 309 | 86 | **154** | 4 | 64 | 2 | N | 0.04 | 0.89 | - | - | - |
| CB | Central Basins | **276** | 6 | 31 | 77 | 185 | 12 | SS | -0.26 | 2.58 | 70.3 | - | - |
| WCS | Western SPCS | **323** | 3 | 229 | 60 | 55 | 30 | SS-R | 0.29 | 0.33 | 44.9 | 71.7 | 99.4 |
| OA | Offshore Atlantic | **152** | 22 | 297 | 64 | 57 | 14 | SS-N | 0.03 | 0.92 | - | - | - |
| GH | Gorringe-Horseshoe | **164** | 15 | 44 | 62 | 260 | 23 | SS-R | 0.33 | 0.26 | 49.6 | 68.7 | 90.2 |
| SVT | Southern Valencia Trough | 18 | 73 | **157** | 13 | 249 | 11 | N | -0.25 | 2.56 | 74.1 | - | - |
| AL | Algarve | **143** | 6 | 31 | 75 | 234 | 14 | SS | 0.41 | 0.13 | 80.3 | - | - |
| WAA | W Alcaraz Arch | **143** | 3 | 333 | 87 | 233 | 0 | SS | -0.23 | 2.30 | 87 | - | - |
| GB | Granada Basin | **250** | 76 | 139 | 5 | 48 | 13 | N | 0.08 | 0.77 | - | - | - |
| NA | Northern Alboran | **334** | 6 | 217 | 76 | 66 | 12 | SS | -0.15 | 1.63 | 88.3 | 49.7 | 100 |
| AR | Albora Ridge | **337** | 20 | 173 | 70 | 69 | 5 | SS | 0.19 | 0.52 | 64.8 | 76.4 | 98.4 |
| AH | Al Hoceima | **329** | 24 | 157 | 65 | 60 | 3 | SS-N | -0.08 | 1.30 | 50.8 | 58.2 | 76.6 |
| WTA | Western Tell Atlas | **324** | 15 | 231 | 10 | 108 | 71 | R | 0.29 | 0.33 | 67.4 | 80.4 | 105.5 |
| ETA | Eastern Tell Atlas | **336** | 26 | 69 | 8 | 175 | 63 | R | 0.08 | 0.77 | 77.8 | 73.5 | 96.3 |
| GC | Gulf of Cadiz | **161** | 15 | 291 | 67 | 66 | 17 | SS-R | 0.24 | 0.42 | 91.9 | 66.6 | 94.4 |
| WB | Western Betics | **324** | 14 | 226 | 29 | 77 | 58 | R-SS | 0.26 | 0.38 | 40.9 | 71.3 | 56.9 |
| BA | Betics Antequera | **305** | 30 | 101 | 58 | 209 | 11 | SS-N | -0.05 | 1.16 | - | - | - |

**Table 4: Summary of the combined seismic moment tensor results for all considered zones. Bold numbers are the trend the greatest horizontal shortening axis trend and underlined numbers are the trend of the most horizontal extensional axis.**

| Label | Tectonic zones | Dey | Shmax | σ1 trend | σ1 plunge | σ2 trend | σ2 plunge | σ3 trend | σ3 plunge | R | friction coeff. | cohesion | N(Nad) | PAM | SLIP |
|---|---|---|---|---|---|---|---|---|---|---|---|---|---|---|---|
| CP | Central Pyrenees | 119±12 | 121.5 | 196 | 83 | 121 | 1 | 3 | 4 | 0.35 | 0.6 | 0.0187 | 28/28 | 23.11 | 17.7 |
| WP | Western Pyrenees | 101±24 | 98.6 | 90 | 59 | 103 | 31 | 190 | 6 | 0.7 | 0.1 | 0.0351 | 12/14 | 33.63 | 30.15 |
| EPCE | Eastern Pyrenees | 166±20 | 165.1 | 349 | 48 | 158 | 41 | 253 | 6 | 0.42 | 0.3 | 0.0312 | 6/6 | 36.1 | 21.1 |
| NWG | NW Galicia | 155±14 | 147.8 | 290 | 79 | 150 | 9 | 59 | 5 | 0.6 | 0.4 | 0.0354 | 17/21 | 17.96 | 27.53 |
| WVT | Western Valencia | 0±14 | 170.8 | 359 | 35 | 125 | 39 | 245 | 31 | 0.34 | 0.2 | 0.0096 | 11/12 | 15.92 | 9.99 |
| IC | Iberian Chain | 166±9 | 141.8 | 54 | 87 | 148 | 0 | 238 | 3 | 0.43 | 0.1 | 0.0114 | 8/8 | 24.37 | 16.44 |
| CB | Central Basins | 78±12 | 84.3 | 270 | 55 | 80 | 34 | 173 | 4 | 0.61 | 0.4 | 0.0055 | 6/6 | 23.33 | 6.19 |
| WCS | Western SPCS | 140±19 | 137.2 | 134 | 13 | 209 | 50 | 54 | 37 | 0.38 | 0.1 | 0.0451 | 20/20 | 33.65 | 29.29 |
| OA | Offshore Atlantic | 140±28 | 133.9 | 151 | 15 | 166 | 72 | 44 | 10 | 0.92 | 0.1 | 0.0059 | 5/5 | 18.75 | 2.97 |
| GH | Gorringe-Horseshoe | 133±26 | 157.3 | 159 | 24 | 251 | 5 | 352 | 65 | 0.41 | 0.1 | 0.0204 | 8/13 | 38.82 | 11.3 |
| SVT | Southern Valencia | 166±21 | 147.8 | 129 | 80 | 149 | 9 | 238 | 3 | 0.5 | 0.1 | 0.0029 | 8/10 | 20.72 | 6.69 |
| AL | Algarve | 117±20 | 146.3 | 140 | 17 | 334 | 72 | 232 | 4 | 0.14 | 0.6 | 0.0658 | 11/11 | 38.22 | 14.681 |
| EB | Eastern Betics | 163±18 | 169.1 | 171 | 13 | 112 | 66 | 256 | 20 | 0.35 | 0.01 | 0.0033 | 47/47 | 31.13 | 29 |
| WAA | W Alcaraz Arch | 160±13 | 152.5 | 153 | 8 | 325 | 82 | 62 | 1 | 0.8 | 0.6 | 0.0614 | 14/14 | 22.43 | 16.13 |
| GB | Granada Basin | 136±13 | 119 | 263 | 84 | 123 | 6 | 33 | 5 | 0.11 | 0.5 | 0.0152 | 16/16 | 18.92 | 14.72 |
| NA | Northern Alboran | 163±20 | 151.2 | 149 | 14 | 187 | 72 | 62 | 10 | 0.73 | 0.01 | 0.0024 | 42/42 | 28.12 | 24.24 |
| AR | Alboran Ridge | 166±17 | 161.8 | 161 | 22 | 159 | 68 | 251 | 1 | 0.34 | 0.01 | 0.0023 | 62/62 | 26.79 | 19.8 |
| AH | Al Hoceima | 147±17 | 153.6 | 152 | 36 | 159 | 54 | 64 | 4 | 0.69 | 0.1 | 0.0186 | 41/41 | 21.63 | 19.77 |
| WTA | West Atlas | 146±15 | 145.3 | 145 | 12 | 199 | 71 | 58 | 15 | 0.14 | 0.2 | 0.0366 | 44/44 | 25.99 | 16.83 |
| ETA | East Atlas | 166±17 | 145.6 | 149 | 20 | 164 | 13 | 185 | 65 | 0.4 | 0.01 | 0.0032 | 52/52 | 25.52 | 23.09 |
| GC | Gulf of Cadiz | 148±12 | 147.7 | 150 | 5 | 251 | 64 | 58 | 25 | 0.19 | 0.1 | 0.0144 | 10/12 | 23.09 | 16.14 |
| WB | Western Betics | 148±22 | 140.7 | 139 | 16 | 212 | 45 | 63 | 40 | 0.18 | 0.1 | 0.031 | 28/28 | 28.68 | 24.54 |
| | Western Betics > 20 km | 148±14 | 78.7 | 114 | 17 | 113 | 72 | 204 | 0 | 0.26 | 0.01 | 0.0041 | 6/7 | 24.27 | 23.28 |
| BA | Betics Antequera | 102±11 | 104.8 | 105 | 3 | 109 | 87 | 15 | 0 | 0.91 | 0.7 | 0.0061 | 5/5 | 13.04 | 7.97 |

**Table 5: Summary of the stress inversion results for all considered zones. Dey defines the maximum shortening trend. R defines the stress ratio. N denotes number of events, and Nad is number of events for the given solution (adjusted). PAM is the principal angles misfit and SLIP is the slip misfit.**

## 5.1 CP Central Pyrenees

The focal mechanisms in this area are mostly normal, although some strike-slip faults are also observed. The seismic

moment release is controlled by the normal fault events giving rise to a combined moment tensor close to double couple, with a k value of 1.39 and an fclvd of 0.1. The Kagan angle between the combined moment tensors of normal and strike-slip earthquakes has a value close to 90°, indicating an almost pure permutation between the B and P axes. The orientations of the principal strain axes (T and B) are consistent with each other and orthogonal to the strike of the mountain range, although





these axes show some variability with two predominant families, one N020°E-N030°E and the other N040°E-N050°E, which
could indicate the activation of normal fault families with slightly different orientations (see Appendix A).

The 28 focal mechanisms located in the central part of the Pyrenees provide a very consistent stress inversion solution (Fig.
7 CP) that indicates a close to uniaxial normal faulting stress regime following an N005°E-N030°E direction, sub-
perpendicular to the topographic axis of the range. The solution like that obtained by de Vicente et al. (2009) and Asensio et
al. (2012). The latter authors also analysed GPS data with similar results (extension perpendicular to the range of 2.5±0.5
305    nanostrain yr-1), which they attributed to post-orogenic collapse. This normal faulting stress regime would account for the
seismic activity of the main active normal faults present in the area, such as the Lourdes fault, which has a 50 km trace and a
50 m fault scarp (Alasset and Meghraoui, 2005; Lacan and Ortuño, 2012), as well as those of Bedous, Laruns, Pierrefitte and
Pic de Midi du Bigorre (Lacan 2008). The latter authors attribute the seismicity in this part of the Pyrenees to a process of
isostatic rebound. In this area, we also include the focal mechanisms in the easternmost part of the area, obtained close to the
Têt Fault, which is 120 km long and accommodated right-lateral movement from the Miocene to the Upper Pliocene
(Cabrera et al., 1988), and which during the Plio-Quaternary seems to have had mainly extensional movement (Briais et al.,
1990). In any case, the three nearby focal mechanisms show normal faulting, consistent with the other mechanisms used for
the inversion. Recent studies, calculated with polarities and temporary seismic networks, provide very similar focal
mechanisms (Ruiz et al., 2023).

**5.2 WP Western Pyrenees**

Seismicity in the western Pyrenees is clustered in the south of the mountain range, in the vicinity of the city of Pamplona.
This seismic activity is characterised on the one hand by normal faulting, with a strike-slip component, and on the other by a
series of oblique events with an important strike-slip component. This component is important in the area, with the combined
moment tensor having mainly shearing characteristics, although with fclvd values of 0.31 and a k of 3.52, which indicates a
prolate-type seismic deformation tensor shape. The strain axis of the combined mechanism is N011E and that of maximum
horizontal shortening (B and P permuting) between N090°E-N110°E.

The stress inversion (Fig. 7 WP) results also in a normal faulting stress regime as in CP (N098E), although the stress tensor
shows characteristics closer to strike-slip, with σ2 at N103°E, which allows activating WNW-ESE to E-W normal faults,
such as those of Leiza, Aralar and Roncesvalles (Lacan and Ortuño, 2012), but also NE-SW strike-slip faults, such as the
Pamplona Fault, which is 125 km long (Ruiz et al., 2006). This is a vertical fault, inherited from the Late Variscan, which
also controlled Mesozoic and Tertiary sedimentation (Ruiz et al., 2006) and which was reactivated as an oblique ramp during
the Pyrenean shortening (Vergés, 2003).



### 5.3 EPCE Eastern Pyrenees-Northern Catalan Coastal Range

This tectonic zone which contains only 6 focal mechanisms was differentiated from the Central Pyrenees population because

here, as in the westernmost part, there seem to be more strike-slip faulting mechanisms (the easternmost ones). Half of the population, with epicentres close to the axis of the mountain range, present pure normal faulting mechanisms while the other half has a thrust faulting component. The Kagan angle between the combined normal and strike-slip faulting mechanisms is close to 80°, suggesting a permutation between the P and B axes. The combined moment tensor departs from a pure double couple, with an fclvd of 0.28 and a k-value of 2.98. The shape of the ellipsoid is therefore of prolate type, with the strain axis

oriented at N72E and the maximum shortening almost horizontal at N163°E.

The stress inversion (Fig. 6 EPCE) indicates a more normal faulting stress regime (N165E) concerning CP, similar to that obtained by Goula et al. (1999) with normal strike-slip stresses (different from the one provided by these authors, which was more thrusting). The onshore focal mechanisms would indicate that the structures associated with the recent NW-SE Olot volcanism, the Amer and Empordà faults (Souriau and Pauchet, 1998; Lacan and Ortuño, 2012) present normal-type

movements, which would have been responsible for the seismic crisis between 1427 and 1428 that caused considerable damage (Olivera et al., 2006). Active faults further south, in the Catalan Coastal Range, such as the Camp (Masana, 1996) Montseny and Pla de Barcelona (Perea et al., 2020) seem to be more related to the opening of the Valencia trough than to the Pyrenees. In any case, the stress solution is very similar to those found to the south in IC, WVT and SVT, so it seems to be less related to the local processes affecting the Pyrenees.

### 5.4 NWG North-Western Galicia

The Pyrenean orogen spans across northern Iberia to the Cantabrian Cordillera (Cantabrian Pyrenees), Galicia and offshore, as far as the Galicia Bank (Fig. 1). The southern part of the Bay of Biscay, the closest to Iberia, also appears to be affected by the deformation of the Pyrenees (e.g. Boillot and Malod, 1988) where the shortening took place from the Upper Eocene to the Middle Miocene (Gallastegui and Pulgar, 2002). After the Middle Miocene, much of the neotectonic activity in Galicia

and Northern Portugal was concentrated in the left-lateral strike-slip NNE-SSW fault system of Vilariça-Bragança, Regua-Verín and Monforte-Orense (Cabral, 1989; de Vicente & Vegas, 2009; Martín González et al., 2012), which, although not strictly a part of the Pyrenees, are closely related to its evolution, as well as to that of the SPCS.

The focal mechanisms in this zone can be grouped into two families, one of normal faulting type releasing most of the seismic moment, and one of oblique strike-slip faults, some with normal components and other with reverse faulting

components (see Appendix). By using the Kiratzi & Papazachos (1995) approximation, which gives equal weight to all combined events in the tensor, we avoid the dominance of events with a much higher energy release than the others. The resulting combined moment tensor shows a prolate form, with k values close to 7 and an fclvd of 0.39, dominated by ENE-WSE extension. The strain axes are very coherent in all focal mechanisms, with the combined one having an orientation of





N057°E. In the horizontal shortening axes (P and B permuting with a rotation angle of 84°) there is more variability, with the
P axis showing an orientation of N147°E.

The 18 focal mechanisms included in the stress inversion (Fig. 6 NWG) are of strike-slip and normal-directional types, resulting in a common normal faulting stress regime according to N147°E and swapping between σ1 and σ2 (σ2 at N150E). It should be noted that the most normal faulting mechanisms are concentrated in the area where small sedimentary basins developed during the Miocene, namely in the interior of Galicia (de Vicente et al., 2011), while the offshore region is
dominated by strike-slip faulting.

We were only able to obtain one focal mechanism in the Cantabrian Mountains (Cantabrian Pyrenees), which shows NW-SE normal faulting. This focal mechanism could indicate that important faults in that orientation, such as the Ventaniella-Ubierna faults, accommodate normal faulting deformation today. A study using a local seismic network (10 stations) on the Ventaniella Fault determined focal mechanisms for earthquakes of Mw<2 without obtaining a consistent pattern (López-
Fernández et al., 2018), so the results in this area are not conclusive. The two focal mechanisms with the highest double-couple component (above 80%) calculated by del Pie Perales (2016) are normal faulting solutions and the focal planes are compatible with the strike-slip fault system. Recently, on September 30th, 2023, a Mw=3.6 thrusting focal mechanism was reported by the IGN in Villamejil (León).

### 5.5 NVT Western Valencia Trough

The Valencia trough, between the Mediterranean rim of Iberia and the Balearic Islands, shows a complex succession of partially inverted Mesozoic rifting events during the Cenozoic (Etheve et al., 2018). During the last 30 Myr, the emplacement of the Calabrian-Tyrrhenian subduction zone with trench retraction and back-arc extension produced intense extension in the Levantine sector of Iberia (Faccenna et al., 2004) during the Neogene (Roca and Guimerà, 1992), forming a broad rifting zone with the development of many horsts and grabens. This extension is moderately active today (Perea et al.,
380    2020).

The Castor $CO_2$ storage project generated a sequence of apparently triggered seismicity. In addition to the moment tensors calculated by the IGN, several groups published focal mechanisms (Cesca et al., 2021; Villaseñor et al., 2020). The moment tensors present mostly similar solutions in both studies, in terms of plane orientations, although the depth and epicentral location vary significantly. Due to the better epicentral relationship with the previously identified NE-SW striking faults, the
results used here were those by Villaseñor et al. (2020). These mechanisms show strike-slip fault offset with a strong normal component, very similar to those obtained by Goula et al. (1999) for the 1991 and 1995 events from P-wave polarities. The combined moment tensor shows an N082°E trending T axis and an immersion of 19°. The P axes show somewhat more scatter, still, the P axis of the combined moment tensor is oriented N002°E, with an immersion of 28°. The value of k obtained is 0.8, which indicates an oblate-type tensor shape, with N-S shortening and E-W extension.





The stress inversion of the 12 focal mechanisms (Fig. 6 WVT) provides a very consistent solution showing a normal faulting stress regime compatible with strike-slip and oriented N170°E. This stress regime activates moderately dipping NE-SW and NW-SE faults, such as those of the Gulf of Rosas, the Amposta Basin, Cape Cullera and the Columbretes Basin (Perea et al., 2020), which affect Plio-Quaternary sedimentary units (Perea et al., 2012). Onshore, the Camp fault (Masana, 1996) can also be activated by this type of stresses. The stress solution for this zone is very similar to that of the SVT.

## 5.6 IC Iberian Chain


The IC is part of the Iberian intraplate orogen, with a main Oligocene-Lower Miocene age of deformation, which inverted Permo-Mesozoic rifts (Álvaro et al., 1979). From the Upper Miocene onwards, its activity is linked to the opening of the Valencia trough (Roca and Guimerà, 1992) and therefore shows a similar evolution to the WVT. To the W, the extension deactivated the thrusts that uplifted the SPCS north of Madrid. The recent extensional process formed Neogene-Quaternary

basins associated with N-S to NW-SSE normal fault activity (Simón, 1989). The associated stress field, from recent fault data, shows triaxial normal faulting, with $\sigma_3$ oriented ENE-WSW (Arlegui et al., 2005). The most important active faults are those bounding the Jiloca graben (Sierra Palomera, Calamocha, Daroca, Munébrega faults) and the Teruel graben (Sierra del Pobo, Valdecebro and Concud) (Simón, 2020).

The 8 focal mechanisms in this tectonic zone are normal faulting events, with a few showing a strike-slip component, which

combined give rise to a pure double-couple tensor with a horizontal T-axis oriented N064°E and B-axis oriented N154°E. The individual moment tensors are divided into two distinct families (see Appendix), with T- and B-axis rotations of about 30°, probably due to the activation of complementary normal fault families.

The focal mechanisms used for the stress inversion, located in the north and centre of the IC, yield a triaxial normal faulting stress regime solution (Fig. 6 IC), with $\sigma_3$ oriented N058°E, similar to that of the WVT and compatible with the results

obtained from recent faults analyses (Arlegui et al., 2005). Thus, the active stress field can activate the aforementioned normal faults (Simón, 2020).

## 5.7 CB Central Basins

In central Iberia, south of the SPCS and W of the IC, in the Cenozoic basins of Madrid and La Mancha (CB), we find 6 focal mechanisms, mainly showing strike-slip and normal faulting. Most of the strike-slip solutions correspond to small seismic

moment releases. The combined moment tensor shows an fclvd of 0.25 and a value of k of 2.58, corresponding to a prolate-type tensor shape dominated by a strain axis oriented N005°E and an axis of maximum shortening nearly oriented E-W.

The stress inversion of the focal mechanisms (Fig. 6 CB) indicates a normal faulting stress regime solution with a small strike-slip component, with $\sigma_3$ almost perpendicular to that obtained for the CI, at N173°E, which also differs from most of the areas considered. However, geodetic velocities obtained from GPS data in this area show a clear westward displacement





(considering Eurasia fixed) with rates of 2 - 3 mm/yr. (Cannavò and Palano 2016; Neres et al., 2018), different from the
NW-SE movement seen further to the south. The results published by Khazaradze et al. (2018) further support a dominant
strike-slip faulting regime. On the other hand, and as a differential element, the area is characterised by volcanism, mainly
sodic alkaline and ultra-alkaline rocks, with radiometric ages between 4 Ma to less than 0.7 Ma (Ancochea and Huertas,
2021) (Campos de Calatrava volcanism), which has been related to a gentle folding of the Iberian lithosphere and the
presence of an anomalous low-density sub-crustal block below the volcanic zone (Granja Bruña et al., 2015). It is therefore
possible that the calculated stress tensor is somehow influenced by this recent volcanic activity. An edge effect related to
large-radius extension and uplift in the CI cannot be ruled out (Casas-Sáinz and de Vicente, 2009), nor is an indentation
effect related to the Betics.

### 5.8 WCSWestern (Spanish Portuguese) Central System

As mentioned before, the central orogenic belt extends to the west, as far as Lisbon and offshore into the Estremadura Spur.
In its Portuguese stretch, the SPCS is an active mountain range (Cabral, 2012), with reported damage to structures that were
translated into earthquake intensities such as the Benavente earthquake in 1909 with an estimated moment magnitude Mw=
6.0 and a significant thrust faulting component (Stich et al., 2005; Fonseca & Vilanova, 2010). The thrust fauts have
predominantly NE-SW strikes with associated NW-SE shortening (de Vicente et al., 2018). To the east of Portugal, in the
westernmost Spanish sector, and along the NE part of the Messejana-Plasencia fault, instrumental seismicity is scarce,
although there is evidence of end-Cenozoic and Quaternary deformation (de Vicente et al., 2022).

The focal mechanisms in this tectonic zone are predominantly reverse or strike-slip faulting events, the former showing a
higher release of seismic energy. There are some normal faulting events whose T and B axes are kinematically compatible
with each other. Considering the combined mechanisms of strike-slip and reverse faulting, we calculated the value of the
minimum rotation angle close to 70°, thus constituting a T and B permutation.  The orientation of the axis of maximum
shortening of the combined moment tensor is N146°E and the axis of maximum extension is oriented N056°E. The overall
combined moment tensor, considering all earthquakes in this zone, is of oblate type, with a low k value of 0.325 and an fclvd
of 0.29; thus being dominated by NW-SE shortening.

The stress inversion for this tectonic zone (Fig. 6 WCS) was computed using 20 focal mechanisms and resulted in a clear
thrust faulting stress regime solution with a small strike-slip component, with σ1 oriented N134°E. This stress regime
activates NE-SW striking thrust faults and left lateral NNE striking and right-lateral ESE striking strike-slip faults. Examples
include the Vilariça-Bragança and Regua-Verín faults with left-lateral movement, faults from S Galicia (NWG, with σ2 at
N150E) to the Serra da Estrela with a thick-skinned tectonic style without tectonic inversion, and the SPCS (Cabral, 2012)
with σ1 at N134°E. The thrusts bordering the Lusitania Basin to the S and the Lower Tagus Cenozoic Basin to the north are
also active with kinematics that are directly related to the inferred stress tensor. Focal mechanisms to the east show mainly





strike-slip faulting in the Arraiolos and Évora seismic zones, within the Ossa-Morena zone of the Variscan basement, consistent with right-lateral motion on N065°E faults (Matos et al., 2018).





**Figure 6 Results of the stress-strain analyses for different zones: a) Right Dihedra solution. b) Rose diagram of the Deys obtained from the Slip model. C) Pitch/Dip plot for the neo-formed nodal planes obtained from the Slip Model. d) Stress Inversion Results. e) Variability of the three principal stress axes of the stress inversion. CP Central Pyrenees; WP Western Pyrenees; EPCE Eastern Pyrenees-Northern Catalan Coastal Range; NWG North-Western Galicia; WVT Western Valencia Trough; IC Iberian Chain, CB Central Basins; WCS Western Spanish Portuguese Central System.**

### 5.9 OA Offshore Atlantic (ES Estremadura Spur)

The focal mechanisms in the Atlantic offshore, to the north of the active plate boundary, are spatially scattered over a wide area and some appear to be deeper than the Moho (although the depth of these earthquakes is poorly constrained by the mainland seismic networks). We grouped all these events given their similar characteristics. Between the Tagus and the Iberian abyssal plains, there is a structural high, the Estremadura Spur, with E-W to NE-SW thrusts, which is the offshore extension of the SPCS, affecting the intermediate crust (Terrinha et al., 2009). Between this tectonic uplift and the Galicia Bank to the north, NNE-SSW faults seem to dominate, with kinematics similar to those onshore (Vilariça-Bragança and Regua-Verín fault systems). In the south, there are some focal mechanisms close to the Hirondelle structural high, to the north of the Gloria fault. In the Estremadura Spur, a NNW-SSW shortening regime dominated the main structuring of the range during the Palaeocene-Miocene (Pereira et al., 2021), although the most recent and active deformation appears to be transpressional (Neves et al., 2009).

There are here 6 strike-slip focal mechanisms in this zone, some of them with a small normal component. These moment tensors show some variability in their principal axes, especially in the P axis. Therefore, the combined moment tensor presents k values below 1, an fclvd greater than 0 (Fig. 7 OA), and a tendency to have an oblate shape, with the P axis having a dominant N152°E orientation. The extension T-axis is in the N057°E direction. The stress inversion indicates a strike-slip stress regime with a normal component with σ1 at N131°E and σ3 at N044°E, which activates right-lateral ESE-WNW faults, like the major strike-slip Gloria fault in the North Atlantic that defines the present-day plate boundary between Eurasia and Africa.

### 5.10 GH Gorring-Horseshoe

At the active plate boundary, and between the NE-SW structural highs (pop-ups and restraining steps with thick-skinned tectonic styles) of Gorringe and Horseshoe, there are 13 reverse and strike-slip focal mechanisms, several of which below the Moho, that indicate a thrust-faulting stress regime (Fig. 7 GH) with σ1 at N159°E. These focal mechanisms are mainly strike-slip with a vertical component and reverse faults with some horizontal component. There are also normal-type mechanisms whose T axes are kinematically compatible with the T axes of the other events. The orientations of the axes of maximum horizontal shortening and extension show quite some variability, with the T axes giving rise to two well-differentiated families, one N040°E-060°E and the other N070°E-N090°E (see Appendix). The more E-W trending axes seem to be more associated with the strike-slip faults in the centre and south of the region, while the reverse fault events in





the northern part have NE-SW B-axis orientations. The combined moment tensor presents an oblate type shape, dominated by shortening, with a k value of 0.623 and an fclvd of 0.14, showing a significant distribution of deformation in different structures. The orientation of the P and T axes are N155E and N65E, respectively.

Deformation along the Gloria fault (zone OA) is purely strike-slip and locally transtensional, whereas between the Gorringe and Horseshoe tectonic uplifts (zone GH) it is compressional (Zitellini et al., 2009). The most prominent structure, the Gorringe Bank, is bounded by NE-SW crustal thrusts, the most important one on its N edge. This structure was the source of destructive landslides during the Miocene, with renewed Plio-Quaternary activity (Gamboa et al., 2021). Several earthquakes have hypocentres at upper mantle depths (>20 km), implying that the observed surface thrusts must be linked to structure at mantle levels (Grevemeyer et al., 2017). Active deformation has reactivated and inverted former Mesozoic normal faults (García-Navarro et al., 2005) and appears partitioned between NW-SE thrusts and WNW-ESE to W-E strike-slip faults (Terrinha et al., 2009). Further south, the Coral Patch Ridge shows a similar tectonic structuring (Martínez-Loriente et al., 2013). The inferred stress tensor activates pure NE-SW thrusts and right-lateral NW-SW reverse-strike-slip and WNW-ESE strike-slip faults, which may be the tear faults of the thrusts (Ferranti et al., 2014).

### 5.11 SVT Southern Valencia Trough

In the easternmost onshore part of the Prebetic front, on the undeformed foreland, Betic compression ceased 10 Myr ago. This area is structured in a series of horsts and grabens, about 25 km long and 5 km wide (Navarrés, Tous), which triggered a diapirism of the Triassic salt (Keuper facies) when the extension associated with the opening of the Valencia Trough was imposed in the area (De Ruig, 1995). The main trend of the grabens is perpendicular to the Betic front (NW-SE), although there are also parallel grabens, indicating triaxial extension. Their position explains why they are not plug-type diapirs, but rather linear ones, which were controlled by multiple faulting episodes (Jackson and Hudec, 2017). The diapirism is still active, affecting Quaternary materials (Gutierrez et al., 2019). The coastline also changes in this sector to NW-SE, while further north, up to the Pyrenees, it has a NE-SW orientation, which is that of the main faults in the Valencia Trough, as in WVT. This orientation, transversal to the main trough, must have been influenced by the Betic front. The offshore Cabo de la Nao fault would be one of the main faults of this transverse fault system (Maillard and Mauffret, 1999).

We have 10 focal mechanisms in this tectonic zone. Most of them indicate normal faulting, but there are also strike-slip focal mechanisms. The combined tensors present minimum rotation angles of 74°, showing a permutation between the B and P axes. The combined tensor has an fclvd value of -0.25 and a clearly prolate tensor shape with k = 2.56, showing the predominance of the N069°E-oriented extension. The stress tensor obtained from the inversion (Fig. 7 SVT) clearly indicates a normal faulting stress regime, with σ3 at N058°E, able to activate the transverse fault system of the southernmost part of the Valencia Trough in the foreland of the Betic orogen. The stress tensor solution is very similar to those of IC, WVT and EPCE.



### 5.12 AL Algarve

The southwestern corner of the Iberian Massif, in the Portuguese Algarve, also comprises a band of Mesozoic and Cenozoic materials, the Algarve Basin, which was inverted during the Cenozoic by N-S to NW-SE shortening. The NW-SE shortening developed later, from the Miocene to the present (Terrinha, 1998; Ramos et al., 2015) and activated E-W to NE-SW striking thrust faults. Still in this zone but north of the basin there are important strike-slip faults, such as the southernmost part of the Messejana-Plasencia Fault (NE-SW, left lateral) and the São Marcos - Quarteira Fault (NW-SE, right lateral). According to GPS data, the southern block of this fault shows a significant movement to the NW relative to the northern block (Cabral et al., 2017) and presents a clear activity during the Quaternary (Cabral et al., 2019). Offshore, the most relevant tectonic structure is the Portimao Bank, bounded by E-W thrusts and co-located with a Cretaceous magmatic intrusion (Terrinha et al., 2009; Vázquez et al., 2015; Neres et al, 2018). Throughout the Algarve basin, there is active salt tectonics (Matías et al., 2011).

We can group the focal mechanisms in this zone into pure and reverse strike-slip faulting. Although there is some scatter in the orientation of the T-axes, the P-axes are very consistent, with the combined strain tensor having an orientation of the maximum shortening axis at N143°E. The combined moment tensor, however, is far from a double couple, showing a very high fclvd of 0.4 and an oblate tensor shape, with a k value of 0.132. This k-value is the lowest value of all obtained solutions, showing the dominance of the shortening deformation tensor.

The inversion of the 11 mechanisms indicates a thrust-faulting stress regime (Fig. 7 AL) with σ1 at N140°E, which activates NE-SW striking thrust faults and left-lateral N-S and NW-SE striking right-lateral east-west striking strike-slip faults. The solution is very similar to that obtained for the WCS, showing that this type of stresses predominates throughout the SW corner of Iberia, to the W of the Betic front. This observation, together with the absence of thrust faulting stresses in the Iberian Betic foreland, indicates that one of the effects of the emplacement of the Alboran Domain to the W was the mechanical decoupling between Iberia and Africa (Nubia).

### 5.13 EB Eastern Betics

The easternmost part of the Betics shows a significant level of seismic activity. This area was affected by extensional tectonics during the opening of the Algero-Balearic Basin throughout the Miocene, the most important extensional episode being the one that took place in the Serravallian-Tortonian (Comas et al., 1999); although it could maintain a certain transcurrent character (Montenat and Ott d'Estevou, 1999). Since the Late Miocene, the dominant tectonics in the region have been related to shortening, producing since then the tectonic inversion of the Miocene basins (Sanz de Galdeano, 1990; Martínez-Díaz, 1998). The active continental indentation of the Águilas Arc to the NW is related to this shortening, linked to the collision of Africa and Eurasia, in which the arc would form part of the African crust (Ercilla et al., 2022; Tendero, 2022). The overthrusting arc is bounded to the E by the right-lateral strike-slip Mazarrón Fault and to the W by the left-lateral strike-slip Palomares Fault. The latter extends to the SW on the Serrata-Carboneras Fault to the offshore Alborán Sea.



The indentation implies a progressive tilting towards the SE of the whole arc (Ercilla et al., 2022; Tendero-Salmerón, 2022).
In the frontal part of the arc, once in the Iberian crust, the deformation is accommodated in a left-lateral NE-SW transpressional corridor, the Eastern Betic Shear Zone (EBSZ) (Alhama Fault), in continuity with the Trans-Alboran Transpressional Shear Zone (TASZ). It is in this shear zone that the most active faults of the Iberian Peninsula are found, with deformation rates between 0.5 - 1.5 mm/yr as indicated by palaeoseismological and geodetic methods (Herrero-Barbero et al., 2020; Gomez-Novell et al. 2022, Moreno et al., 2015, Echeverría et al., 2015, Martín-Banda et al., 2016). The
simultaneous activity of these two macrostructures implies the presence of a deformation partitioning process (s.s) potentially with a certain degree of mechanical decoupling between the African and Iberian crusts.

The focal mechanisms in this tectonic zone are dominantly strike-slip and reverse events, with the most frequent having a certain oblique character, while some normal faulting events with a strike-slip component are also present. The minimum rotation angle between the strike-slip moment tensor and the extensional moment tensor is about 47o, indicating the
predominance of the transcurrent character also in the normal mechanisms (see Appendix). The combination of all the mechanisms gives rise to an oblate-shaped strain tensor, dominated by shortening, with fclvd values of 0.21 and a k value of 0.46. The P axis has an N172°E orientation, although the P axes of all mechanisms have a large scatter, controlled by the location of the events (see Appendix). This variability of the shortening axes may be influenced by the presence of large crustal structures that generate local block rotations (Martínez-Díaz, 2002) or due to local deformation distribution patterns
in the area (Alonso-Henar et al., 2019).

The inversion of the 47 focal mechanisms (Fig. 7 EB) provides a thrust-to-strike-slip faulting stress regime with $\sigma 1$ at N171E, which can activate NE-SW to ENE-WSW striking  thrust faults, right-lateral NW-SE and left-lateral NNE-SSW striking strike-slip faults. There are also three NW-SE striking normal fault mechanisms in the area, which have a common $\sigma 3$ (NE). This stress tensor solution explains the simultaneous movement of the Aguilas Arch and Alhama fault.

**5.14 WAA Western Alcaraz Arch**

This tectonic zone includes the westernmost part of the Alcaraz (or Prebetic) arch and a sector of the Guadix Basin, which has recorded intense seismic activity in recent years. The structural morphology is like that of other arcs in the Betics (Águilas) and the Alborán Sea, which are interpreted resulting from as processes of tectonic indentation, which in this case would be incipient (Tendero-Salmerón, 2022). The seismicity is concentrated along two NW-SE to ESE-WNW right-lateral
strike-slip faults, the Tiscar and Guadiana Menor faults (Tendero-Salmerón et al., 2020), which affect Quaternary materials and the Cenozoic sediments of the Guadalquivir Cenozoic Basin. The Torreperogil - Sabiote seismic series, with strike-slip focal mechanisms, also seems to be related to faults possibly rooted in the basement (Pedrera et al., 2013). To the south, the intramountain Guadix Basin, with a general NW-SE trend, appears to be filled with Tortonian to Pleistocene sediments (Pla-Pueyo et al. 2009) bounded by NW-SE normal faults (Alfaro, et al., 2008; Sanz de Galdeano et al., 2012).





Of the 14 focal mechanisms, two are normal faulting events (in the Guadix Basin) and the rest are strike-slip events (also one in Guadix). The orientation of the T axes in both types of mechanisms is consistent, with the minimum rotation angle between strike-slip and normal events being close to 90°. The combined strain tensor has a prolate form, dominated by the T axis, with an fclvd value of -0.23 and a k of 2.3. The orientation of the T axis is N052°E, while the P axis has an orientation of N142°E.

The stress tensor obtained (Fig. 7 WAA) by inversion shows a strike-slip stress regime with a normal component, with σ1 at N153°E. This stress regime activates left lateral N-S and right lateral ESE-WNW strike-slip faults, as well as NW-SE normal faults. The inferred stress tensor is like that obtained by Tendero-Salmerón et al. (2020) from 5 mechanisms in the Tiscar and Guadiana Menor faults (shear with σ1 at N143°E).

**5.15 WAA Western Alcaraz Arch**

Within the tectonic context of NW-SE convergence between Africa (Nubia) and Eurasia (Iberia), as well as the westward emplacement of the Alboran Domain, accompanied by the rollback of the southern Iberian slab, the presence of NW-SE extensional basins within the orogen appears to be kinematically necessary. The easternmost part of the Betic Orogen is dominated by processes of thrust arc indentation with tectonic transport to the NW (Tendero-Salmerón 2022), while the westernmost arc formed by the Betic and Rif arcs surrounding the Strait of Gibraltar is located to the W. This kinematics is

supported by the presence of a NW-SE extensional basin within the orogen, which is dominated by the presence of NW-SE normal faults. This kinematics is confirmed by GPS data (Cannavò and Palano 2016; Neres et al., 2018). The boundary between these two zones within the Betic Orogen is marked by extensional basins aligned and bounded by NW-SE normal faults, such as the Guadix fault in WAA and also by the Granada Basin. The age of the deformation of the Granada basin is Upper Miocene-present, defining a seismicity corridor about 300 km wide (Galindo-Zaldivar et al., 1999). The most

prominent faults here are the Granada, Sierra Elvira-Dílar and Padul-Dúrcal faults (Sanz de Galdeano et al., 2012).
      The focal mechanisms in the Granada Basin are extensional and cluster in two families whose T axis orientations form an angle of 30-40° between each other. The combined moment tensor is of oblate type, dominated by vertical shortening and with fclvd values of 0.08 and k of 0.768. The T axis has an orientation N048°E, and the B axis of N139°E; is consistent with the dominant shortenings in the surrounding zones.

The seismic sequence that occurred in 2021 allowed the determination of the stress tensor from 5 focal mechanisms of normal faulting stress regime (R=0.28), with σ3 at N049°E (Madarieta-Txurruka et al., 2022). In our stress inversion (Fig. 7 GB), we used 16 focal mechanisms that provide a very consistent triaxial normal faulting stress regime with σ3 striking at N033°E. Therefore, this stress orientation is congruent with that obtained to the E (WAA and EB), facilitating the emplacement of WB to the W.



## 5.16 NA Northern Alboran

The offshore deformation of the N margin of the Alborán Sea has been explained as a result of the ongoing slab rollback in the Alborán Domain, in the Gibraltar Arc (Betics-Rif), and of the indentation tectonics that predominates to the east and south, giving rise to a complex faulting pattern (Galindo-Zaldivar et al., 2022). On the other hand, in the easternmost part of this zone, the most important tectonic structure is the Carboneras Fault, a continuation of the Palomares Fault (EB). This structure has an NE-SW strike, extends offshore and has been active since the Late Miocene until the present. This fault shows a left-lateral strike-slip movement that occurs at a rate of 1.3 mm/yr (Moreno et al., 2015) and an offset of more than 15 km (Gràcia et al., 2006; Rutter et al., 2012), in which 4 palaeoseismic events have been identified (Masana et al., 2018). Among the 42 focal mechanisms analysed, we find reverse and normal solutions, but mainly strike-slip ones. Except for 3 normal faulting focal mechanisms indicating N-S extension, the others show a T (or B) axis towards NE-SW. The shortening axes in the focal mechanisms are very consistent, with orientations between N130°E - and N180°E. The minimum rotation angles between the normal faulting mechanisms and the strike-slip and thrust mechanisms are high, at 90-100°; while the minimum rotation angle between the reverse and strike-slip mechanisms is about 50°, indicating the strong strike-slip component present in the thrust earthquakes (see Appendix). The combined moment tensor is a slightly prolate strike-slip deformation tensor, with a k value of 1.63 and an fclvd of -0.14. The extensional stress axis, which would be dominant in the deformation of the zone, has an N065°E trend, while the shortening axis has an N154°E trend. The stress inversion result is alike those of WAA and EB. The obtained solution (Fig. 7 NA) does not seem to reflect the structural complexity indicated by the field and GPS data (Galindo-Zaldivar et al., 2022). The solution shows a strike-slip stress regime with a normal component, with σ1 in N149°E and σ3 in N064°E. Therefore, in this zone, the indentation process would predominate over the rollback process.







**Figure 7: Results of the stress-strain analyses for different zones: a) Right Dihedra solution. b) Rose diagram of the Deys obtained from the Slip model. C) Pitch/Dip plot for the neo-formed nodal planes obtained from the Slip Model. d) Stress Inversion Results. e) Variability of the three main stress axes of the stress inversion. OA Offshore Atlantic; GH Gorringe-Horseshoe; SVT Southern Valencia Trough; AL Algarve; EB Eastern Betics; WAA Western Alcaraz Arch; GB Granada Basin; NA Northern Alboran.**

## 5.17 AR Alboran Ridge

The most prominent structures in the Alborán Sea, in the central part of the Alborán Domain, are the Alborán Ridge, a crustal pop-up with the main tectonic transport to the NW and outcropping Neogene volcanic materials, the left-lateral Al Idrissi Fault and the right-lateral Yusuf strike-slip Fault. The three structures draw an indentor with similar kinematics to those of Águilas and Cazorla (in EB and WAA) (e.g., Tendero-Salmerón, 2022). Since the Late Miocene, magmatic intrusions in the Alborán Ridge seem to have acted as a backstop that favoured its uplift relative to the indentation (Tendero-Salmerón, 2022). The continental crust to the S of the aforementioned structures appears to belong to the African plate, so these faults are considered to constitute the active plate boundary between Nubia and Eurasia (Iberia), with the Yusuf Fault extending to the Tell Mountains orogen in Algeria (Martínez-García. 2012; Gómez de la Peña, 2017). The seismic crisis of 2016, which included an Mw 6.4 earthquake located in the southern limit of the Al Idrissi fault system, enabled defining its trace by connecting it with the Bokkoya and Trougout Faults, which enters the African onshore and connects with the Nekor Fault. It is therefore a very recent plate boundary (Gràcia et al., 2019). The seismic sequence indicates the presence of restraining steps along the strike-slip trace (Stich et al., 2020).

The focal mechanisms in this tectonic zone include pure reverse faulting, oblique faulting, with mainly normal component, and normal faulting with strike-slip component. The minimum rotation angles between the three fault types are high, showing the activation of different structures in response to a consistent strain tensor. The orientation of the T axes is very congruent in all mechanisms, with directions between N050°E and N080°E. The shortening axes show somewhat more variability. The combined moment tensor has an oblate shape, with k 0.519 and fclvd 0.19, and is dominated by shortening with an N157°E orientation. The mean T strain axis is N069°E.

The inversion of the 63 focal mechanisms (Fig. 8 AR) indicates a transpressive strike-slip faulting stress regime, with σ1 at N161°E, which activates mostly N-S to NNE-SSW left-lateral faults, NW-SW right-lateral strike-slip faults and E-W to ENE-WSW thrust faults. There are also 5 normal faulting mechanisms with nodal planes sub-parallel to the inferred σ1 orientation, indicating the presence of secondary normal faulting steps along faults in the principal direction. The stress solution is intermediate to that of the Águilas (EB) and Alcaraz (WAA) indentors.

## 5.18 ALH Al Hoceima

The S sector of the Alboran Domain, on the N coast of Morocco, shows significant seismic activity (2004 Mw 6.4 and 1994 Mw 5.9 earthquakes) in the Al Hoceima area with ruptures up to 20 km, which allowed us to infer an associated strike-slip stress regime (R=0.5) with σ1 at N161°E and σ3 at N071°E (van der Woerd et al., 2014). However, the Trougout (N171°E)




and Bokkoya (N030°E) faults, which continue offshore in AR, together with the Bousekkour-Aghbal (N020°E) fault, bound the Plio-Quaternary Nekor basin and define a transtensional area between the Nekor and Al-Idrissi faults (d'Acremont, 2014). Backstripping analyses of the sediments of Al Houcena Bay during the last 280 kyr indicate a westward migration of deformation with vertical throw rates of 0.47 mm/year because of the interaction between the northwesternwards movement of the Alboran indentor and the south-westward displacement of the Rif (Tendero-Salmerón, 2021). Further to the E, the NE-SW Kert and Nador faults appear to have a normal component (Ammar et al., 2007), although the considered focal mechanisms are mainly strike-slip faults like those of Al Hoceima. Therefore, we grouped them into a single population.

The focal mechanisms indicate mainly strike-slip and normal faulting, with a significant population of oblique faults with both components. Reverse-type events with strike-slip components are also present. All focal mechanisms have highly consistent axes of maximum shortening and horizontal extension, with little variability (see supplementary). The combined moment tensor indicates a directional deformation with an extension component close to a double couple, with an fclvd value of -0.08 and a k value of 1.3. The P axis of the combined moment tensor has an orientation N143°E and the T axis N060°E.

As there are more normal fault-type focal mechanisms in this population than in AR, the stress tensor shows a strike-slip faulting solution (Fig. 8 ALH) with a normal component, with σ1 at N152°E and σ3 at N064°E. Therefore, the tectonics in this zone is transtensional.

### 5.19 WTA-ETA Western-Eastern Tell Atlas

The Algerian Tell Atlas is the most seismically active area in the western Mediterranean, including among others the 1980 El Asnam, Mw 7.3, earthquake, which occurred on a 36 km long thrust linked to a NE-SW fault propagation anticline. The focal mechanism of this earthquake indicated the presence of a NW-dipping thrust plane (Meghraoui et al., 1986). In general, this is in good agreement with the tectonics of the range: dipping faults related to fault adaptation-propagation anticlines. These main neotectonic structures correspond to E-W striking to NE-SW striking thrusts that cut Quaternary rocks. The main intramountain basins are the Cheliff, Mitidja, Soummam, Hodna and Constantine basins (Maouche et al., 2019 and references therein). The coast shows evidence of folding and uplift, with marine terraces uplifted during the Pleistocene and Holocene (Maouche et al., 2011).

In the WTA, the most frequent focal mechanisms are reverse, coexisting with strike-slip and some normal faulting, all of them kinematically compatible with NW-SE horizontal shortening axes. The combined moment tensor is oblate inverse, with k 0.328 and fclvd 0.28, showing the predominance of horizontal shortening. The P axis has an orientation N143°E and the B axis N050°E. Further east, the population of focal mechanisms is similar, mainly with thrusting events. The combined moment tensor is therefore of shortening type and oblate, although closer to the double couple, with k 0.767 and fclvd 0.08. The P axis is oriented N155°E and the B axis N069°E; therefore, the shortening is more northerly in this area than it is further to the west.



Because the inferred moment tensor vectors and shortening directions are not coaxial, it has recently been suggested that, from the Alboran domain to the E, transpressional tectonics predominates, activating E-W striking right-lateral strike-slip faults and NE-SW striking  thrusts (Meghraoui and Pondrelli, 2012). Stress inversions based on earthquake focal mechanisms indicate that the deformation is accommodated by E-W striking reverse-strike-slip faults in the Eastern Tell, whereas the Western Tell is dominated by strike-slip faults (Soumaya et al., 2018). The stress inversions obtained in this

study (Fig. 8 WTA, ETA) show a very similar σ1 orientation for the eastern and western of the Tell-Atlas: N149°E (east) and N145°E (east), with a thrust-faulting stress regime to the east, and with a larger strike-slip component in the west, contrary to Soumaya et al. (2018). Both solutions activate NE-SW striking thrust faults, NW-SE right-lateral and N-S left-lateral strike-slip faults.

## 5.20 GC Gulf of Cadiz

The Gulf of Cádiz appears to be dominated by the southwestward movement of the Betic-Rif orogen, which has built up a sediment stack that is up to 12 km thick in an accretionary prism characterised by W-vergent thrust-spreading anticlines. The prism is related to the subduction of the S margin of Iberia below the Rif-Betic-Alboran microplate. Subduction seems to have slowed down significantly during the last 5 Myr, although deformation in the accretionary prism still affects recent sediments (Gutscher et al., 2012). Thermo-mechanical modelling indicates that although the subduction process has ceased,

deep slab motion still induces a mantle flow that produces a W-directed basal drag of the Alboran domain lithosphere (Gea et al., 2023). Some of the selected focal mechanisms are located in the S margin of the Algarve Basin, S of the Portimao Bank (AL) (Ramos et al., 2015; Neres et al, 2018), providing a fairly homogeneous population.

The focal mechanisms in this tectonic zone are mainly strike-slip, although normal and reverse faulting events are also present. The combination of the mechanisms results in a directional seismic moment tensor but with an oblate shape, with

values of k 0.417 and fclvd 0.24. The P-axis has an orientation N161°E and the T-axis N066°E. The inversion provides a strike-slip stress regime (Fig. 8 GC) with σ1 at N150°E, with less thrusting component than that obtained for AL, GH and WCS. Therefore, it does not seem that there is significant seismicity related to thrusting with tectonic transport to the W, but rather to the SE or NW.

## 5.21 WB Western Betics

The emplacement of the Rif-Betics-Alboran Domain to the W during the Early Middle Miocene, together with the NW-SE oblique convergence between Eurasia and Africa, has conditioned the structuring of the Betic Orogen. However, since the Upper Miocene, it is the latter process that seems to have dominated (Ruiz-Constán et al., 2011). The NW Betic Mountain front is the most seismically active sector, and although seismogenic structures do not outcrop at the surface moderate-depth earthquakes indicate the presence of NE-SW thrusts with some related tear faults (Ruiz-Constan 2009).



In our analysis we used only earthquakes at crustal depths. The population of focal mechanisms is dominated by strike-slip faulting, several of which are oblique with a reverse component, but a significant number of thrust faulting events are also observed. The orientations of the axes of maximum shortening and horizontal extension are scattered, characterised by two families, a major one with a N120°E-N150°E shortening direction and a minor one with a N-S shortening. The combination of the focal mechanisms gives rise to a shortening-directional seismic moment tensor with an oblate shape, showing values

of k 0.376 and fclvd 0.26. The shortening axis P has an orientation of N143°E and the B axis N046°E, although with a 28° plunge.

The total population (32 mechanisms) provides a well-constrained stress inversion result (Fig. 8 WB). The stress tensor inferred indicates a strike-slip stress regime with $\sigma 1$ at N139°E, which activates NE-SW striking thrusts and strike-slip faults, which experience right lateral offset when striking ENE-WSW and left lateral at NNE-SSW.

Stress inversions in this area considered the hypocentral depth, as the deepest stresses/deformations could be related to the Iberian slab under the Alboran Domain. Ruiz-Constán et al. (2012) obtained N166°E (4 mechanisms) and N018°E (4 mechanisms) $\sigma 1$ trends for the shallow seismicity, while for the intermediate earthquakes, they obtain a $\sigma 1$ between N113°E-N126°E (29 mechanisms). The $\sigma 1$ trend is located more towards the ESE-WSW concerning the surrounding areas, probably influenced by the remnant effect of the slab (Gea et al., 2023), more evident at depth (Ruiz-Constán et al., 2012). To test this

effect, we inverted the 7 focal mechanisms corresponding to earthquakes with hypocentral depths of more than 20 km in this zone (WBD), obtaining a $\sigma 1$ of N114°E, more E-W than the shallow ones (Fig. 8 WB>20). Therefore, our results confirm that the slab effect is more pronounced at greater depths, being negligible in shallow earthquakes.

### 5.22 BA Betics Antequera

Between the Granada Basin (GB) and the thrusts at the NW edge of the Betics (WB), there is a 70 km long right-lateral

transpressional brittle-ductile shear zone, The Torcal Shear Zone, that has been active from the Late Miocene until the Quaternary (Barcos et al., 2015). In 1989, a seismic series (117 earthquakes) was reported between Loja and Palenciana, which indicated that the fault zone had a strike of N070°E-N080°E (Posadas et al., 1993). From the focal mechanisms obtained for this crisis, a strike-slip stress regime with $\sigma 1$ at N135E was previously determined by Vadillo (1999).

The focal mechanisms in this tectonic zone are of strike-slip type with an extensional component (5 events). The

combination of these events results in a strain tensor very close to a double couple, with a value of k 1.16 and fclvd -0.04. The orientation of the P axis is N125°E and the T axis N028°E. The inversion of the 5 focal mechanisms (Fig. 8 BA) shows a strike-slip stress regime with a normal component, with $\sigma 1$ at N105°E, which activates right-lateral strike-slip-normal (transtensional) faults. This $\sigma 1$ orientation is more similar to that of the WB than to those of the other adjacent areas and could indicate a greater effect of the rollback process of the Iberian slab, from here to the W. The extension-normal faulting

in the Granada Basin (GB) may therefore be explained by the greater effect of the westernward remnant movement of the Alboran Domain, which may be partially decoupled from the indentation zone of the Betic arcs further to the east.









**Figure 8: Results of the stress-strain analyses for different zones: a) Right Dihedra solution. b) Rose diagram of the Deys obtained from the Slip model. C) Pitch/Dip plot for the neo-formed nodal planes obtained from the Slip Model. d) Stress Inversion Results.**
**e) Variability of the three principal stress axes of the stress inversion. AR Alboran Ridge; ALH Al Hoceima; WTA Western Tell Atlas; ETA Eastern Tell Atlas; GC Gulf of Cadiz; WB Western Betics (WB depth > 20 km); BA Betics Antequera.**

## 6 Stress map of Greater Iberia

The new compilation of earthquake focal mechanisms and the results from the stress inver-sions can be used to update the stress map of Iberia using the quality ranking scheme of the World Stress Map (WSM) project. The WSM is the global
resource for stress information on the present-day stress field of the Earth's crust (Heidbach, 2016; Heidbach et al., 2018; Zoback, 1992) and compiles the orientation of maximum horizontal stress $\sigma_{Hmax}$ from a wide range of stress indicators such as earthquake focal mechanism solutions (FMS), drilling-induced tensile fractures (DIF), borehole breakouts (BO), hydraulic fracturing tests (HF), overcoring (OC) as well as geologic data from seismogenic fault-slip analysis (GFI, GFS) and volcanic vent alignments (GVA) (Amadei and Stephansson, 1997; Ljunggren et al., 2003; Sperner et al., 2003). The stress
information is compiled in a standardized data format and quality-ranked to make data from very different methods comparable (Heidbach et al., 2010).

The various stress indicators reflect the in situ stress of very different rock volumes ranging from $10^{-3}$ to $10^9$ m$^3$. Furthermore, except for the earthquake focal mechanisms and a few very deep boreholes, all stress indicators sample only the stress patterns within the upper 6 km of the Earth's crust, with deep boreholes as a major contributor. The most common
visualization of stress data is through stress maps where data from depths between 0 and 40 km is integrated (Heidbach et al., 2004; Heidbach and Höhne, 2008) assuming that the $\sigma_{Hmax}$ orientation does not change significantly with depth. This assumption was tested qualitatively at the beginning of the WSM project (Zoback, 1992) and confirmed later with significantly higher data density on global (Heidbach et al., 2018) and regional scales (Pierdominici and Heidbach, 2012; Rajabi et al., 2017a).

For the new stress map of Greater Iberia, we re-evaluate all data records from geological data (n=141), borehole breakouts (n=129), overcoring (n=16), and hydraulic fracturing (n=5) and combine these with the σHmax orientations derived from the new compilation of earth-quake focal mechanisms (FMS) and stress inversion results (FMF) obtained in this study. Given that the majority of the WSM data records have not been revisited for almost 30 years, our re-evaluation resulted in a reduction of data records due to double entries and typos (from n=295 to n=271) and down-ranking in quality due to a
stricter data assessment. There-fore, the number of stress data records with A-C quality decreased from n=172 to n=132. A-C quality means that the $\sigma_{Hmax}$ orientation is reliable within ± 25°, D quality data records are only reliable within ± 40° and thus should be used with caution (Rajabi et al., 2024; Tingay et al., 2006; Tingay et al., 2005). E-quality data are poor in quality and X-quality data have not sufficient or missing information to assign a quality. The latter is a new assignment class that is used already in the new WSM quality ranking for stress magnitude data records (Morawietz et al., 2020) and will be also
used in the next WSM database release for stress orientation data records (Table 6).





| Type/Quality | A | B | C | D | E | X | Total |
|---|---|---|---|---|---|---|---|
| **FMS** | 0 | 0 | 456 | 0 | 86 | 0 | **542** |
| **FMF** | 5 | 15 | 0 | 0 | 11 | 18 | **49** |
| **FMA** | 0 | 0 | 0 | 24 | 0 | 1 | **25** |
| **BO** | 8 | 30 | 36 | 38 | 5 | 5 | **122** |
| **HF** | 1 | 0 | 1 | 1 | 0 | 0 | **3** |
| **OC** | 0 | 0 | 0 | 2 | 0 | 14 | **16** |
| **GFS** | 0 | 0 | 0 | 12 | 0 | 0 | **12** |
| **GFI** | 0 | 8 | 48 | 10 | 22 | 30 | **118** |
| **Total** | **14** | **53** | **541** | **53** | **148** | **68** | **887** |

**Table 6: Overview of data quality and stress indicators in the new compilation of stress information shown in Fig. 10. Note that most of the downranking of data records from borehole breakouts (BO), hydraulic fracturing (HF), overcoring (OC) and geological indicator (GFI, GFS) are due to missing information in the papers and reports where**
**data are presented. The other abbreviations are FMS for single focal mechanisms, FMF for stress inversion from population of focal mechanisms, and FMA for composite focal mechanisms.**

The resulting stress map shows that, at first order, the data records from boreholes agree with earthquake focal mechanisms data from greater depths. An exception is the borehole data in the Aquitaine Basins north of the Pyrenees where some $\sigma_{Hmax}$ orientations from borehole breakouts confirm the prevailing WNW-ESE strike, but others show different orientations. These
data result from a comprehensive study of 55 wells by Bell et al. (1992) and the authors discuss in great detail this somewhat controversial result of varying $\sigma_{Hmax}$ orientations on local scales which has not been observed in other foreland basins (Reinecker et al., 2010; Reiter et al., 2014). However, for the remaining areas, there is an overall agreement between the $\sigma_{Hmax}$ orientations inferred from borehole data in comparison with earthquake focal mechanisms results.

To analyze the prevailing $\sigma_{Hmax}$ orientation pattern we estimate the mean $\sigma_{Hmax}$ orientation on a 0.5° grid using the tool
stress2grid from Ziegler and Heidbach (2019) with a 150 km search radius. For the estimation, a minimum of five data records is required within the search radius. Weights are applied considering data quality and distance to the grid point. The distance weight is cut off when the data record is within 15 km of the grid point to avoid an overrepresentation of data records close to the grid point. Furthermore, we distinguish the resulting mean $\sigma_{Hmax}$ orientations according to their standard deviation SD. Dark grey bars in Fig. 9 denote mean $\sigma_{Hmax}$ orientations with SD ≤ 25°, light grey ones with SD > 25°. The
resulting mean $\sigma_{Hmax}$ orientation in Fig. 9 shows that in particular in the center of Iberia, the stress pattern does not show a clear trend, in contrast to mostly all the other regions, with the exception of the Pyrenees. This is expressed in rotations of the mean $\sigma_{Hmax}$ orientation and the significantly higher SD values in center of Spain. The changes of the mean $\sigma_{Hmax}$ orientation on short scales could be either due to low data density, which allows for a single outlier or local deviation from the stress pattern has a high impact on the mean trend, or to the stress pattern being indeed quite variable, for example due to
low anisotropy of the horizontal stresses which, results in less stable horizontal stress orientations (Heidbach et al., 2007; Lundstern and Zoback, 2020; Rajabi et al., 2017b).



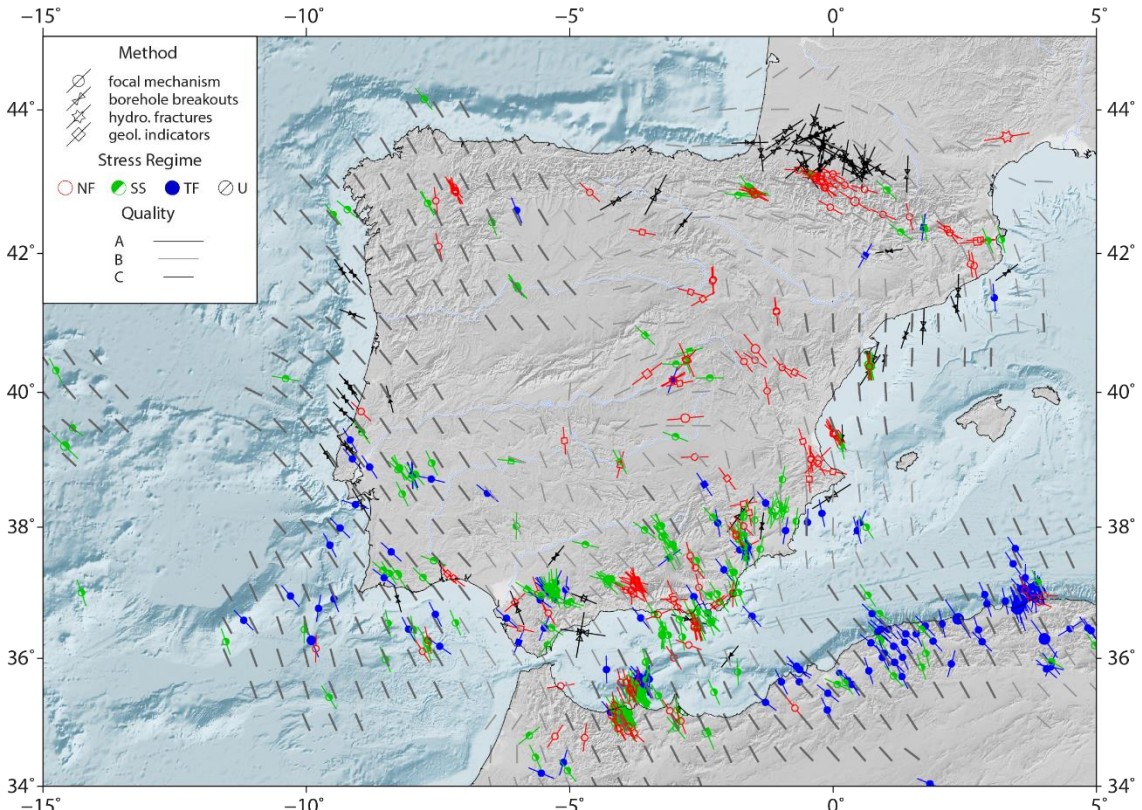

**Figure 9: Stress map of Greater Iberia based on A-C quality data records from this study and from re-evaluated data**
of the WSM database release 2016 (*Heidbach et al.*, 2016). Plotted is the orientation of maximum horizontal stress
$\sigma_{Hmax}$ for depths between 0-40 km. See the inset legend for details on data types, stress regime (NF= normal faulting,
SS= strike-slip, TF= thrust faulting, U= unknown). Data symbols indicate the type of stress indicator and line length
are proportional to data quality. Grey bars on a 0.5° grid show the mean $\sigma_{Hmax}$ orientation that is estimated with a
search radius of 150 km using weights for data quality and distance to the grid point. Dark grey bars on the grid
show mean $\sigma_{Hmax}$ orientations with a standard deviation ≤ 25°, light grey bars with standard deviation > 25°.
Topography and bathymetry is taken from SRTM15+ (Tozer et al., 2019)

## 7 Discussion

Our analysis focuses on the Iberian Peninsula, not including the Gloria Fault between the Terceira Ridge in the Azores
Islands and the Gorringe Bank, where the tectonic stresses that generate seismicity are related to strike-slip faults (e.g., de
Vicente et al., 2008).

As can be seen in Fig. 10a, the combined tensors have distinct characteristics depending on their position in the tectonic
context. Thus, in the plate boundary zone, the tensors present typical characteristics of reverse or reverse directional faults
with positive fclvd values, while the tensors of the Pyrenees and the central-eastern Iberian Peninsula present normal or




normal-directional focal mechanisms with negative fclvd values. An exception to this rule is the combined tensor of the Granada Basin, with a very pure normal faulting tensor.

When we consider the combined moment tensor by rupture type, the reverse component (Fig. 10b) shows the predominant NW-SE shortening orientation in the area, with nodal planes of approximately N040°E to N070°E strike. This broadly northeast to east-northeast orientation of the moment tensor nodal planes strike is found in all combined plate boundary tensors from northern Algeria to the Gulf of Cadiz. Only the combined tensor at the eastern tip of the Betics (EB) has an orientation of its nodal planes that is more E-W, corresponding to an N-S shortening. These fault orientations are found in the northern sector of the Arco de Águilas and the Bajo Segura Fault system. In the north of the Iberian Peninsula, the seismic moment released in the form of reverse fault earthquakes is much smaller. They present oblique tensors whose P-axes are consistent with the regional shortening axes marked more clearly on the combined strike-slip tensors in these cases.

The strike-slip combined tensors (Fig. 10c) present nodal planes of N000°E to N050°E for the left-lateral kinematics, compatible with the NW-SE shortening orientation described above. All the strike-slip tensors present these characteristics except for the Western Pyrenees (WP) and Central Basins (CB) tensors, which present right-lateral kinematics for NE-SW planes. These two cases are also characterised by E-W shortening axis orientations and prolate tensor shapes (K values > 1, fclvd < 0), having the highest K values together with the tensor from the northwest of the Iberian Peninsula (NWG). Although the strike-slip tensors are dominant throughout the peninsula (Fig. 10e), they are particularly relevant in the Trans-Alboran system.

Regarding the combined normal faulting tensors, we can find two groups. On the one hand, tensors whose extension axes have an orientation from E-W to NE-SW and on the other hand, those with an extension axis orientation N-S. The latter are found in the central and western Pyrenees (CP, WP), as well as in the central basins (CB). In the rest of the peninsula, the normal tensors present nodal planes whose orientation is approximately parallel to the axes of maximum horizontal shortening. The normal tensors are more important in a band that joins the Al Hoceima area in North Africa with the central Pyrenees passing through the Iberian chain (Figs. 10d and 11b). Also, in the northwest of the Iberian Peninsula (NWG), normal faulting mechanisms have a predominant role in the combined tensor (Fig. 12b).

If we project the combined tensors in classification diagram (Fig. 10e), we see how most of them are tensors clearly located in the fields of pure rupture types, as well as the predominance of strike-slip tensors, in more or less important combination with compression or extension. If in the diagram we represent the combined tensors by rupture type for each zone (Fig. 10f) we see how the pure tensors are still dominant, which indicates that the combination between different rupture types takes place by permutation of the principal axes, being compatible with each other and therefore defining a distributed deformation.

The orientations of the principal axes for each zone are shown in Fig. 11a. NW-SE shortening is predominant in North Africa, the Alboran Sea and the western half of the Iberian Peninsula. In the eastern part, the shortenings tend to be N-S, except in the areas where the extension is predominant (Fig. 11b), i.e. in the Pyrenees and the central basins, where the extension is N-S and therefore the maximum horizontal shortening is E-W.





In every zone for which we have composed focal mechanisms for the various faulting types, we computed the minimum rotation angle between pairs of tensors (Fig. 11 c). Most of the rotation angles are between 60° and 110°. These values are
indicative of the activation of structures sharing a common strain tensor orientation, where permutations between the axes (rotations of 90o) take place, responding to a coherent tectonic framework. An example, of such process is the permutation between the shortening axis of reverse faults and the intermediate axis of normal faults. The results for the WVT zone deserve a special mention. They consist of the focal mechanisms related to the Castor Project seismic crisis, a natural gas submarine storage project (Villaseñor et al., 2020; Cesca et al., 2021). As can be seen, the rotation angle between the tensors
of the different rupture types is very small, showing that the rupture type of all the events is similar and consists of oblique focal mechanisms located near the centre of the classification diagram (Fig. 10 f).

In Fig. 11 d, the combined tensors are represented in the Flinn-type diagram for moment tensors with the colour showing the k-value. The zones with prolate ellipsoids (K>1), located above the plane strain diagonal of the Flinn-type diagram, are those also characterised by a significant normal rupture component, these are the Pyrenean zones (EPCE, WP, NWG), the
Valencia Trough (SVT) and the Central Basins (CB). These results evidence the relationship of these prolate ellipsoids with extensional and transtensional tectonic settings. On the other hand, the areas with oblate ellipsoids are those related to transpressional tectonic settings, mainly in the southwestern margin of the Iberian Peninsula (AL, GH, WB, GC, WCS). This rela-tionship between of the strain tensor shape and the position with respect to the plate bounda-ry is highlighted when we plot the k-value as a function of the latitude (Fig. 12). Near the plate boundary the shapes of the strain tensors are mainly
oblate, with k < 1; and as we move away from the plate boundary the k values increase, with mostly prolate strain tensors to the north of the Betics (k > 1).



**Figure 10: a) Combined strain focal mechanisms for each of the tectonic zones. The colour of the beach ball represents the compensated linear vector dipole factor fclvd. b) Combined focal mechanism for reverse faulting earthquakes in each zone. c) Combined focal mechanism for strike-slip earthquakes in each zone. d) Combined focal mechanism for normal earthquakes in**




each zone. e) Focal mechanism classification diagram for the combined mechanisms shown in a). f) Focal mechanism classification diagram for the different rupture type combined mechanisms shown in b), c) and d).

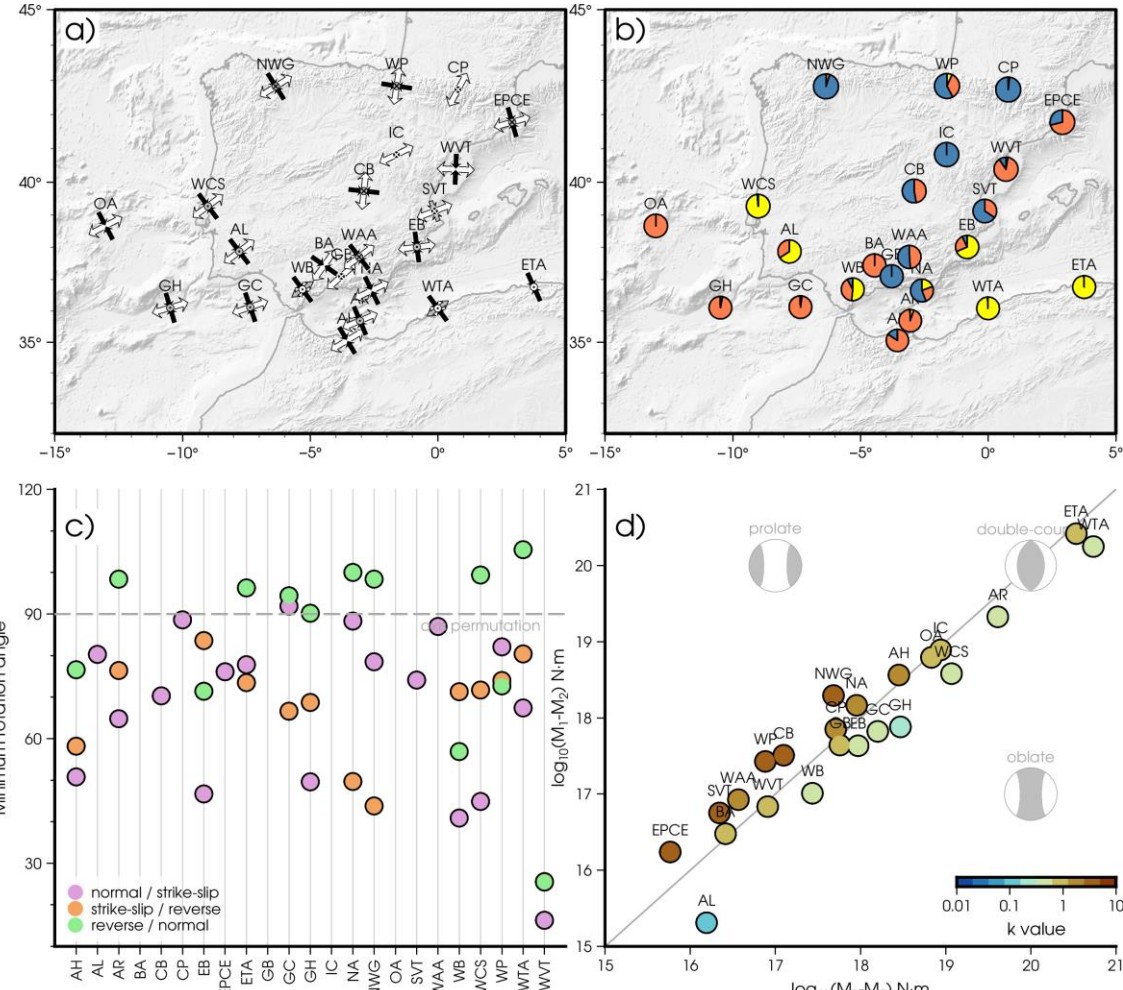

**Figure 11: a)** Strain principal directions derived from the combined moment tensors. Black shows the orientation of the shortening
**(P)** axis; in white is shown the stretching axis (T) and grey the intermediate axis. The orientations are simplified showing only the trend of the principal axes and omitting the plunge. Note that the intermediate axis (in grey) can be a shortening or a stretching axis; when it is the vertical axis it is represented by a circle with a cross or with a dot respectively. **b)** Pie diagram showing the proportion of seismic moment released by strike-slip (red), reverse (yellow) and normal (blue) events on each zone. **c)** Representation of the minimum rotation angle (Kagan angle) between combined focal mechanisms for the different rupture types
in each zone. A 90º angle represents a pure axes permutation. **d)** Flinn-like diagram showing the combined seismic moment strain tensor shape. The k-value is defined as (M1-M2)/(M2-M3).





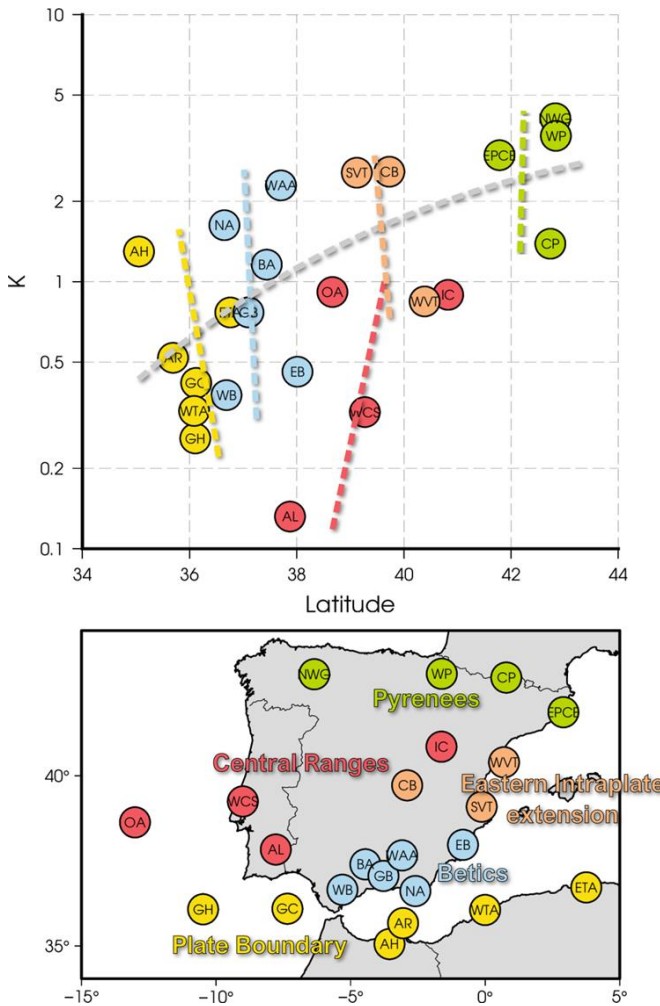

**Figure 12: Relationship between the k-value of the strain tensor and the latitude of the zone. The colours are related to the tectonic setting as shown in the map. The coloured dashed lines show the trend of each population, while the grey dashed line shows the general trend of K-value increment with latitude.**




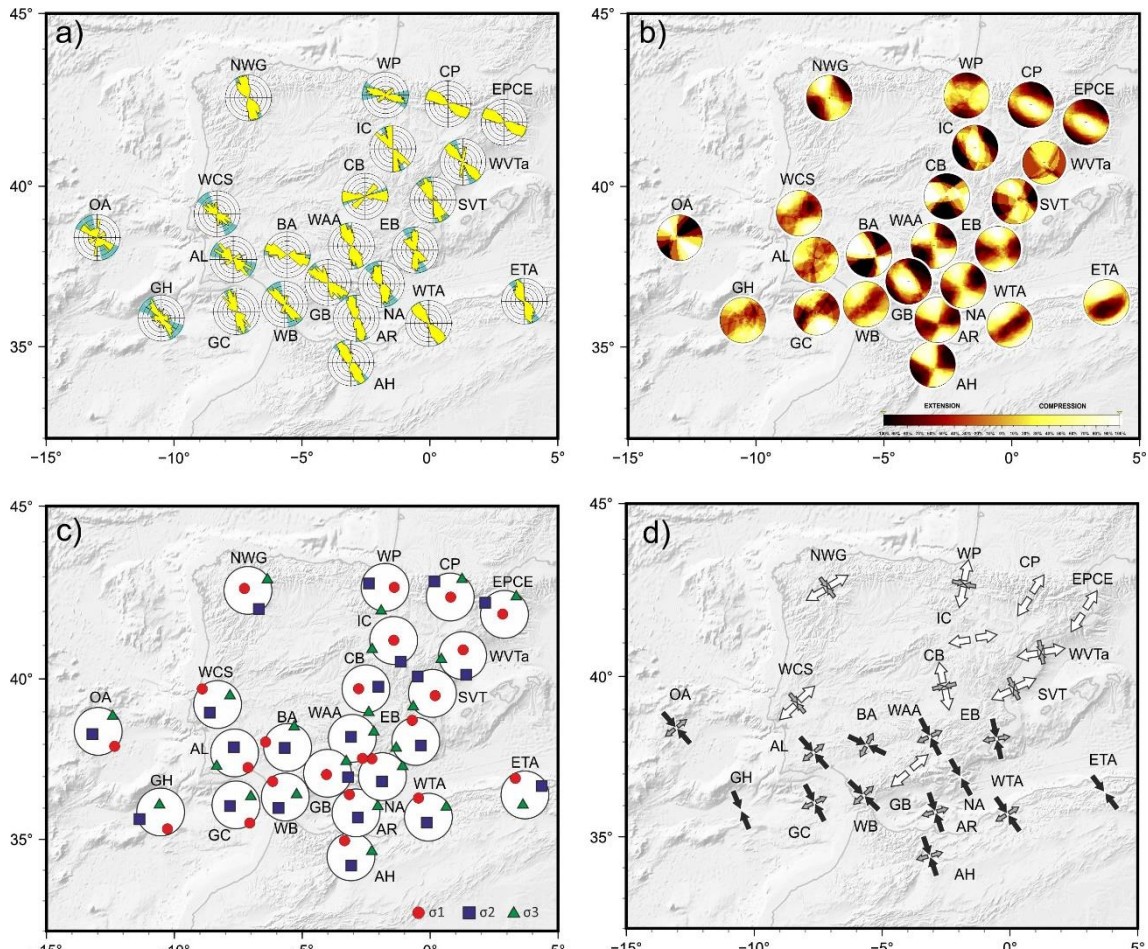

**Figure 13: a) Deduced $E_{Hmax}$ (Dey, maximum horizontal shortening / minimum horizontal extension) in every considered zone from the Slip Model. b) Combined Right Dihedra plots. c) Stress inversions from focal mechanism populations in all the zones. d)**
**scaled sizes of the stress horizontal axes, $\sigma_1$ (black), $\sigma_2$ (grey) and $\sigma_3$ (white).**

Results regarding the type of active deformation from the slip model analysis are shown in Fig. 3. The different distribution of data density leads to poorly defined interpolations in areas where data density is low. Nevertheless, the inferred $E_{Hmax}$ (Dey, maximum horizontal shortening / minimum horizontal extension) in every considered zone (Fig. 13 a) leads to a well-defined image of the progressive rotations of $E_{Hmax}$ in Iberia. Composed Right Dihedra plots qualitatively show similar

results (Fig. 13 b).

The stress inversion solutions (Figs. 13 c, 13 d and 14) indicate that the clearest thrust faulting stress regimes appear at the edges of the study area (Gorringe and Eastern Atlas) with a vertical $\sigma_3$ and an R around 0.4. In between, and throughout southern Iberia, strike-slip faulting stress regimes with a thrusting component (transpression) predominate (vertical $\sigma_2$ and $0.5< R > 0$), as well as in the SW (Western Spanish-Portuguese Central System and Algarve). The exception to this rule is

the Granada Basin, where almost radial normal faulting stress regime occurs. These stresses influence the closest





earthquakes to the E and W (Western Alcaraz and Antequera), where strike-slip faulting stress regimes with a normal component (transtension) (vertical $\sigma_2$ and $1< R > 0.5$) are active (Fig. 14). This type of stresses is also found in Al Hoceima and the offshore Atlantic. The remaining inversions (N and NE) yield normal faulting stress regimes (vertical $\sigma_1$ and R close to 0.5). The $\sigma1$ trend along the plate limit (yellow in Fig. 14) remains very consistently close to N154E. Some more
variability occurs in the Betics (blue in Fig. 14), although the mean value of $\sigma_1$ is also close to N155E, except in the Granada Basin, Antequera and the deep population of the Western Betics, which approaches N114°E, probably influenced by the remanent effect of the Alboran Slab. In the Central Ranges and offshore Atlantic (red in Fig. 14), the $\sigma_1$ trend rotates slightly anticlockwise to N140°E, in agreement with the Eulerian pole of plate motion between Africa and Eurasia. In the Pyrenees, the clusters E and W of the main topographical relief show $\sigma1$ orientations close to N154°E, as in most Iberian populations.
Nevertheless, the central part (CO and WP) has a $\sigma_3$ perpendicular to the range, so that local stresses (post-orogenic collapse or isostatic reset) predominate over the regional ones. The Central Basins is clearly out of this global arrangement. The solution in the Eastern Pyrenees is very similar to those of the Valencia Through and the Iberian Chain (EPCE, WVT, SVT and IC), defining a cluster of solutions for the E of the Iberian Peninsula (Fig. 14). It should be noted that, in these solutions, $\sigma_2$ is oriented in the NW-SE direction and not in NE-SW so that the present-day normal faulting stress regime is likely to be
more affected by the convergence between Africa and Eurasia than by the back-arc extension of the easternmost subduction zones.

The presence of thrusting focal mechanisms in front of the Alboran Domain supports the idea that its emplacement has been mechanically decoupling of Iberia from Nubia, from E to W. Similarly, the SPCS has been losing its thrust faulting stress regime from E to W, deactivating the thrusts N of Madrid since the Late Miocene, while in its Portuguese sector, it is still an
active intraplate orogen. Therefore, in central Iberia, from E to W, the state of stress changes from a clearly normal faulting stress regime in the IC (R=0.43 with $\sigma_1$ in the vertical), with normal faulting also in the middle (CB), to a thrust-faulting stress regime in the WCS, and a strike-slip faulting stress regime in the offshore (OA), progressively passing $\sigma_2$ or $\sigma_1$ from N148°E to N131°E (except in CB), according to the position of the Eulerian pole. Thus, in the offshore continuation of the SPCS, the expected tectonic environment would be transtensive.

Fig. 15 summarizes the stress inversion results over an Alpine tectonic map of Iberia. Where normal faulting stress regimes are inferred (blue arrows in Fig. 15), Cenozoic thrusts are not active (North, Central and Northeast Iberia), but normal faulting stress regimes are active in areas that display previous compressional features, as in the IC. Transtensional areas (green arrows in Fig. 15) are also probably deactivating thrusts in the Estremadura Spur (OC) and the closest zones to the Granada Basin (WAA and BA), where there is an active normal faulting stress regime. Mapped thrusts are predominant in
the Gorringe Bank and eastern Algeria (red arrows in Fig. 15). In the remaining areas, easternmost and westernmost Betics, and southwestern Iberia, transpression prevails. Therefore, mapped thrusts and strike-slip faults must be considered active structures.





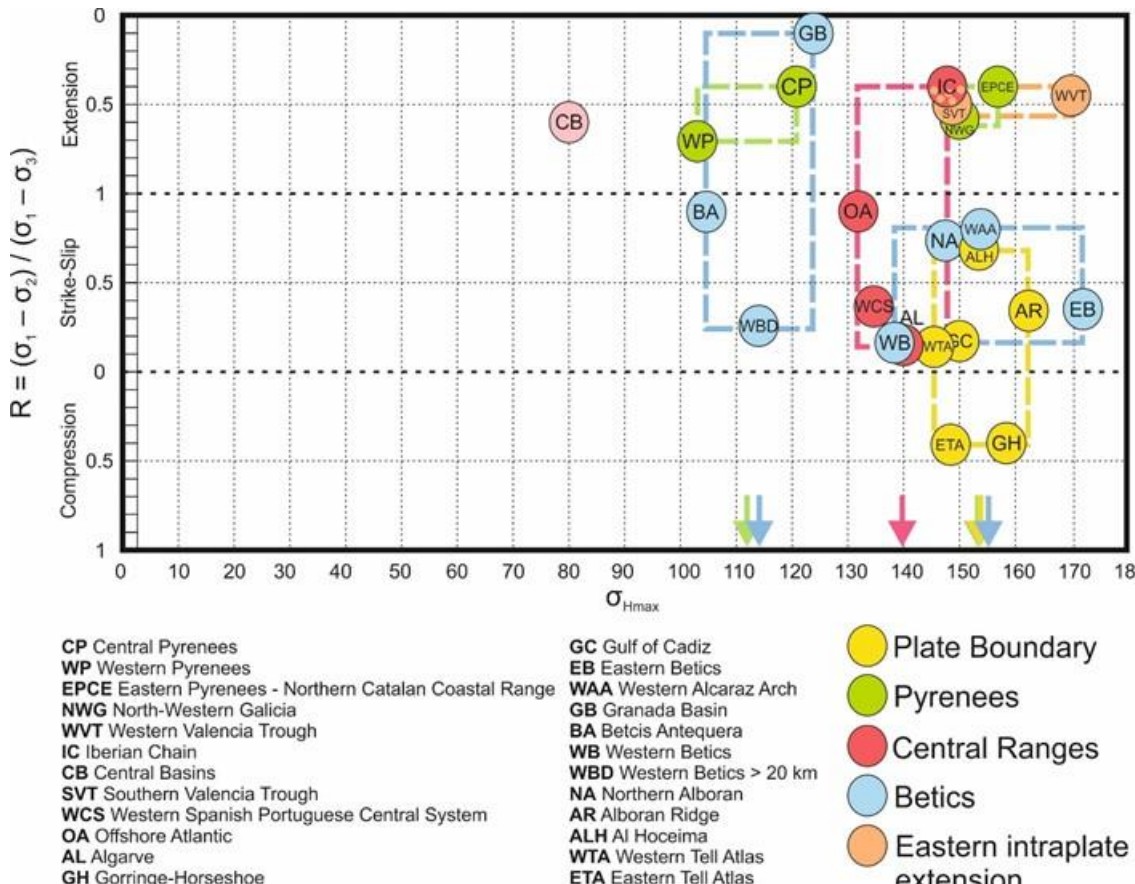


**Figure 14: R (Stress tensor ratio) Vs $\sigma_{Hmax}$ trend ($\sigma_1$ for thrusts and strike-slip faults stress regimes, $\sigma_2$ normal faulting stress regimes). The N-S direction is marked by 0° and 180°, while the E-W direction coincides with the value of 90. The areas considered have been grouped into those at the plate boundary (yellow, AR, GH, ETA, GC, WTA, and ALH), those in the Pyrenees (green, EPCE, NWG, CP, and WP), those in the Central Ranges (in red, OA, WCS, AL and IC), those in the Betics (in blue, separated BA,**
**GB, WBD from the strike-strike solutions BA, GB, WBD), those related to the intraplate extension in E Iberia (Mediterranean) (in orange, WVT, SVT and IC. The latter is also included in the Central Ranges solution). CB (in pink) does not seem to be related to any of the previous groupings. The dashed rectangles mark each group's maximum and minimum values of R and $\sigma_{Hma}$, with their corresponding colour. The lower arrows indicate the average values of $\sigma_{Hma}$ for each group.**



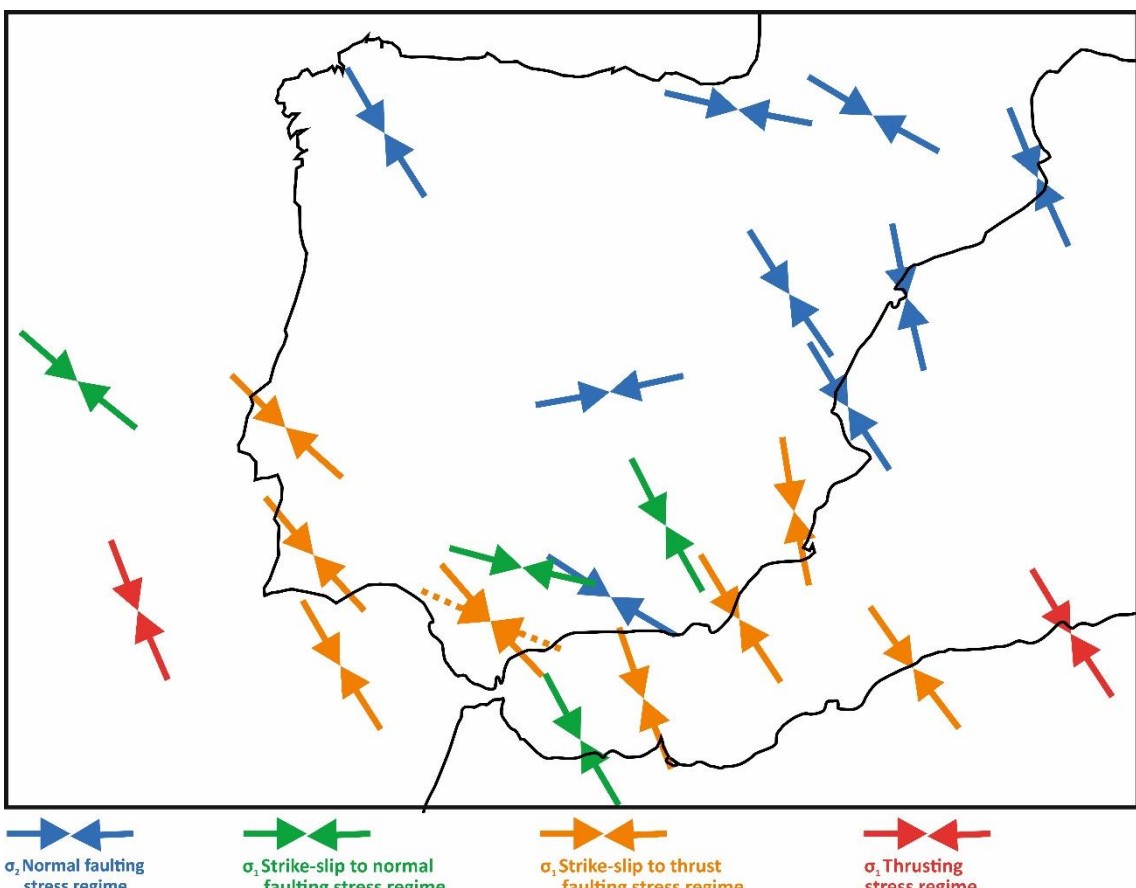

**Figure 15: Summarized state of stress of Iberia based on focal mechanism stress inversion from this study.**

## 7 Conclusions

In general, the shape of the combine seismic moment tensors at the plate boundary is oblate (k < 1), showing a predominance of transpressive tensors. As we move away from the plate boundary, the tensors become prolate in shape (k > 1), characterising the extensional contexts of the Pyrenees, the Valencia Trough, and the Central Basins. An exception to this 970 rule is the combine tensor of the Granada Basin, which yielded a pure normal faulting tensor close to the plate boundary. Similar tendencies are observed from the rescaled shape factor (k´). Northern Algeria and SW Portugal represent the maximum compressive areas, related to the plate boundary, while the Pyrenees and the Iberian Chain are extensional. Individual calculations obtained from the Slip Model allow to distinguish local variations. The proposed methodology optimizes the selection of populations selection for kinematic and dynamic analyses.

Although in most of the defined tectonic zones we find focal mechanisms of different faulting types, the minimum rotation angle between their combined mechanisms shows that they are compatible with each other considering the permutation of their axes, in a context of distributed deformation.





The obtained orientations of the shortening and extension axes in the combined deformation tensors are similiar the maximum and minimum principal stress axes obtained from the stress inversion. However, the results may differ
significantly for some populations.

Outside of the Iberian Peninsula, our stress inversions yielded thrust faulting stress regimes in the Gorringe and Eastern Atlas zones, with a vertical $\sigma_3$ and R around 0.4. In southern Iberia, transpression predominates (strike-slip faulting stress regimes with a thrusting component), with vertical $\sigma_2$ and 0.5> R > 0. This type of stresses is also found in the southwestern corner of the Iberian Peninsula (Western Spanish-Portuguese Central System and Algarve). Transtensional (strike-slip
faulting stress regimes with a normal faulting component, vertical $\sigma_2$ and 1> R > 0,5) surround the normal faulting stress regime inferred for the Granada Basin (Western Alcaraz and Antequera), and are also present in northern Morocco (Al Hoceima) and offshore Atlantic. Towards the Betics foreland and to the east of the westernmost sector of the Spanish-Portuguese Central System, including the Pyrenees, a normal faulting stress regime predominates. Within these zones, a slightly more strike-slip component is observed westwards of the Central Pyrenees (Western Pyrenees and North-Western
Galicia). Regardless, R values in these zones are close to 0.5, except in the Granada Basin inversion, where an almost radial normal faulting stress regime is found.

The $\sigma_{Hmax}$ mean values range from N105°E to N155°E (except for the Central Basins solution). The $\sigma_{Hmax}$ orientation from this study based on individual focal mechanisms and the stress inversion are in good overall agreement with the data records from other stress indicators, in particular from the numerous borehole logging data previously published as part of the World
Stress Map (Heidbach et al., 2018). Only in the center of Iberia this is not true, either due to the low data density resulting in large rotations of $\sigma_{Hmax}$ from the regional trend, or to horizontal stress magnitudes being close to each other, allowing for local stress variability due to stiffness and density contrasts.

The ESE-WNW closest stress inversion results that can be considered anomalous at a regional scale are related to the Granada Basin and the Pyrenees, where local stresses arise. Out of this, the Central Ranges (IC, WCS, OA and AL) have a
common $\sigma_{Hmax}$ trend around N140°E. Although the Betics solutions (far from the Granada Basin) have some variability, they fit an N155°E mean $\sigma_{Hmax}$ trend, like the ones found in the NW and NE corners of Iberia (EPCS and NWG). Solutions along the plate limit also concur with this $\sigma_{Hmax}$ trend; N155°E can then be considered the likely result of Africa approaching Iberia. The 15o anticlockwise rotation to the north must be related to the location of the Eulerian pole between both plates. This general tectonic context seems to be overprinting the back-arc subduction-related extension in the east of Iberia and the
Alboran Sea.




## Appendix A



**Fig. A1: Al Hoceima tectonic zone. Top map: focal mechanisms with maximum horizontal axis (P or B for normal ruptures) and minimum horizontal axis (T or B for reverse ruptures) orientation. Bottom left classification diagram for earthquake rupture types. Middle right: Average tensors (complete in gray) and Kagan angles between the average tensors for each rupture type. Bottom right: Rose diagram of maximum and minimum horizontal axes orientation.**





**Fig. A2: Algarve tectonic zone. Top map: focal mechanisms with maximum horizontal axis (P or B for normal ruptures) and minimum horizontal axis (T or B for reverse ruptures) orientation. Bottom left classification diagram for earthquake rupture types. Middle right: Average tensors (complete in gray) and Kagan angles between the average tensors for each rupture type. Bottom right: Rose diagram of maximum and minimum horizontal axes orientation.**




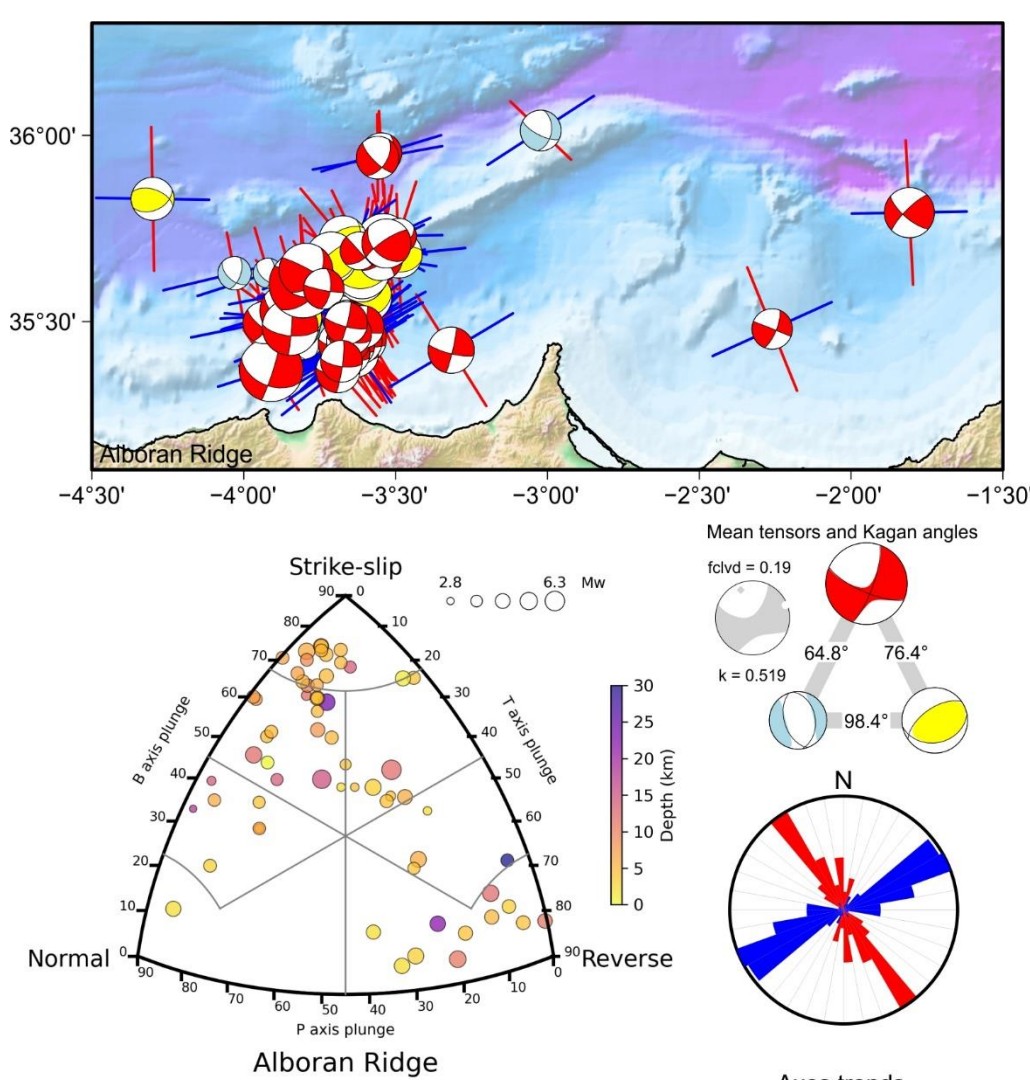

Fig. A3: Alboran Ridge tectonic zone. Top map: focal mechanisms with maximum horizontal axis (P or B for normal ruptures) and minimum horizontal axis (T or B for reverse ruptures) orientation. Bottom left classification diagram for earthquake rupture types. Middle right: Average tensors (complete in gray) and Kagan angles between the average tensors for each rupture type. Bottom right: Rose diagram of maximum and minimum horizontal axes orientation.







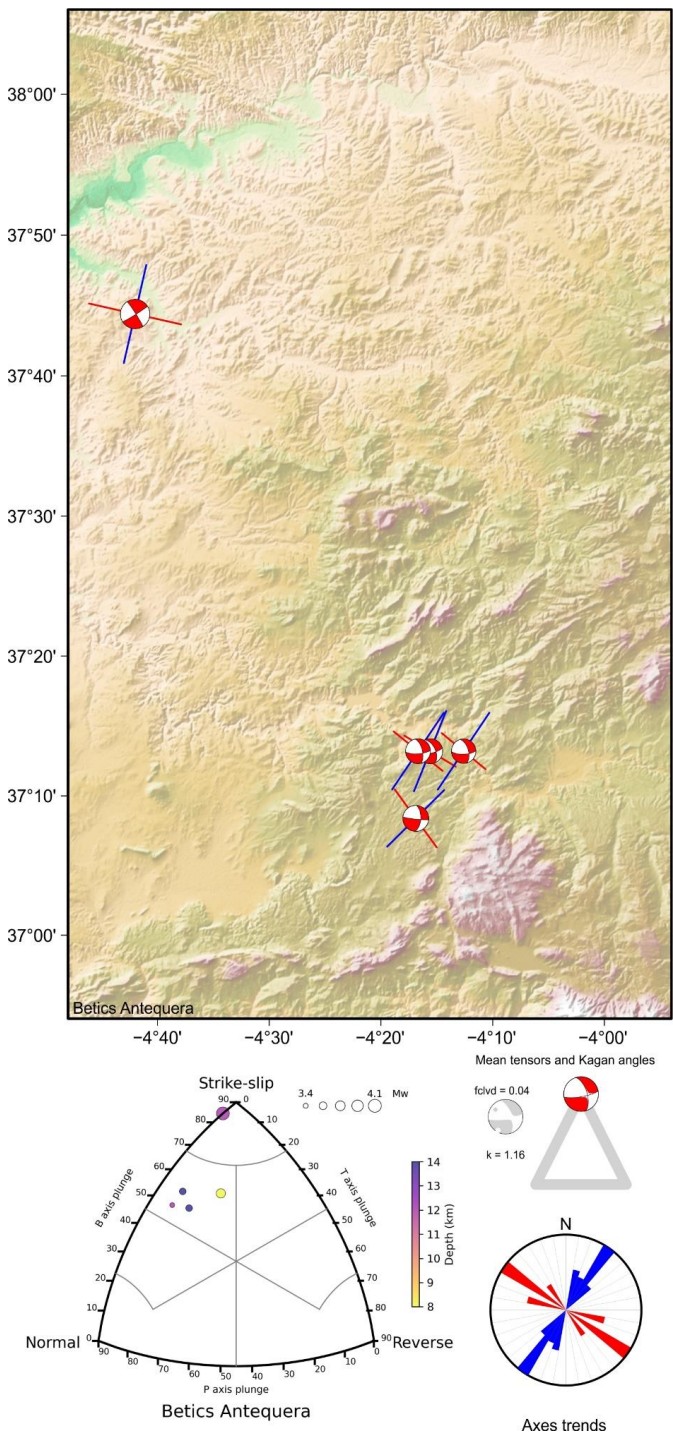

**Fig. A4: Betics Antequera tectonic zone. Top map: focal mechanisms with maximum horizontal axis (P or B for normal ruptures) and minimum horizontal axis (T or B for reverse ruptures) orientation. Bottom left classification diagram for earthquake rupture types. Middle right: Average tensors (complete in gray) and Kagan angles between the average tensors for each rupture type. Bottom right: Rose diagram of maximum and minimum horizontal axes orientation.**



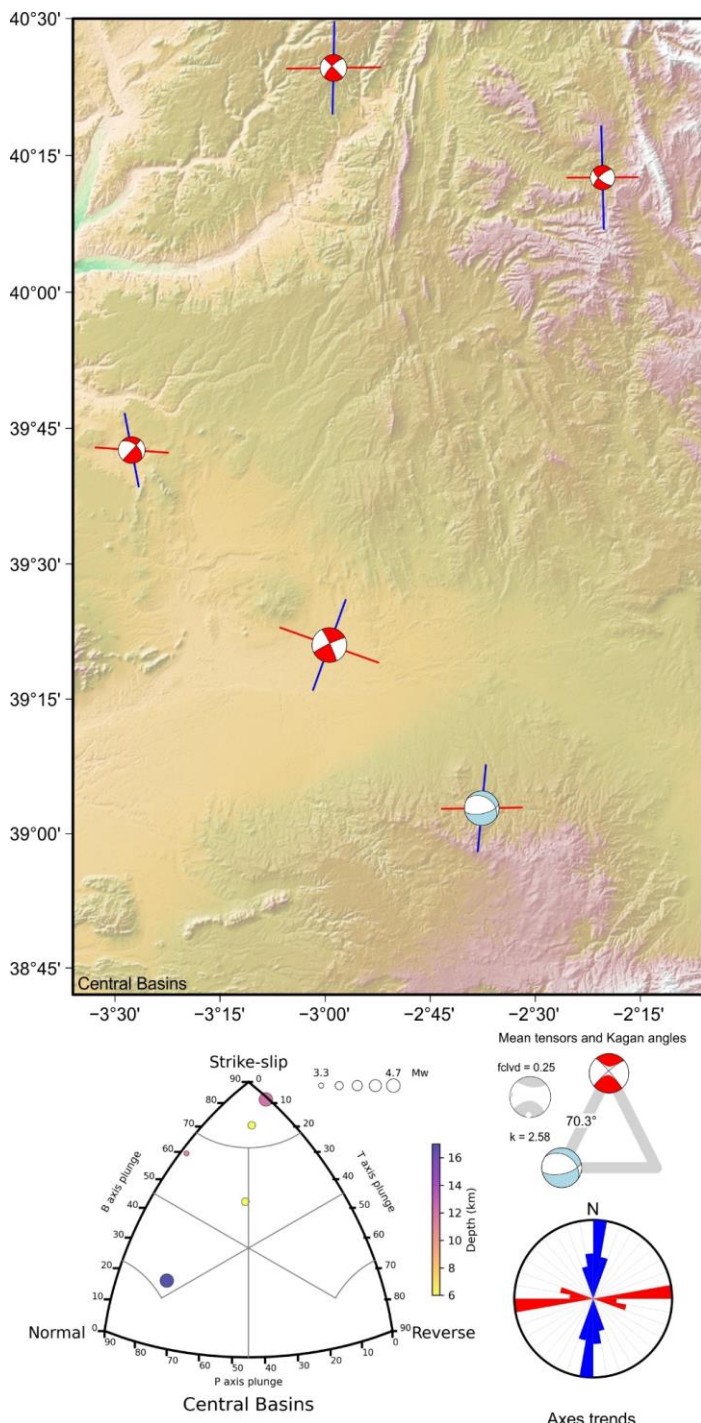

**Fig. A5: Central Basins tectonic zone. Top map: focal mechanisms with maximum horizontal axis (P or B for normal ruptures) and minimum horizontal axis (T or B for reverse ruptures) orientation. Bottom left classification diagram for earthquake rupture types. Middle right: Average tensors (complete in gray) and Kagan angles between the average tensors for each rupture type. Bottom right: Rose diagram of maximum and minimum horizontal axes orientation.**



**Fig. A6: Central Pyrenees tectonic zone. Top map: focal mechanisms with maximum horizontal axis (P or B for normal ruptures) and minimum horizontal axis (T or B for reverse ruptures) orientation. Bottom left classification diagram for earthquake rupture types. Middle right: Average tensors (complete in gray) and Kagan angles between the average tensors for each rupture type. Bottom right: Rose diagram of maximum and minimum horizontal axes orientation.**





**Fig. A7: Eastern Betics tectonic zone. Top map: focal mechanisms with maximum horizontal axis (P or B for normal ruptures) and minimum horizontal axis (T or B for reverse ruptures) orientation. Bottom left classification diagram for earthquake rupture types. Middle right: Average tensors (complete in gray) and Kagan angles between the average tensors for each rupture type. Bottom right: Rose diagram of maximum and minimum horizontal axes orientation.**





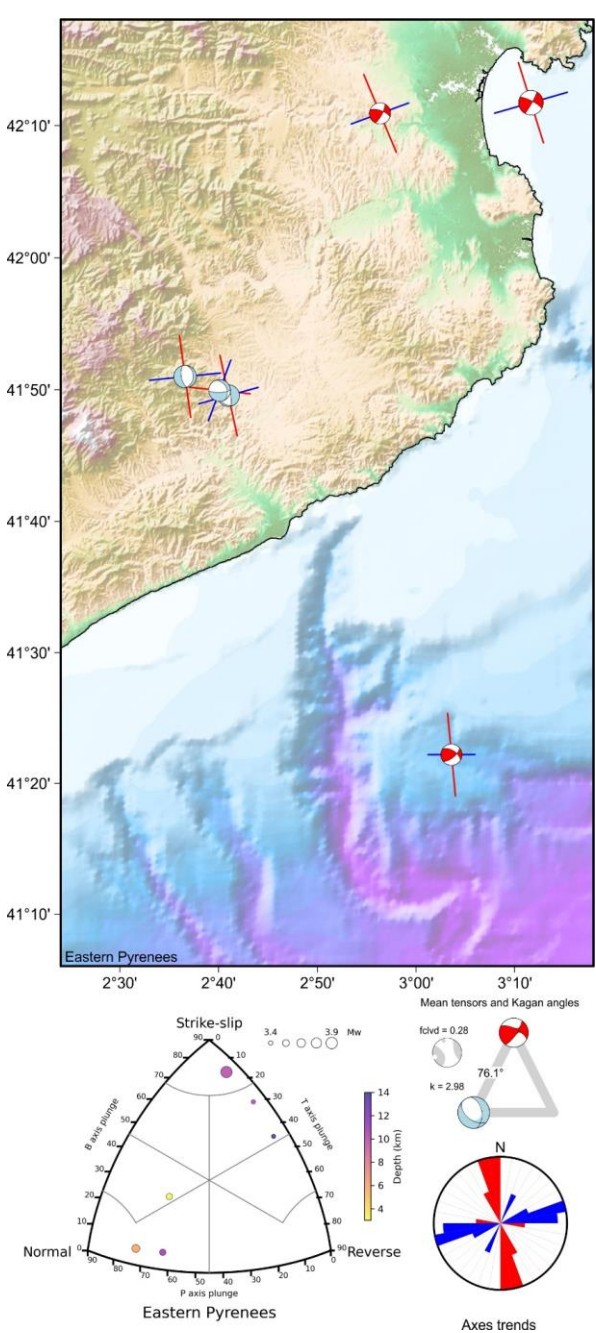

Fig. A8: Eastern Pyrenees tectonic zone. Top map: focal mechanisms with maximum horizontal axis (P or B for normal ruptures) and minimum horizontal axis (T or B for reverse ruptures) orientation. Bottom left classification diagram for earthquake rupture types. Middle right: Average tensors (complete in gray) and Kagan angles between the average tensors for each rupture type. Bottom right: Rose diagram of maximum and minimum horizontal axes orientation.





**Fig. A9: East Tell Atlas tectonic zone. Top map: focal mechanisms with maximum horizontal axis (P or B for normal ruptures) and minimum horizontal axis (T or B for reverse ruptures) orientation. Bottom left classification diagram for earthquake rupture types. Middle right: Average tensors (complete in gray) and Kagan angles between the average tensors for each rupture type. Bottom right: Rose diagram of maximum and minimum horizontal axes orientation.**







**Fig. A10: Granada Basin tectonic zone. Top map: focal mechanisms with maximum horizontal axis (P or B for normal ruptures) and minimum horizontal axis (T or B for reverse ruptures) orientation. Bottom left classification diagram for earthquake rupture types. Middle right: Average tensors (complete in gray) and Kagan angles between the average tensors for each rupture type. Bottom right: Rose diagram of maximum and minimum horizontal axes orientation.**



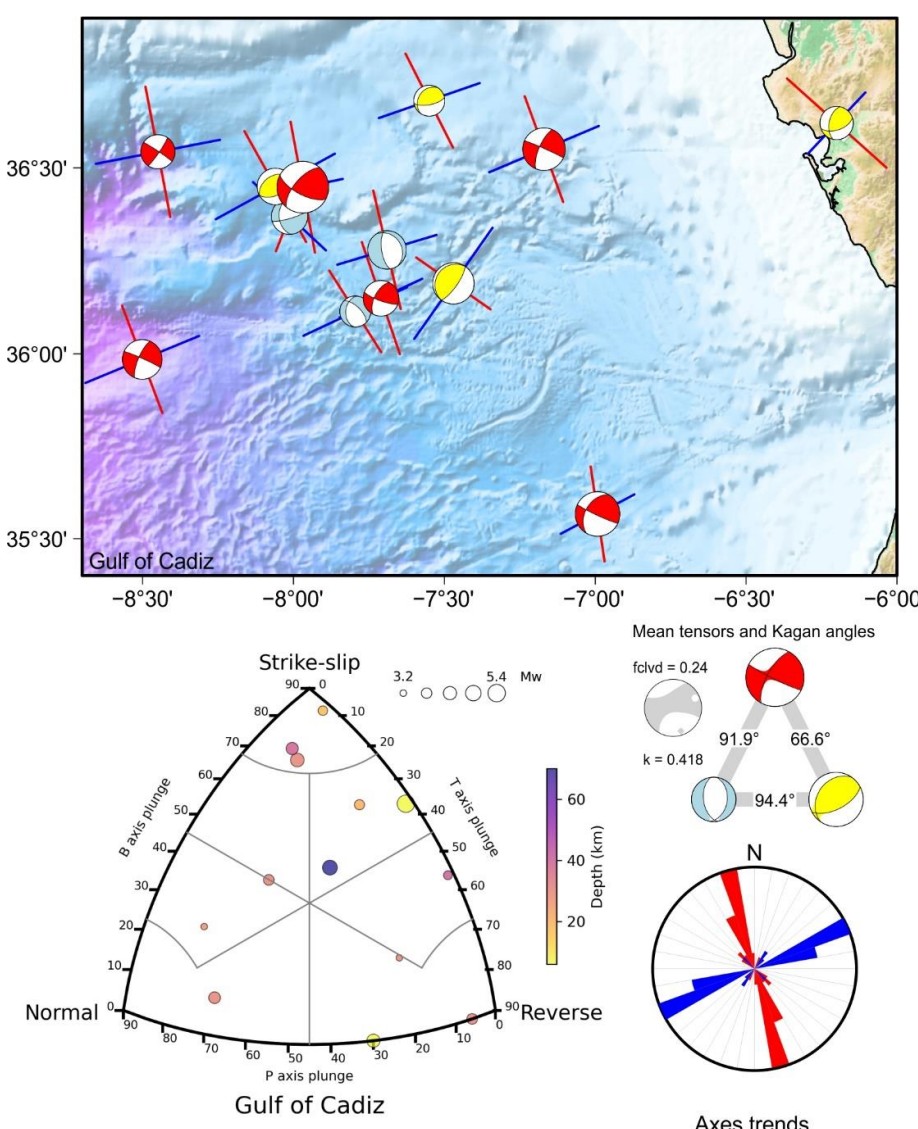

**Fig. A11: Gulf of Cadiz tectonic zone. Top map: focal mechanisms with maximum horizontal axis (P or B for normal ruptures) and minimum horizontal axis (T or B for reverse ruptures) orientation. Bottom left classification diagram for earthquake rupture types. Middle right: Average tensors (complete in gray) and Kagan angles between the average tensors for each rupture type. Bottom right: Rose diagram of maximum and minimum horizontal axes orientation.**





**Fig. A12: Gorringe-Horseshoe tectonic zone.** Top map: focal mechanisms with maximum horizontal axis (P or B for normal ruptures) and minimum horizontal axis (T or B for reverse ruptures) orientation. Bottom left classification diagram for earthquake rupture types. Middle right: Average tensors (complete in gray) and Kagan angles between the average tensors for each rupture type. Bottom right: Rose diagram of maximum and minimum horizontal axes orientation.





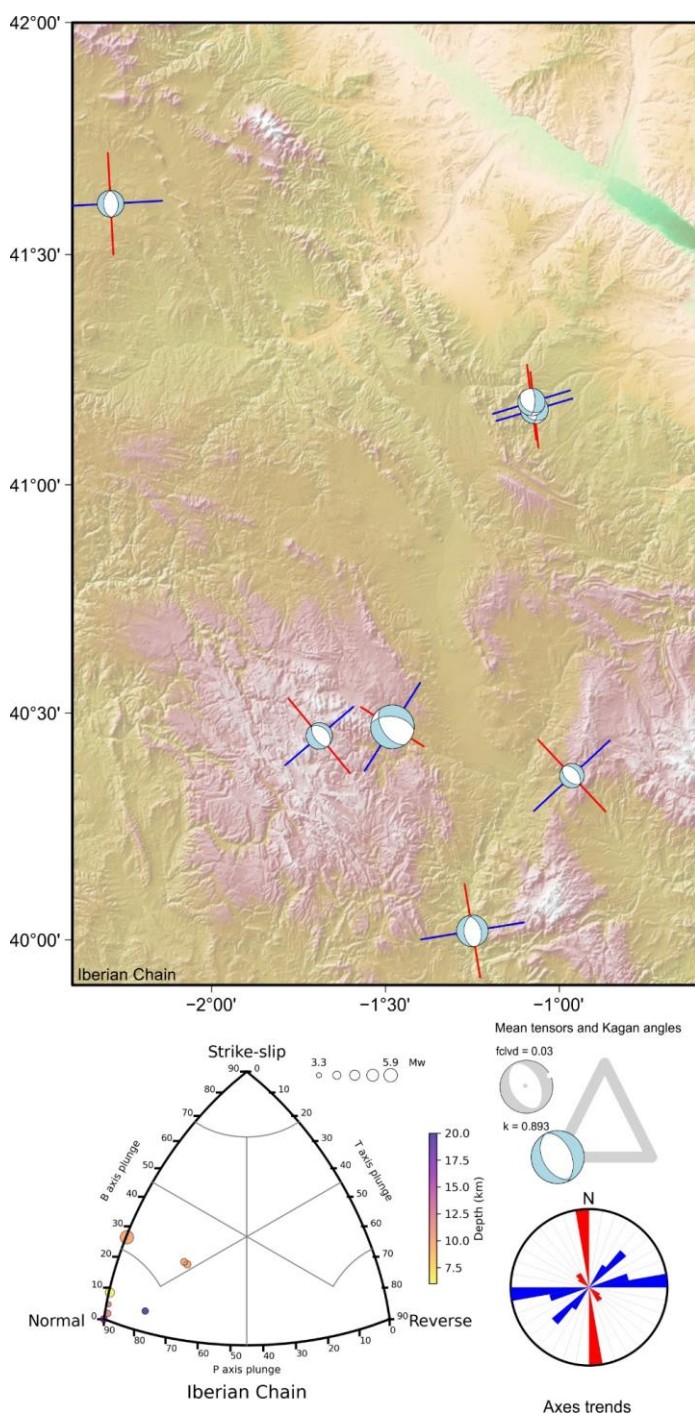

**Fig. A13: Iberian Chain tectonic zone. Top map: focal mechanisms with maximum horizontal axis (P or B for normal ruptures) and minimum horizontal axis (T or B for reverse ruptures) orientation. Bottom left classification diagram for earthquake rupture types. Middle right: Average tensors (complete in gray) and Kagan angles between the average tensors for each rupture type. Bottom right: Rose diagram of maximum and minimum horizontal axes orientation.**



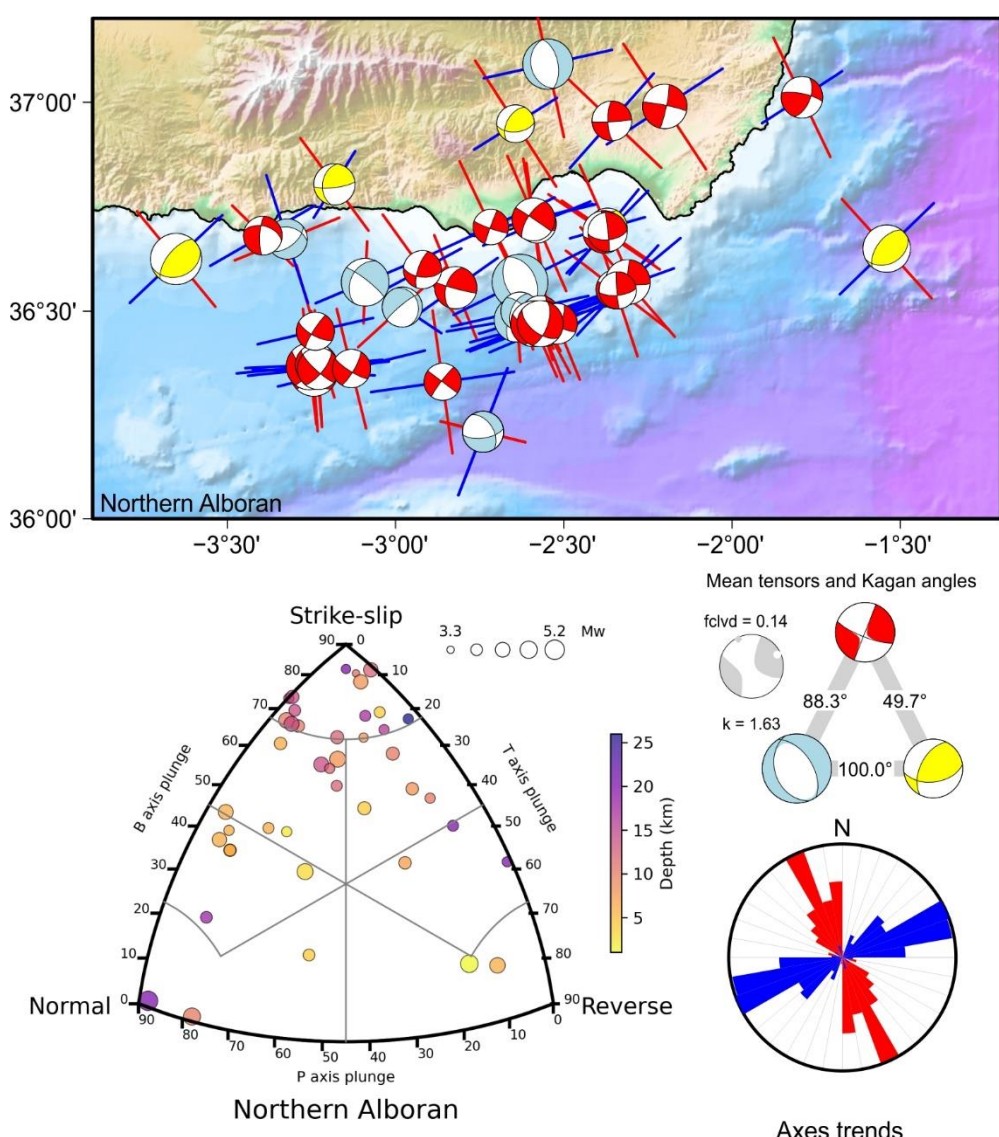

**Fig. A14: Northern Alboran tectonic zone. Top map: focal mechanisms with maximum horizontal axis (P or B for normal ruptures) and minimum horizontal axis (T or B for reverse ruptures) orientation. Bottom left classification diagram for earthquake rupture types. Middle right: Average tensors (complete in gray) and Kagan angles between the average tensors for each rupture type. Bottom right: Rose diagram of maximum and minimum horizontal axes orientation.**





**Fig. A15: NW Galicia tectonic zone. Top map: focal mechanisms with maximum horizontal axis (P or B for normal ruptures) and minimum horizontal axis (T or B for reverse ruptures) orientation. Bottom left classification diagram for earthquake rupture types. Middle right: Average tensors (complete in gray) and Kagan angles between the average tensors for each rupture type. Bottom right: Rose diagram of maximum and minimum horizontal axes orientation.**







**Fig. A16: Offshore Atlantic tectonic zone. Top map: focal mechanisms with maximum horizontal axis (P or B for normal ruptures) and minimum horizontal axis (T or B for reverse ruptures) orientation. Bottom left classification diagram for earthquake rupture types. Middle right: Average tensors (complete in gray) and Kagan angles between the average tensors for each rupture type. Bottom right: Rose diagram of maximum and minimum horizontal axes orientation.**



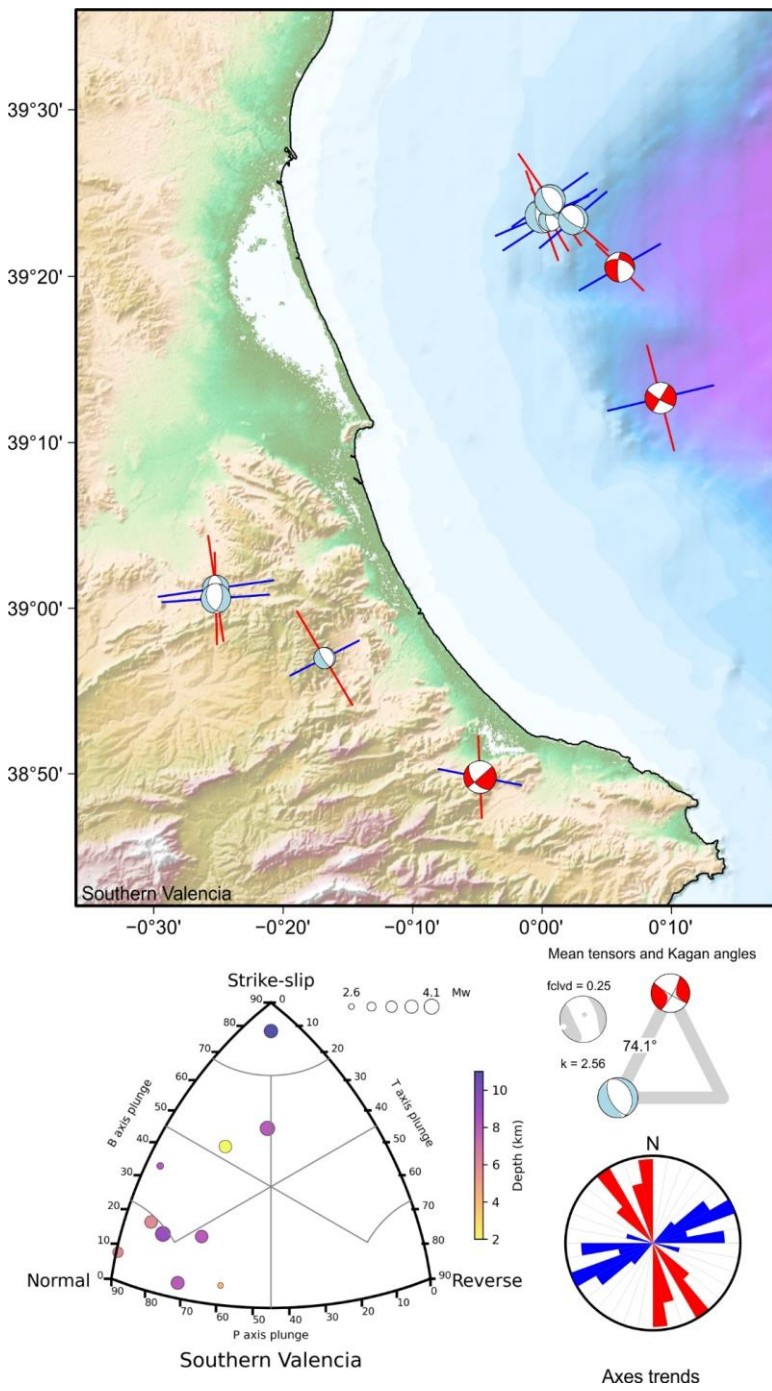

**Fig. A17: Southern Valencia tectonic zone. Top map: focal mechanisms with maximum horizontal axis (P or B for normal ruptures) and minimum horizontal axis (T or B for reverse ruptures) orientation. Bottom left classification diagram for earthquake rupture types. Middle right: Average tensors (complete in gray) and Kagan angles between the average tensors for each rupture type. Bottom right: Rose diagram of maximum and minimum horizontal axes orientation.**



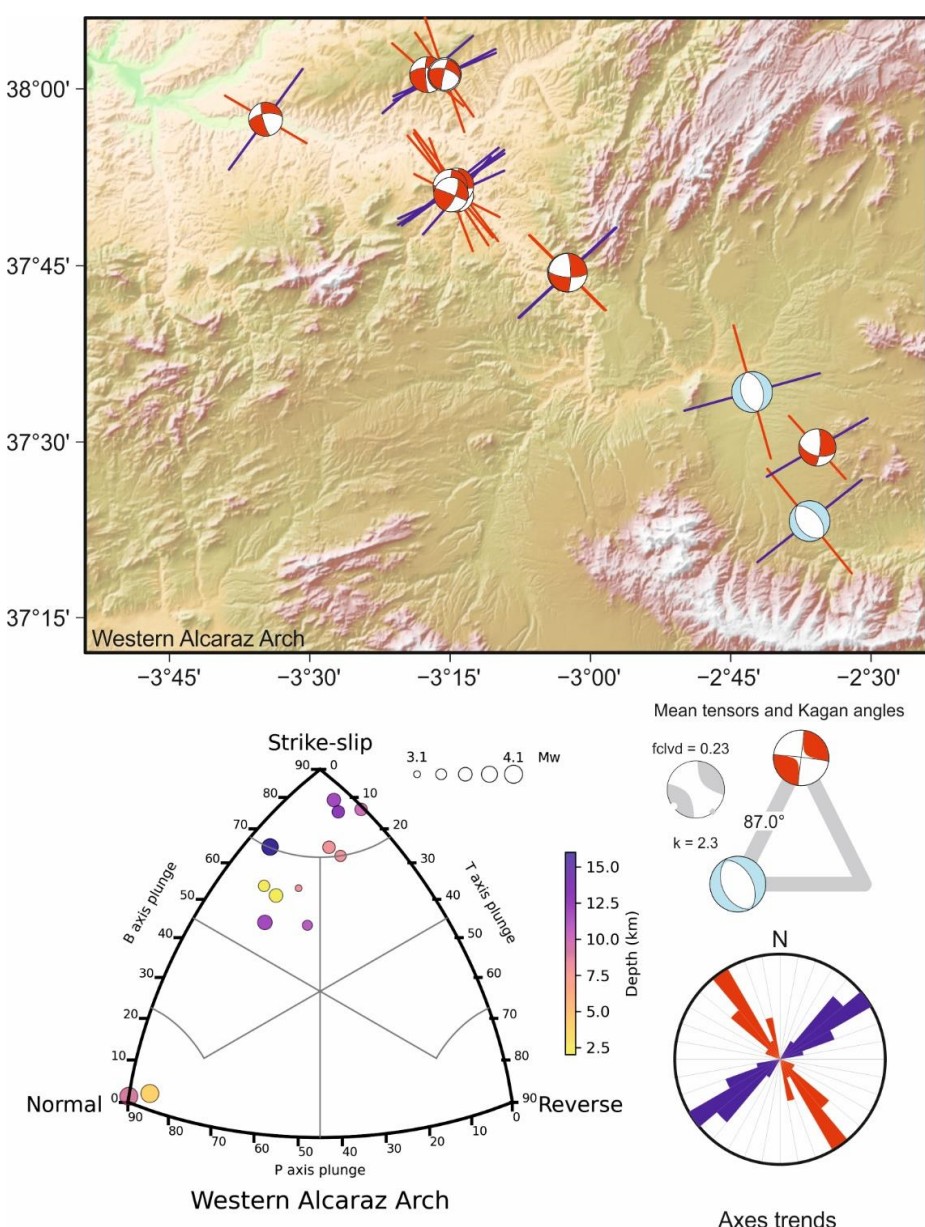

**Fig. A18: Western Alcaraz Arch tectonic zone. Top map: focal mechanisms with maximum horizontal axis (P or B for normal ruptures) and minimum horizontal axis (T or B for reverse ruptures) orientation. Bottom left classification diagram for earthquake rupture types. Middle right: Average tensors (complete in gray) and Kagan angles between the average tensors for each rupture type. Bottom right: Rose diagram of maximum and minimum horizontal axes orientation.**





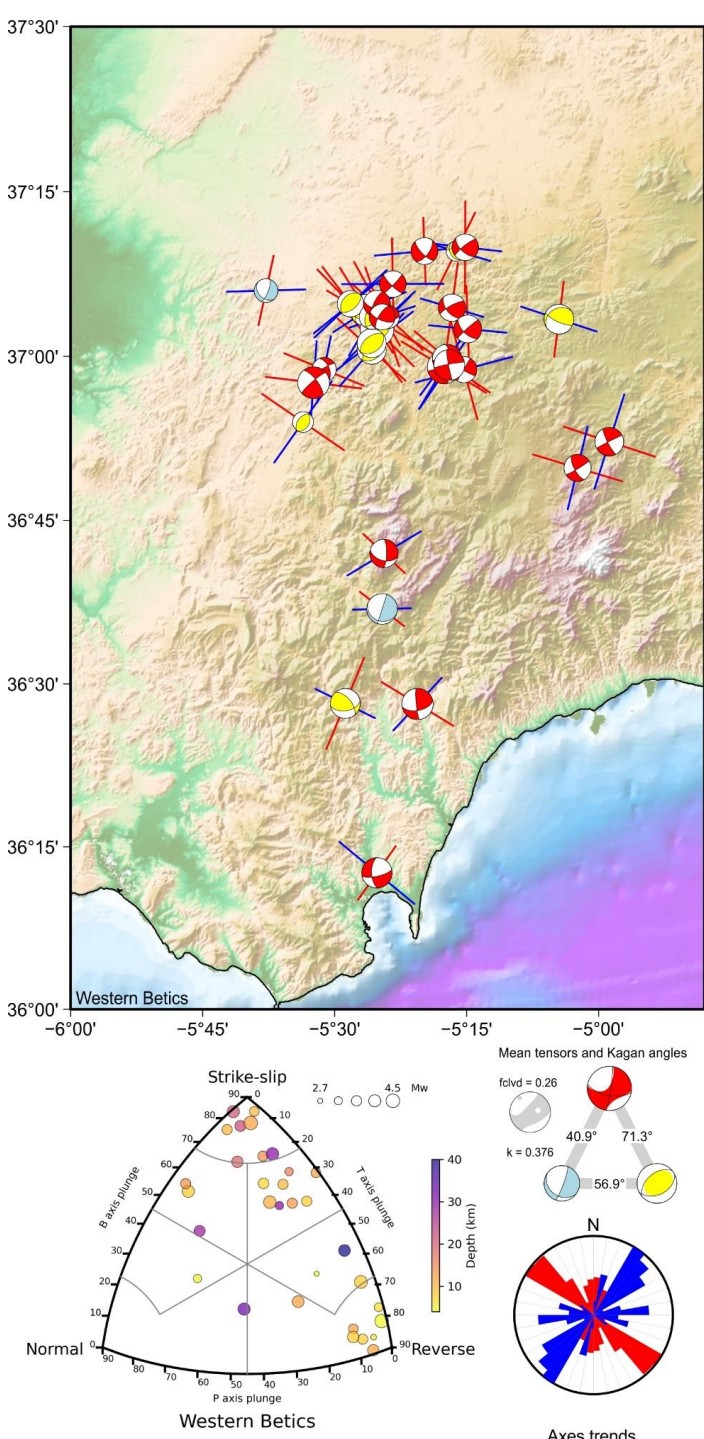

**Fig. A19: Western Betics tectonic zone. Top map: focal mechanisms with maximum horizontal axis (P or B for normal ruptures)**
1130 **and minimum horizontal axis (T or B for reverse ruptures) orientation. Bottom left classification diagram for earthquake rupture types. Middle right: Average tensors (complete in gray) and Kagan angles between the average tensors for each rupture type. Bottom right: Rose diagram of maximum and minimum horizontal axes orientation.**



**Fig. A20: Western Spanish Portugues Central System tectonic zone. Top map: focal mechanisms with maximum horizontal axis (P or B for normal ruptures) and minimum horizontal axis (T or B for reverse ruptures) orientation. Bottom left classification diagram for earthquake rupture types. Middle right: Average tensors (complete in gray) and Kagan angles between the average tensors for each rupture type. Bottom right: Rose diagram of maximum and minimum horizontal axes orientation.**





**Fig. A21: Western Pyrenees tectonic zone. Top map: focal mechanisms with maximum horizontal axis (P or B for normal ruptures) and minimum horizontal axis (T or B for reverse ruptures) orientation. Bottom left classification diagram for earthquake rupture types. Middle right: Average tensors (complete in gray) and Kagan angles between the average tensors for each rupture type. Bottom right: Rose diagram of maximum and minimum horizontal axes orientation.**





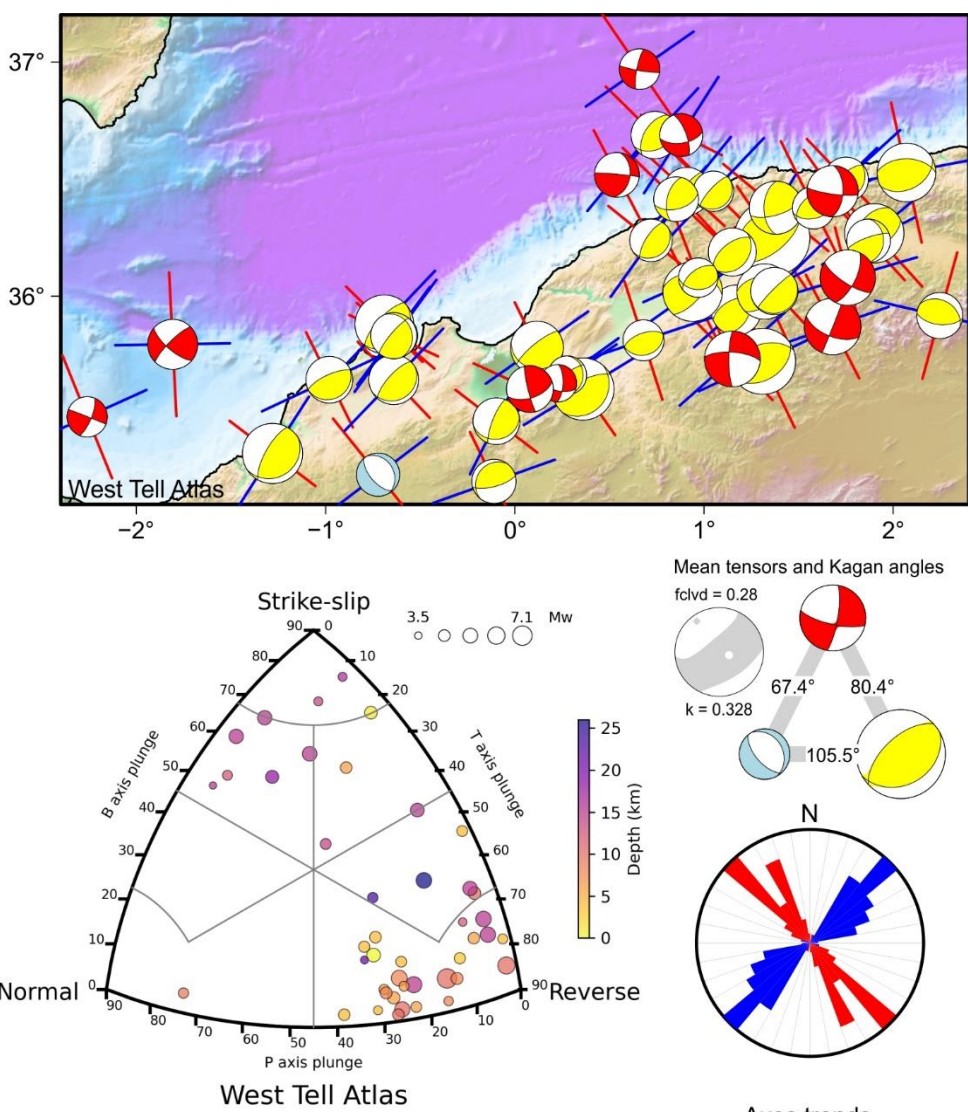

**Fig. A22: West Tell Atlas tectonic zone. Top map: focal mechanisms with maximum horizontal axis (P or B for normal ruptures) and minimum horizontal axis (T or B for reverse ruptures) orientation. Bottom left classification diagram for earthquake rupture types. Middle right: Average tensors (complete in gray) and Kagan angles between the average tensors for each rupture type. Bottom right: Rose diagram of maximum and minimum horizontal axes orientation.**





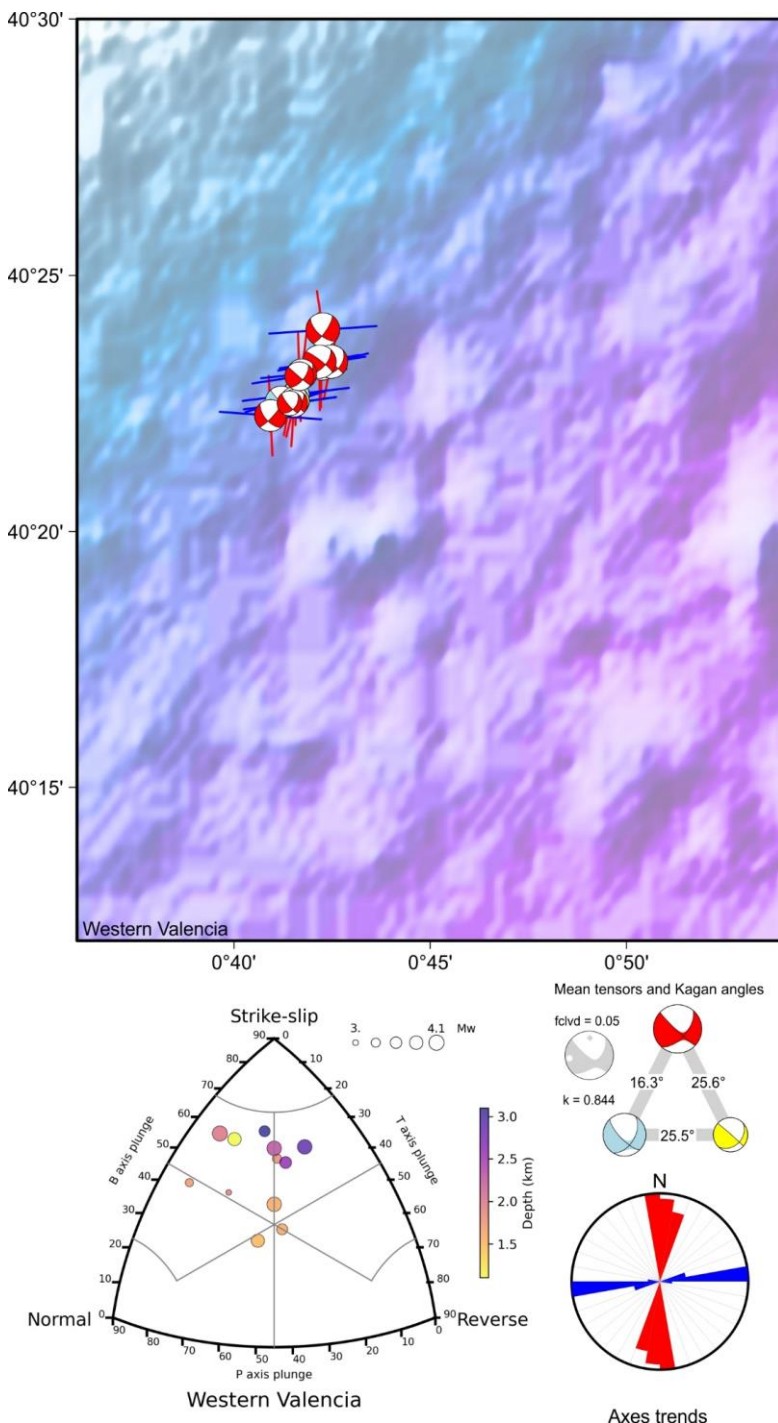

**Fig. A23: Western Valencia tectonic zone. Top map: focal mechanisms with maximum horizontal axis (P or B for normal ruptures) and minimum horizontal axis (T or B for reverse ruptures) orientation. Bottom left classification diagram for earthquake rupture types. Middle right: Average tensors (complete in gray) and Kagan angles between the average tensors for each rupture type. Bottom right: Rose diagram of maximum and minimum horizontal axes orientation.**





 **Appendix B**

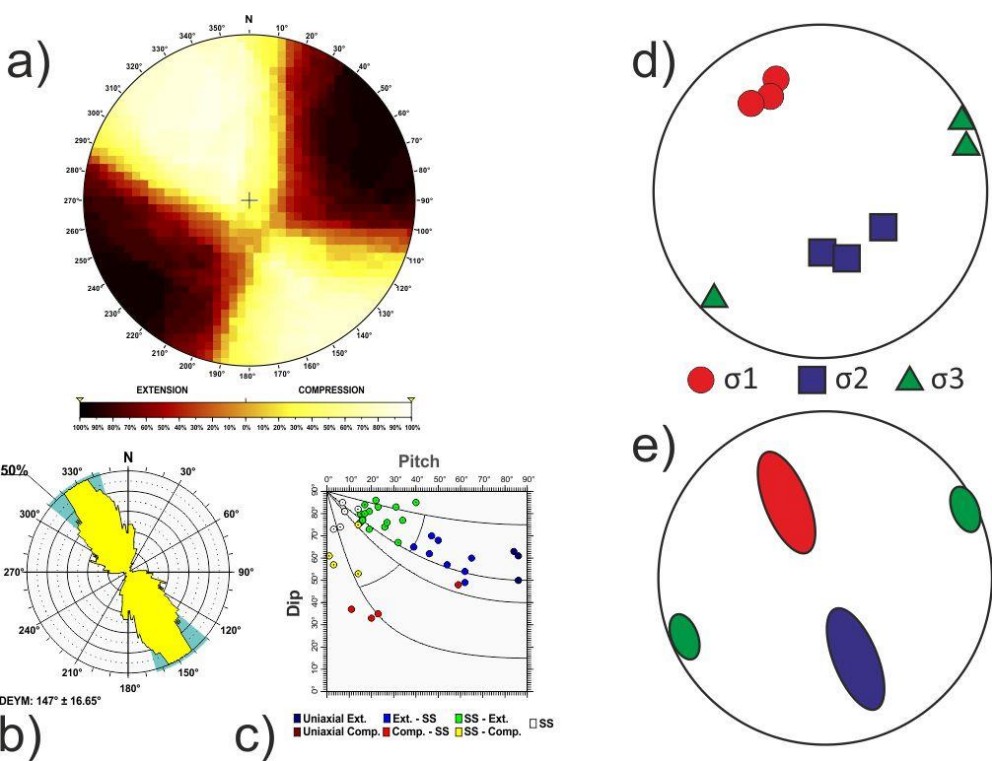

**Fig. B1 Al Hoceima tectonic zone. Results of the stress and strain analyses for different zones: a) Right Dihedra solution. b) Rose**
1160 **diagram of the Dey (horizontal shortening direction) obtained from the Slip model. C) Pitch/Dip plot for the neo-formed nodal**
**planes obtained from the Slip Model. d) Stress Inversion Results. e) Variability of the three principal stress axes of the stress**
**inversion. e) Variability of the three principal stress axes of the stress inversion.**

1165

1170



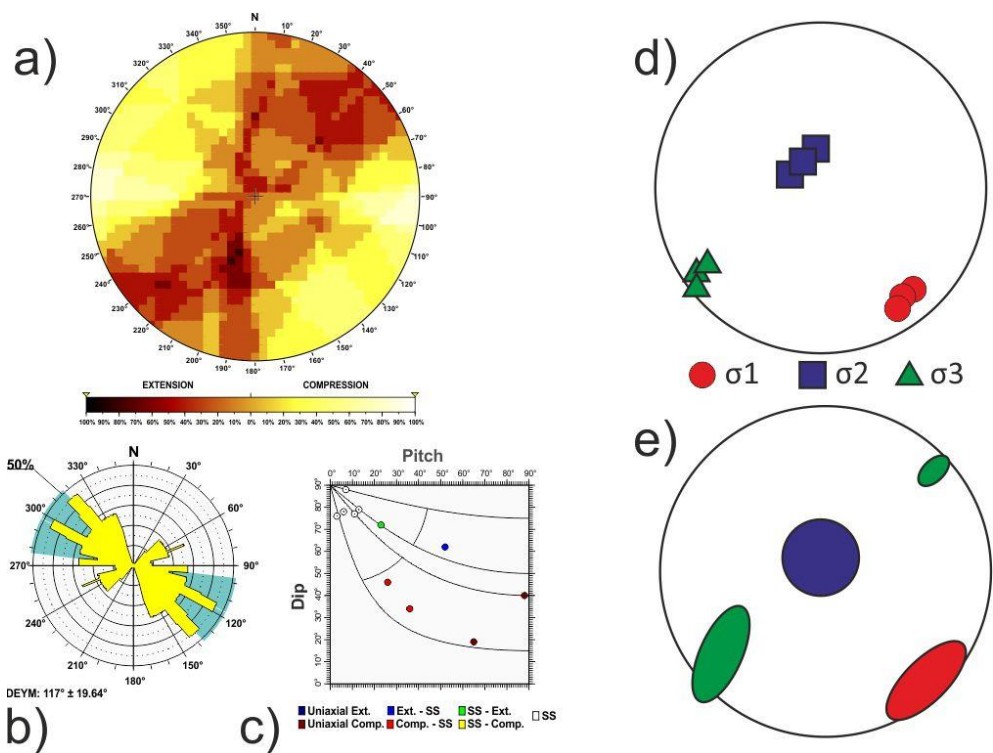

**Fig. B2 Algarve tectonic zone. Results of the stress and strain analyses for different zones: a) Right Dihedra solution. b) Rose diagram of the Dey (horizontal shortening direction) obtained from the Slip model. C) Pitch/Dip plot for the neo-formed nodal planes obtained from the Slip Model. d) Stress Inversion Results. e) Variability of the three principal stress axes of the stress inversion.**



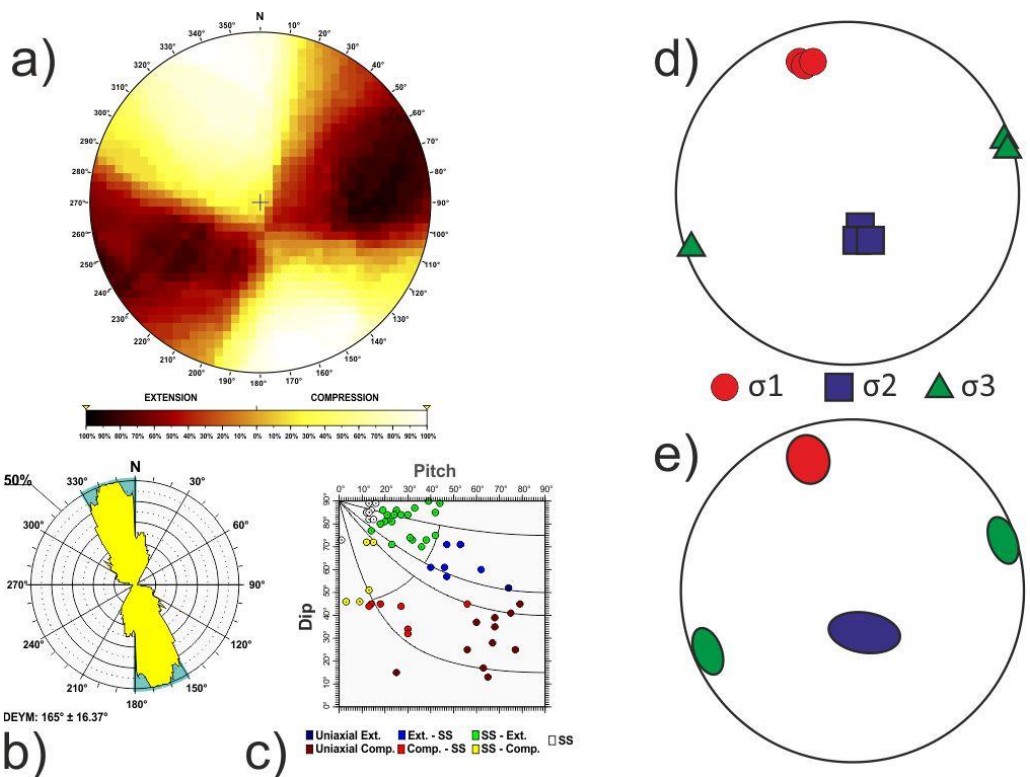

Fig. B3 Alboran Ridge tectonic zone. Results of the stress and strain analyses for different zones: a) Right Dihedra solution. b) Rose diagram of the Dey (horizontal shortening direction) obtained from the Slip model. C) Pitch/Dip plot for the neo-formed nodal planes obtained from the Slip Model. d) Stress Inversion Results. e) Variability of the three principal stress axes of the stress inversion.





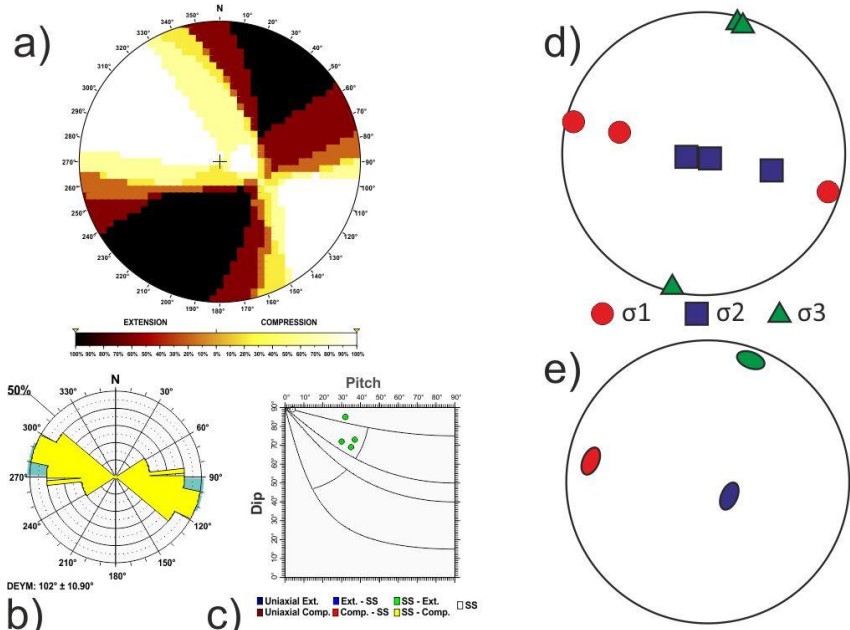

**Fig. B4 Betics Antequera tectonic zone. Results of the stress and strain analyses for different zones: a) Right Dihedra solution. b) Rose diagram of the Dey (horizontal shortening direction) obtained from the Slip model. C) Pitch/Dip plot for the neo-formed nodal planes obtained from the Slip Model. d) Stress Inversion Results. e) Variability of the three principal stress axes of the stress inversion.**



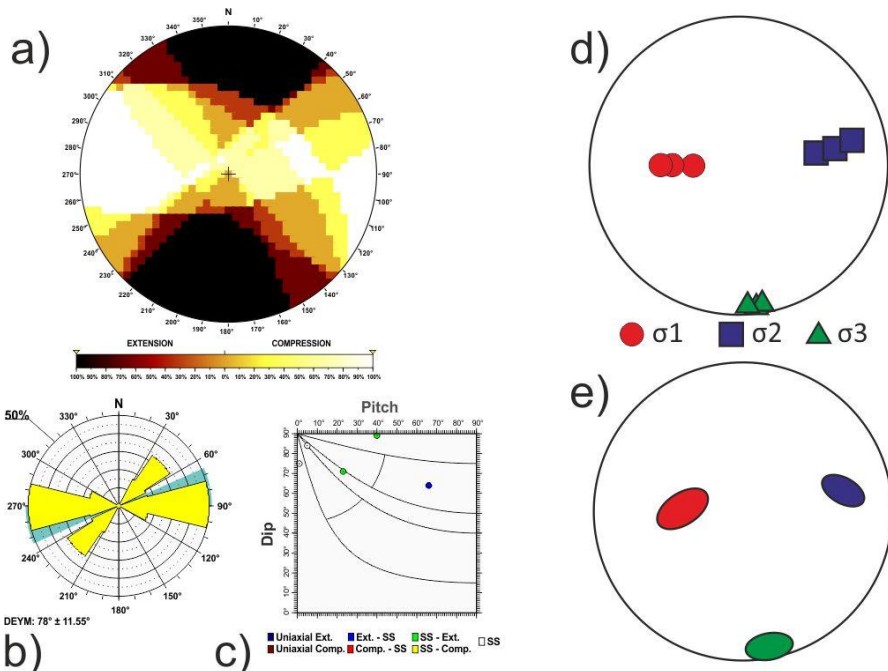

**Fig. B5 Central Basins tectonic zone. Results of the stress and strain analyses for different zones: a) Right Dihedra solution. b) Rose diagram of the Dey (horizontal shortening direction) obtained from the Slip model. C) Pitch/Dip plot for the neo-formed nodal planes obtained from the Slip Model. d) Stress Inversion Results. e) Variability of the three principal stress axes of the stress inversion.**






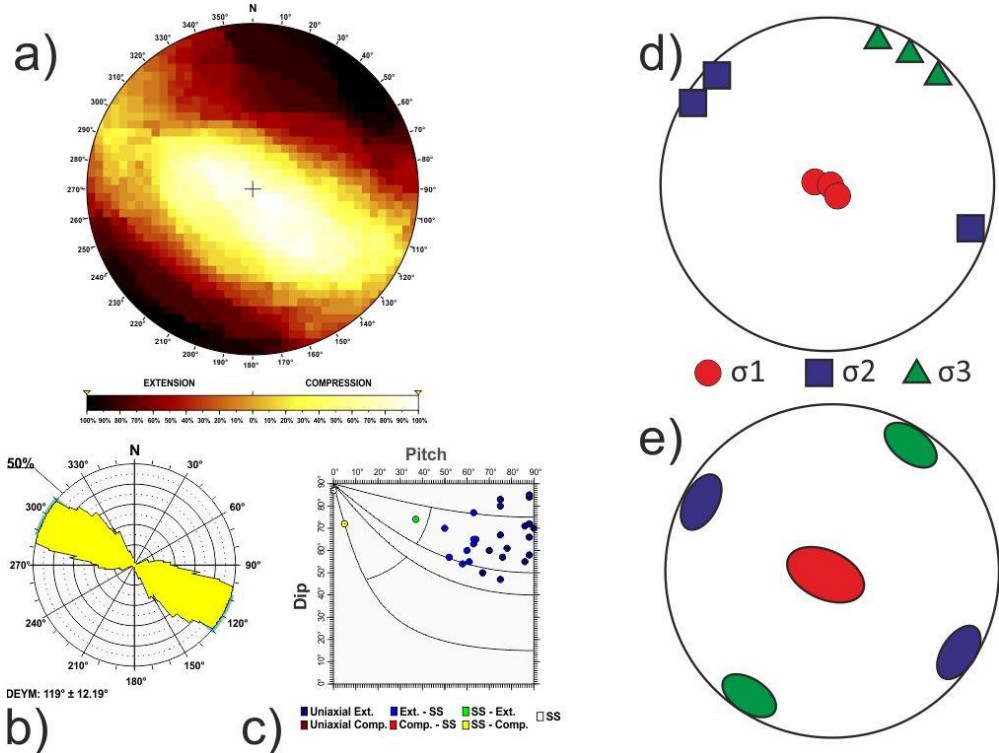

**Fig. B6 Central Pyrenees tectonic zone. Results of the stress and strain analyses for different zones: a) Right Dihedra solution. b) Rose diagram of the Dey (horizontal shortening direction) obtained from the Slip model. C) Pitch/Dip plot for the neo-formed nodal planes obtained from the Slip Model. d) Stress Inversion Results. e) Variability of the three principal stress axes of the stress inversion.**



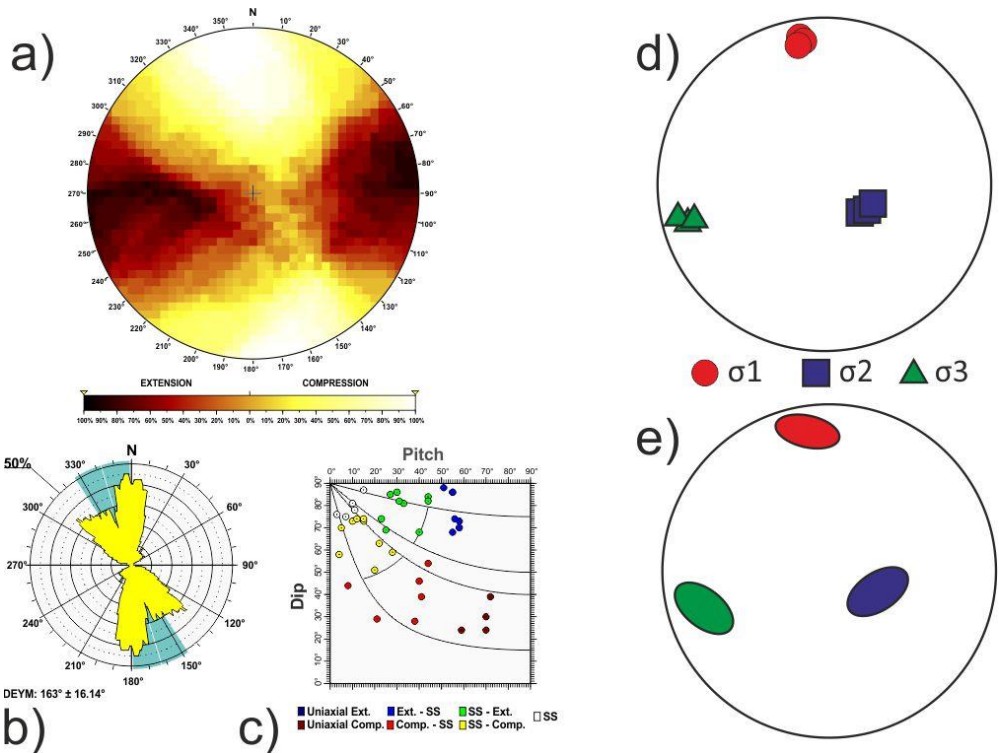


**Fig. B7 Eastern Betics tectonic zone. Results of the stress and strain analyses for different zones: a) Right Dihedra solution. b) Rose diagram of the Dey (horizontal shortening direction) obtained from the Slip model. C) Pitch/Dip plot for the neo-formed nodal planes obtained from the Slip Model. d) Stress Inversion Results. e) Variability of the three principal stress axes of the stress inversion.**







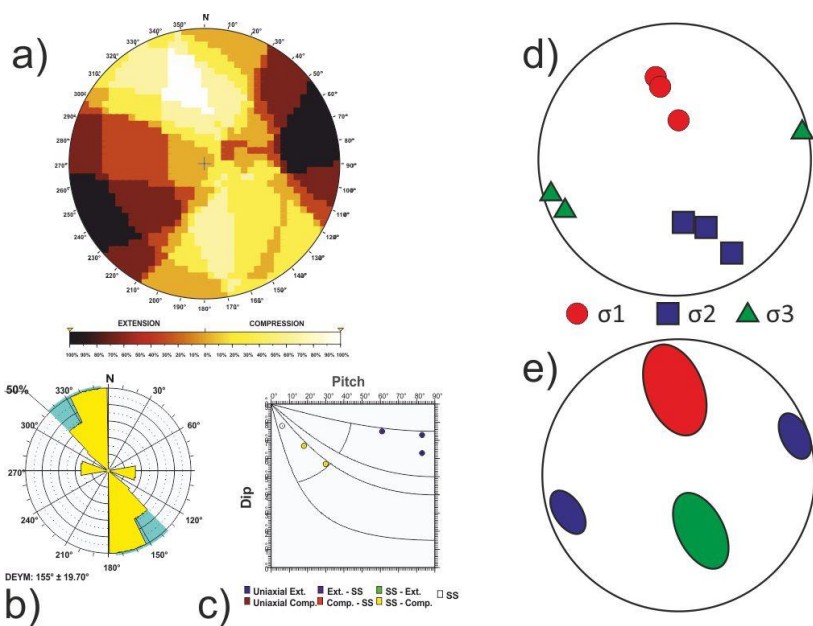

**Fig. B8 Eastern Pyrenees tectonic zone. Results of the stress and strain analyses for different zones: a) Right Dihedra solution. b) Rose diagram of the Dey (horizontal shortening direction) obtained from the Slip model. C) Pitch/Dip plot for the neo-formed nodal planes obtained from the Slip Model. d) Stress Inversion Results. e) Variability of the three principal stress axes of the stress inversion.**





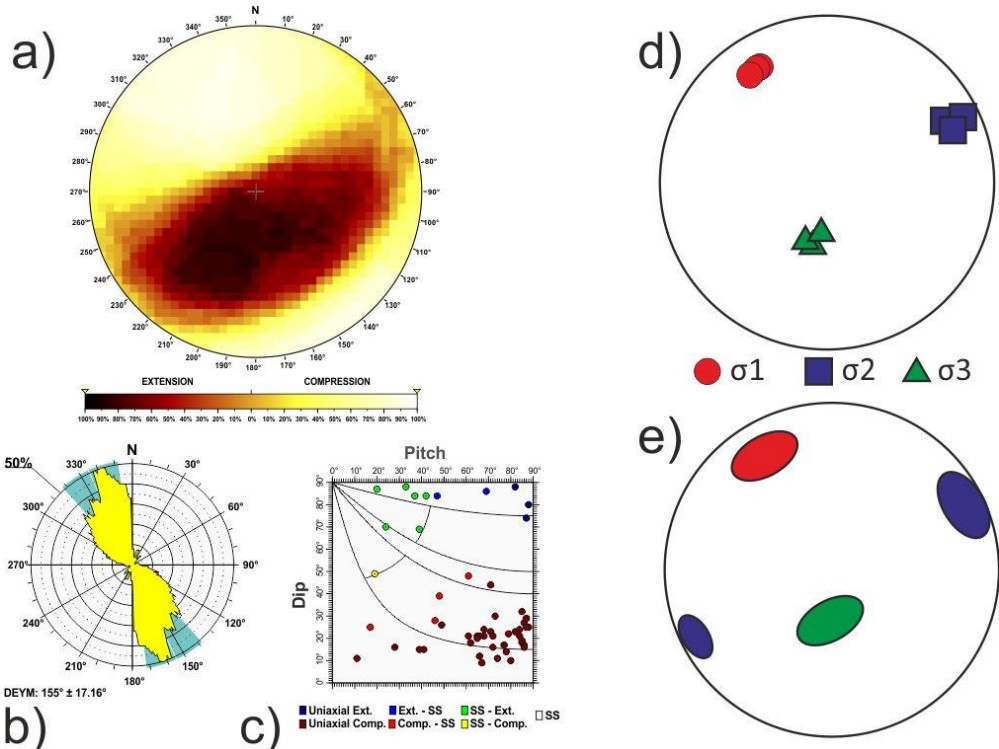

**Fig. B9 East Tell Atlas tectonic zone. Results of the stress and strain analyses for different zones: a) Right Dihedra solution. b) Rose diagram of the Dey (horizontal shortening direction) obtained from the Slip model. C) Pitch/Dip plot for the neo-formed nodal planes obtained from the Slip Model. d) Stress Inversion Results. e) Variability of the three principal stress axes of the stress inversion.**





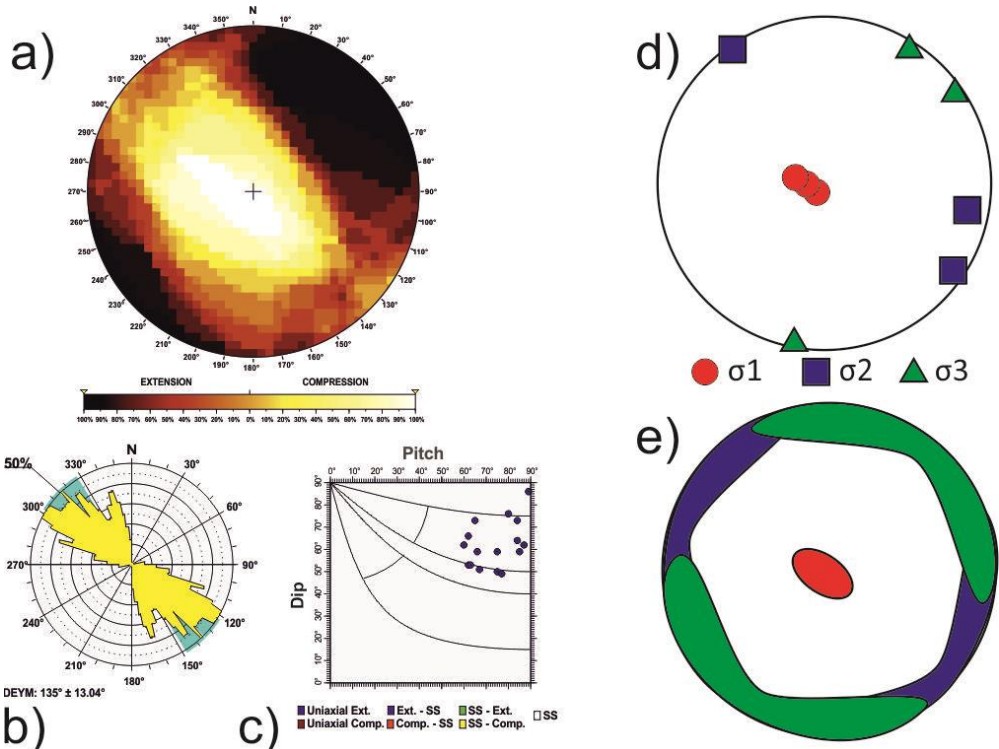

**Fig. B10 Granada Basin tectonic zone. Results of the stress and strain analyses for different zones: a) Right Dihedra solution. b)**
**Rose diagram of the Dey (horizontal shortening direction) obtained from the Slip model. C) Pitch/Dip plot for the neo-formed**
**nodal planes obtained from the Slip Model. d) Stress Inversion Results. e) Variability of the three principal stress axes of the stress**
**inversion.**





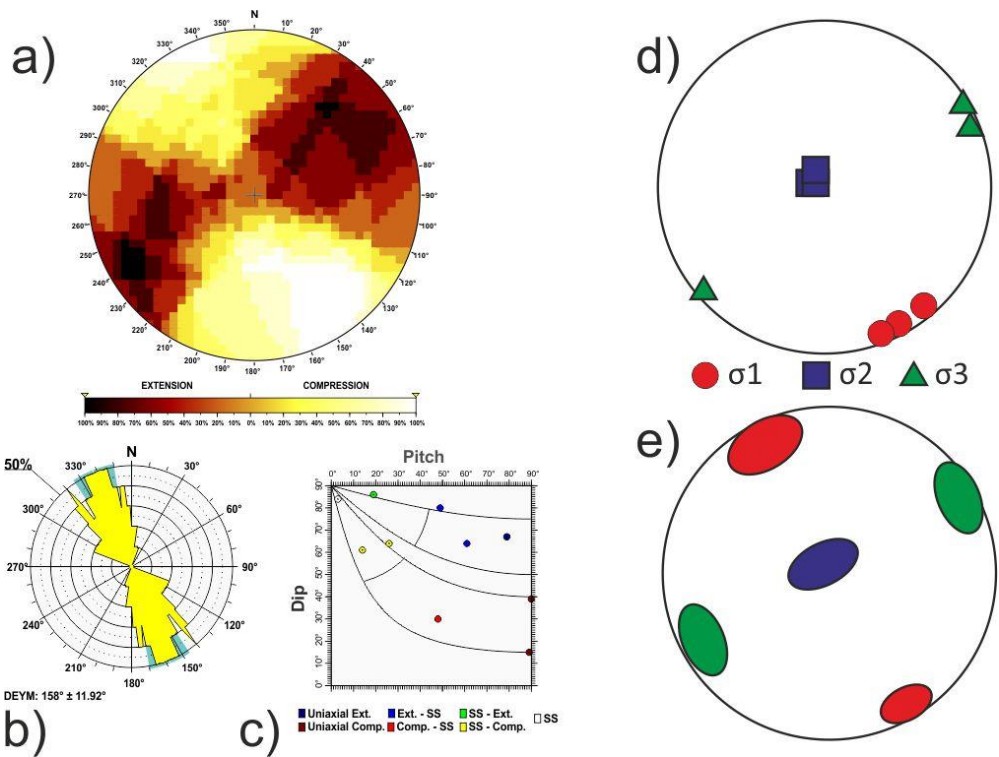


**Fig. B11 Gulf of Cadiz tectonic zone. Results of the stress and strain analyses for different zones: a) Right Dihedra solution. b) Rose diagram of the Dey (horizontal shortening direction) obtained from the Slip model. C) Pitch/Dip plot for the neo-formed nodal planes obtained from the Slip Model. d) Stress Inversion Results. e) Variability of the three principal stress axes of the stress inversion.**






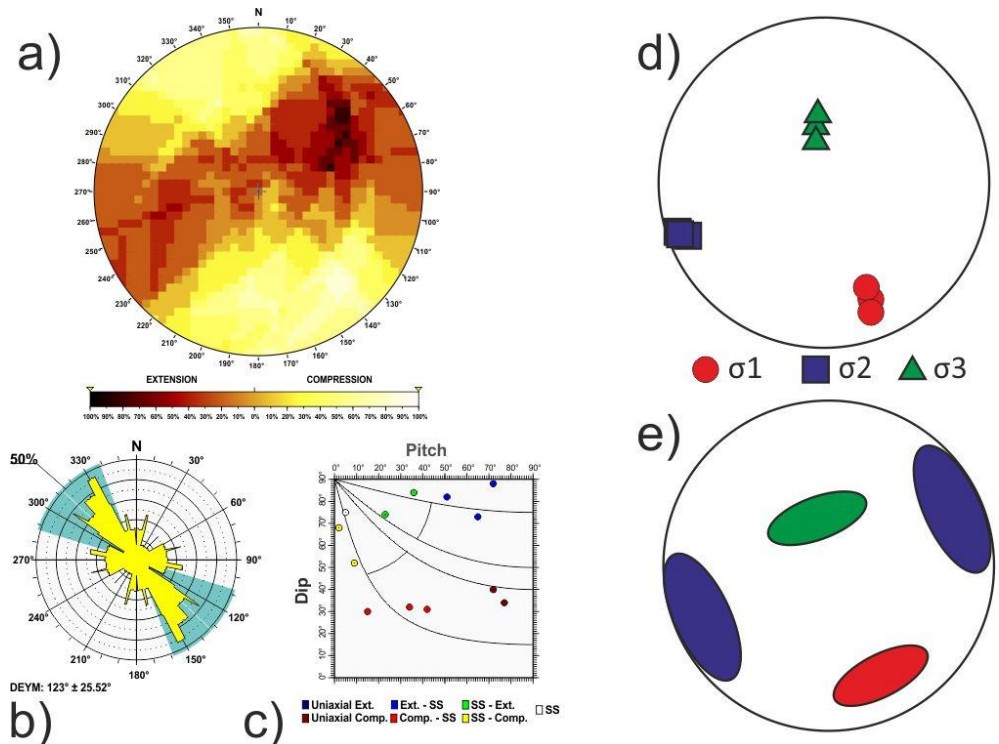

**Fig. B12 Gorringe - Horseshoe tectonic zone.  Results of the stress and strain analyses for different zones: a) Right Dihedra solution. b) Rose diagram of the Dey (horizontal shortening direction) obtained from the Slip model. C) Pitch/Dip plot for the neo-formed nodal planes obtained from the Slip Model. d) Stress Inversion Results. e) Variability of the three principal stress axes of the stress inversion.**



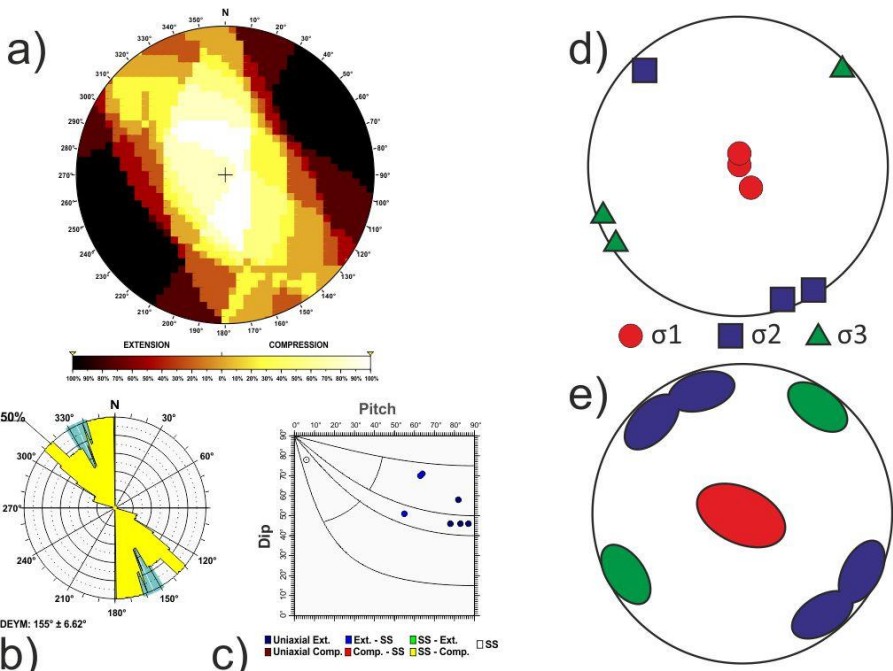

**Fig. B13 Iberian Chain tectonic zone. Results of the stress and strain analyses for different zones: a) Right Dihedra solution. b) Rose diagram of the Dey (horizontal shortening direction) obtained from the Slip model. C) Pitch/Dip plot for the neo-formed nodal planes obtained from the Slip Model. d) Stress Inversion Results. e) Variability of the three principal stress axes of the stress inversion.**





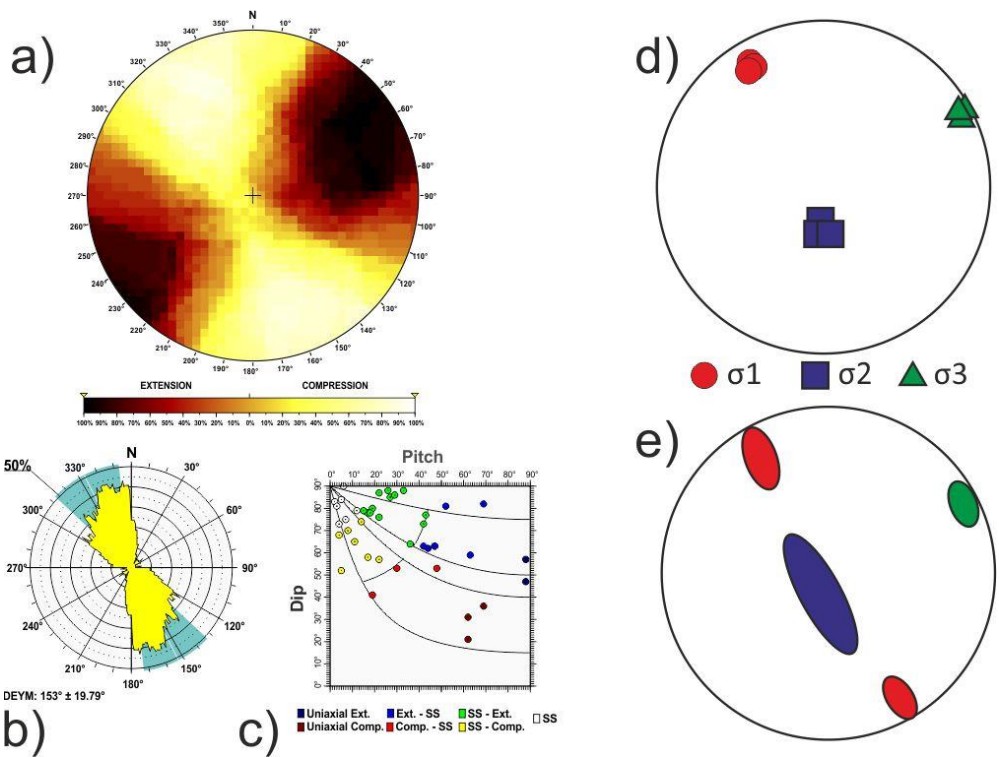

**Fig. B14 Northern Alboran tectonic zone. Results of the stress and strain analyses for different zones: a) Right Dihedra solution. b) Rose diagram of the Dey (horizontal shortening direction) obtained from the Slip model. C) Pitch/Dip plot for the neo-formed nodal planes obtained from the Slip Model. d) Stress Inversion Results. e) Variability of the three principal stress axes of the stress inversion.**



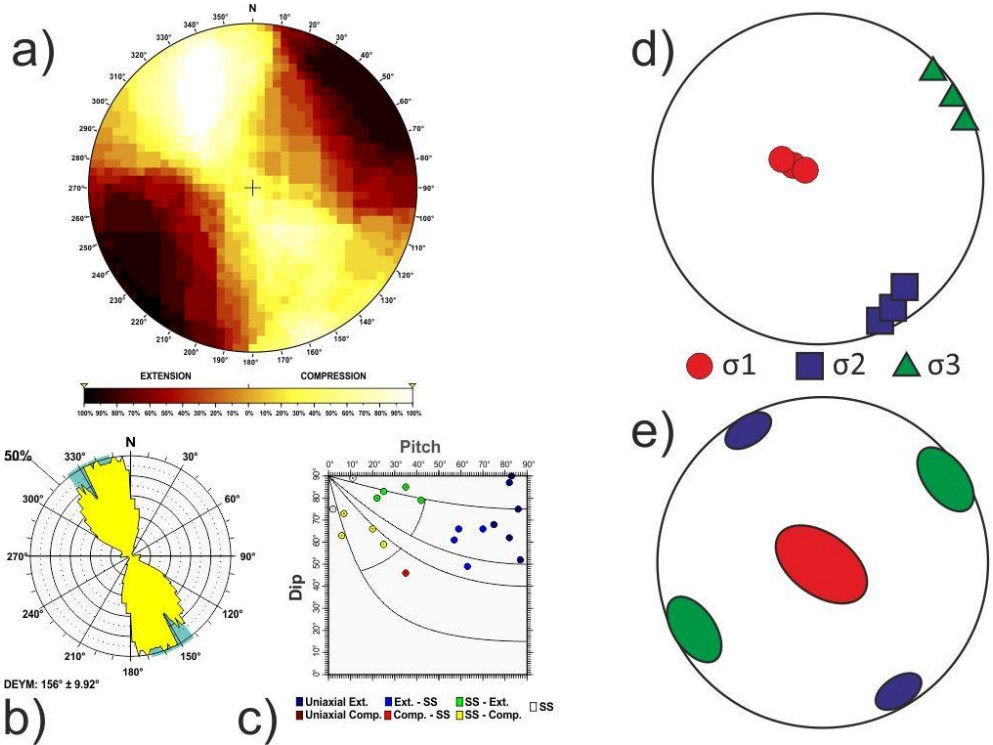

**Fig. B15 NW Galicia tectonic zone. Results of the stress and strain analyses for different zones: a) Right Dihedra solution. b) Rose diagram of the Dey (horizontal shortening direction) obtained from the Slip model. C) Pitch/Dip plot for the neo-formed nodal planes obtained from the Slip Model. d) Stress Inversion Results. e) Variability of the three principal stress axes of the stress inversion.**




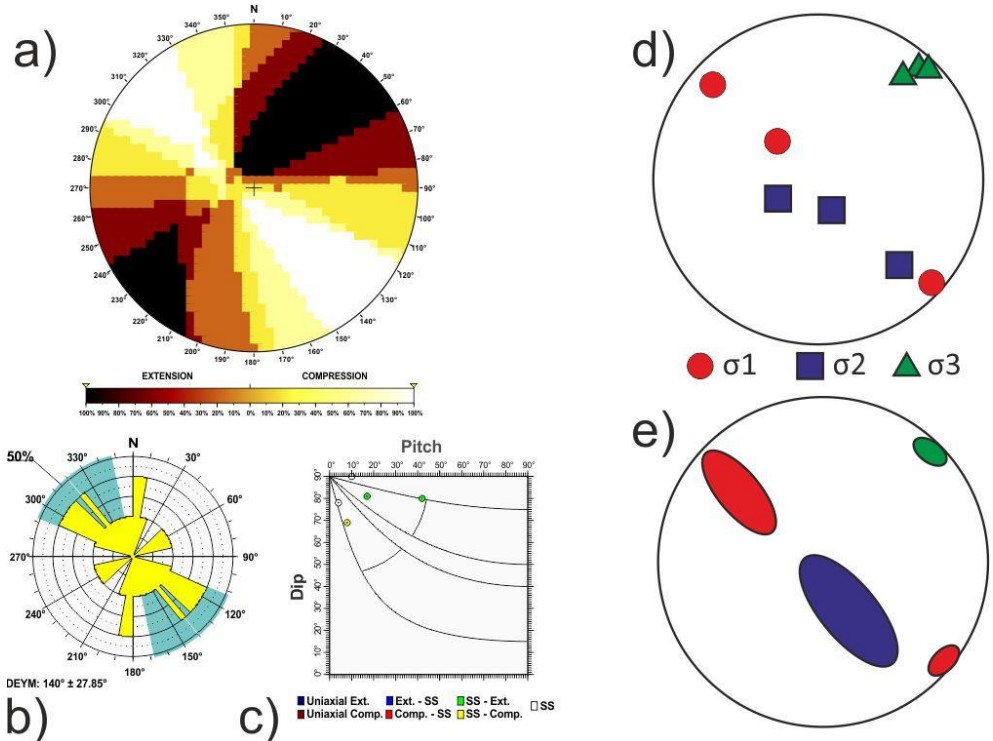

**Fig. B16 Offshore Atlantic tectonic zone. Results of the stress and strain analyses for different zones: a) Right Dihedra solution. b) Rose diagram of the Dey (horizontal shortening direction) obtained from the Slip model. C) Pitch/Dip plot for the neo-formed**
**nodal planes obtained from the Slip Model. d) Stress Inversion Results. e) Variability of the three principal stress axes of the stress inversion.**








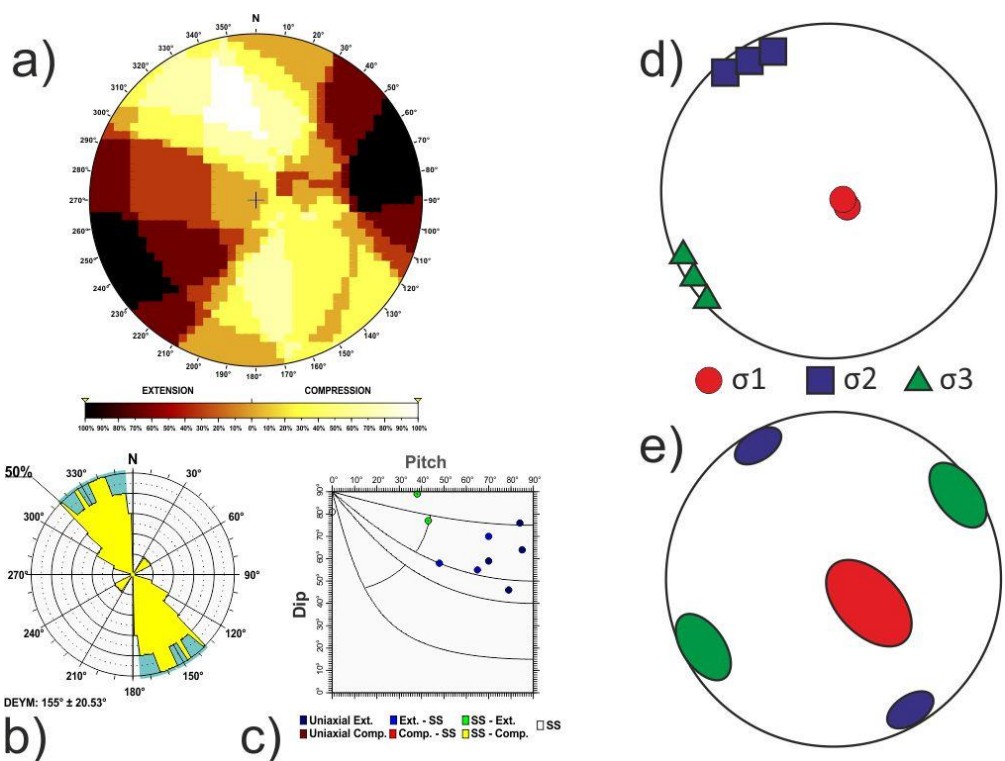


**Fig. B17 Southern Valencia tectonic zone. Results of the stress and strain analyses for different zones: a) Right Dihedra solution. b) Rose diagram of the Dey (horizontal shortening direction) obtained from the Slip model. C) Pitch/Dip plot for the neo-formed nodal planes obtained from the Slip Model. d) Stress Inversion Results. e) Variability of the three principal stress axes of the stress inversion.**




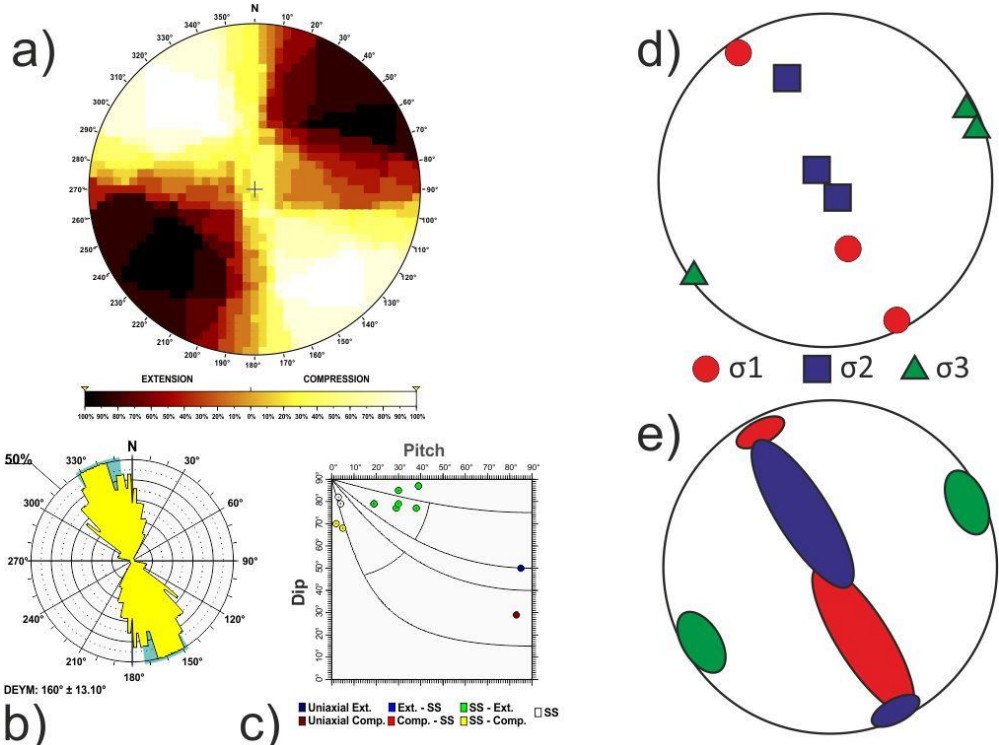


**Fig. B18 Western Alcaraz Arch tectonic zone. Results of the stress and strain analyses for different zones: a) Right Dihedra solution. b) Rose diagram of the Dey (horizontal shortening direction) obtained from the Slip model. C) Pitch/Dip plot for the neo-formed nodal planes obtained from the Slip Model. d) Stress Inversion Results. e) Variability of the three principal stress axes of the stress inversion.**





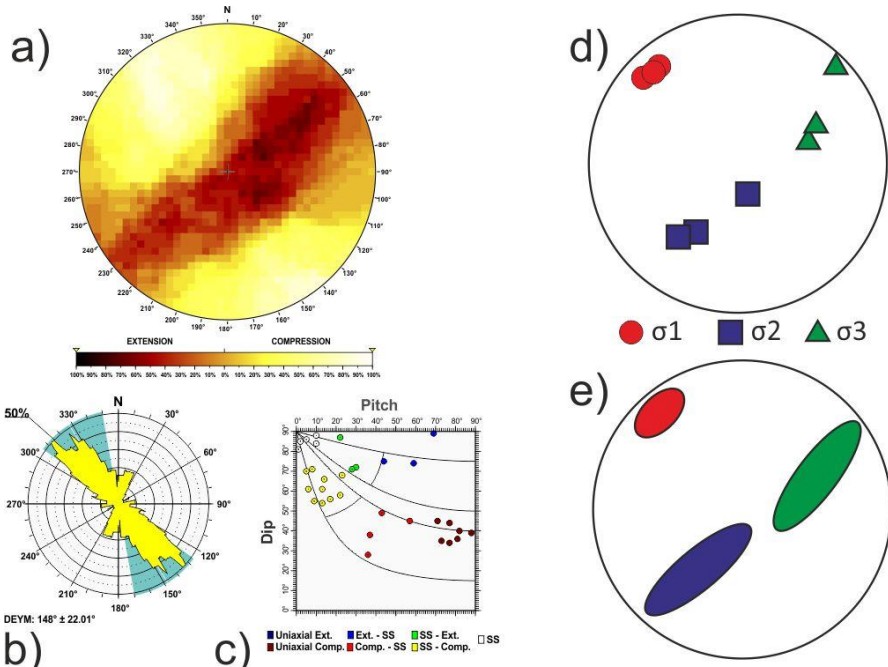

**Fig. B19 Western Betics tectonic zone. Results of the stress and strain analyses for different zones: a) Right Dihedra solution. b) Rose diagram of the Dey (horizontal shortening direction) obtained from the Slip model. C) Pitch/Dip plot for the neo-formed nodal planes obtained from the Slip Model. d) Stress Inversion Results. e) Variability of the three principal stress axes of the stress inversion.**





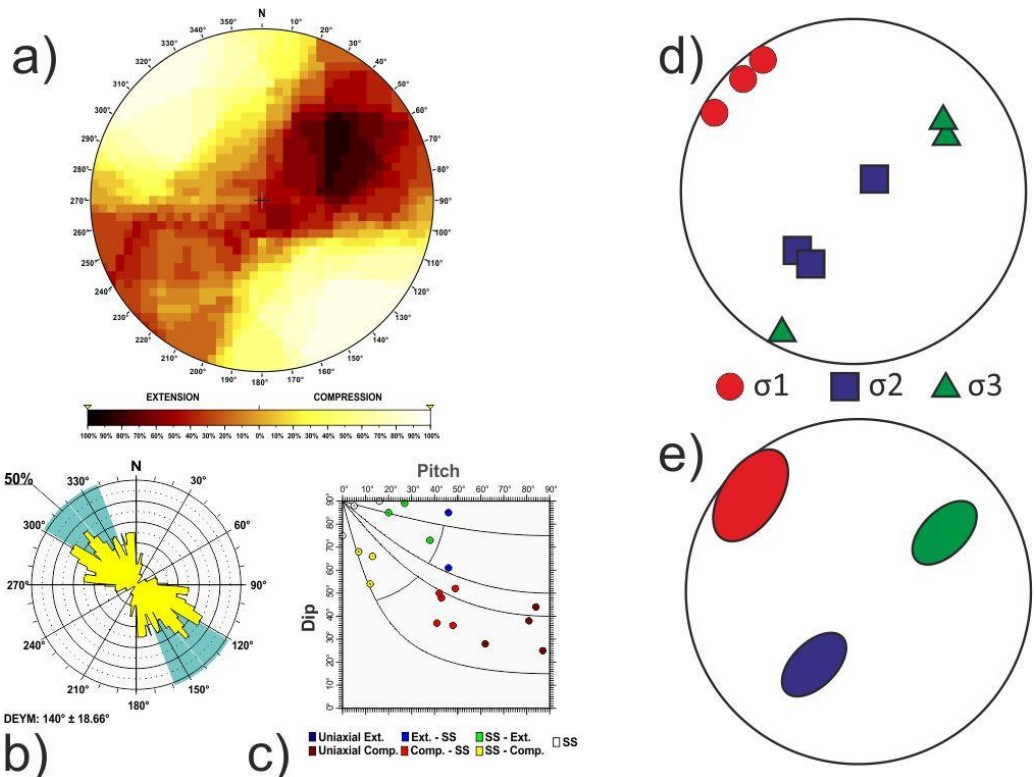


**Fig. B20 Western Spanish Portuguese Central System tectonic zone. Results of the stress and strain analyses for different zones: a) Right Dihedra solution. b) Rose diagram of the Dey (horizontal shortening direction) obtained from the Slip model. C) Pitch/Dip plot for the neo-formed nodal planes obtained from the Slip Model. d) Stress Inversion Results. e) Variability of the three principal stress axes of the stress inversion.**






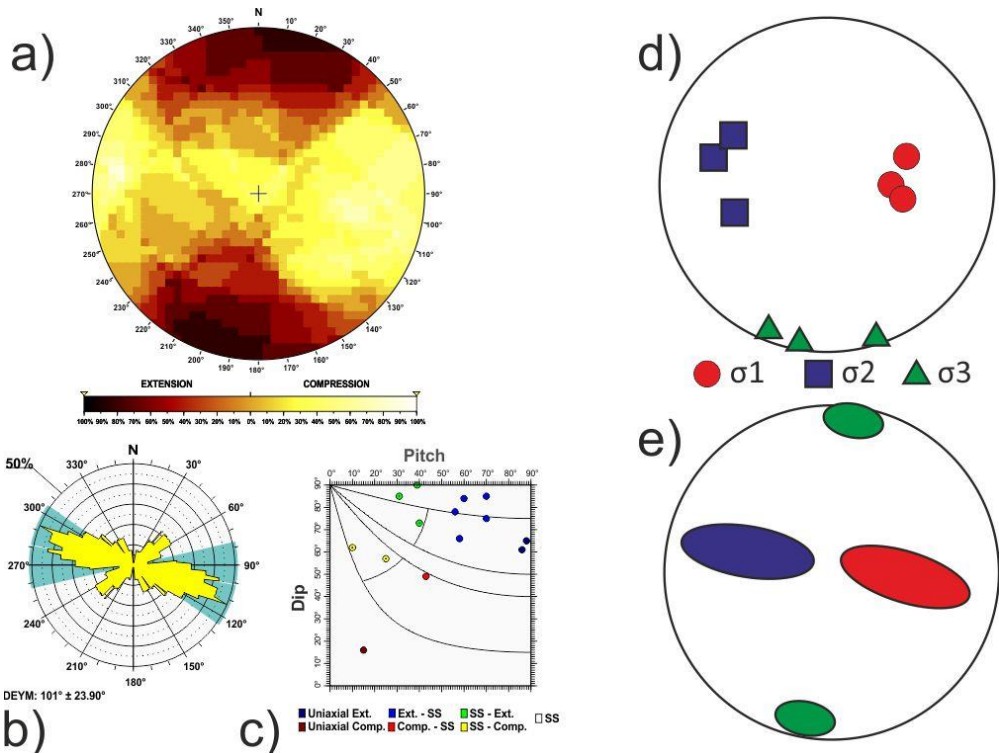


**Fig. B21 Western Pyrenees tectonic zone. Results of the stress and strain analyses for different zones: a) Right Dihedra solution. b) Rose diagram of the Dey (horizontal shortening direction) obtained from the Slip model. C) Pitch/Dip plot for the neo-formed nodal planes obtained from the Slip Model. d) Stress Inversion Results. e) Variability of the three principal stress axes of the stress inversion.**





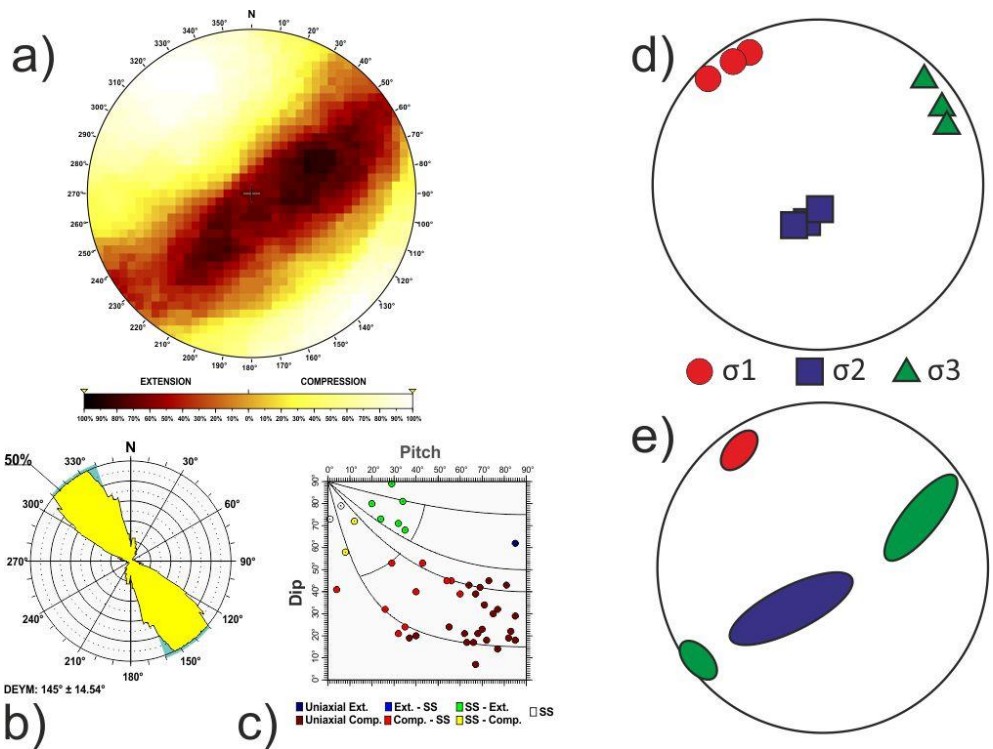

**Fig. B22 West Tell Atlas tectonic zone. Results of the stress and strain analyses for different zones: a) Right Dihedra solution. b) Rose diagram of the Dey (horizontal shortening direction) obtained from the Slip model. C) Pitch/Dip plot for the neo-formed nodal planes obtained from the Slip Model. d) Stress Inversion Results. e) Variability of the three principal stress axes of the stress inversion.**



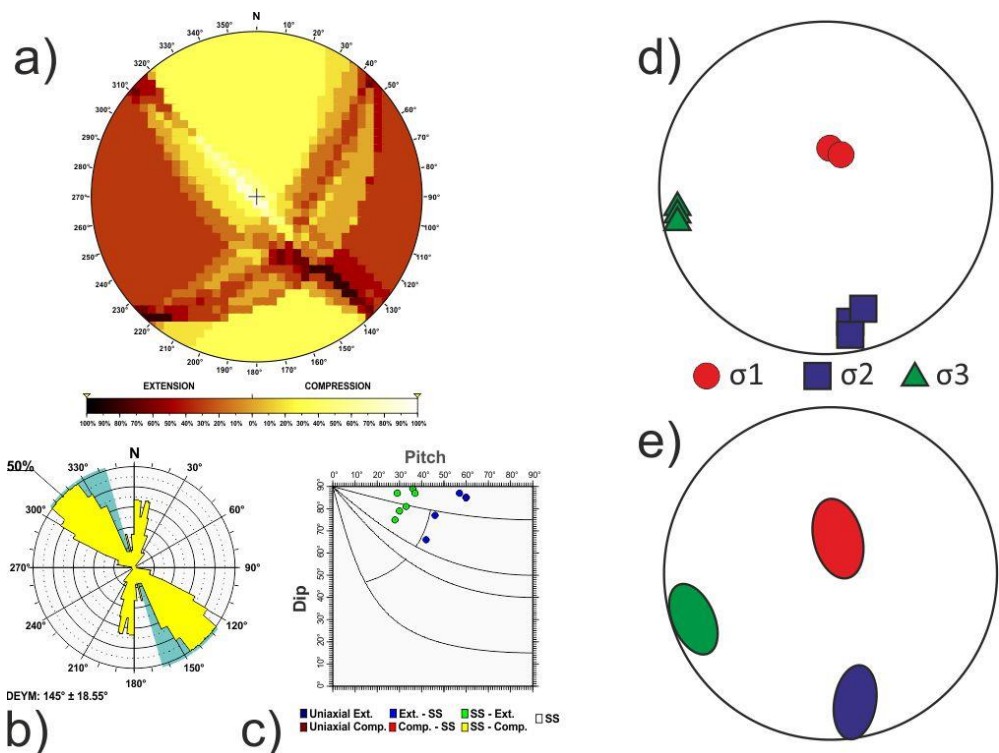

**Fig. 1B23 Western Valencia tectonic zone. Results of the stress and strain analyses for different zones: a) Right Dihedra solution.**
**b) Rose diagram of the Dey (horizontal shortening direction) obtained from the Slip model. C) Pitch/Dip plot for the neo-formed nodal planes obtained from the Slip Model. d) Stress Inversion Results. e) Variability of the three principal stress axes of the stress inversion.**





**Data availability**

Focal mechanisms used in this study are compiled from: Braunmiller et al., 2002; Carreño et al., 2008; Cesca et al., 2021; Chevrot et al., 2011; Custodio et al., 2016; Del Pie Perales, 2016; Dziewonski et al., 1981; Ekström et al., 2012; ETH-Swiss Seismological Service; GFZ-Postdam; Global Centroid Moment Tensor; IAG Instituto Andaluz de Geofísica; IGN Instituto Geográfico Nacional de España; IPMA Portuguese Institute for Sea and Atmosphere; INGV Istituto Nazionale di Geofisica e Vulcanologia; Martín et al., 2015; Matos et al., 2018; Olaiz et al.2024; Pondrelli et al., 2002, 2004; Rueda and Mezcua, 2005; Scognamiglio et al., 2006; Stich et al., 2003, 2005, 2006, 2010, 2020 and Villaseñor et al., 2020.

Supplementary material includes the focal mechanism compiled and the calculated for this study. The database, encompassing both the results of this study and prior data, is standardized in accordance with World Stress Map guidelines and is accessible.

The dataset from Olaiz et al., 2024 is available at Zenodo repository https://doi.org/10.5281/zenodo.14326528.

**Author contribution**

AO, JAAG, GDV, AMM conceived the idea. AO, JAAG, GDV, AMM, JVC, SC, DV and OH performed the formal analysis, investigation and methodology. JVC and DN calculated the new focal mechanisms. AO, JAAG, GDV, AMM, SC and OH completed the supervision and visualization of the manuscript. AO and OH worked on the data curation. AO, JAAG, GDV, AMM, OH prepared the original manuscript; all the authors contributed to the review and editing. JAAG and AMM worked on funding acquisition

**Competing interest**

The contact author has declared that none of the authors has any competing interests

**Acknowledgements**

The Generic Mapping Tools (GMT) were used for figure plotting (Wessel and Smith, 1995). The authors thank XXX anonymous reviewers for their comments and suggestions that have improved the previous version of this manuscript.



**Financial support**

The authors acknowledge support from the projects GEOMARHIS (PID2022-138360NB-I00) and model_SHaKER (PID2021-124155NB-C31) funded by the Ministry of Science, Innovation and Universities of Spain.

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
