# Peer review of "Onshore and offshore seismotectonics of Iberia: An updated review"

_EGUsphere, 2024_

## Referee Comment (RC2)

*[handwritten annotation: ? this is the name of the peninsula!]*

[referee-annotated manuscript omitted]

---

## Author Comment (AC2)

The manuscript "Seismo-tectonics of Greater Iberia: An updated review" provides a comprehensive analysis of the seismo-tectonic characteristics and stress regimes in the Iberian Peninsula, based on an extensive compilation of 542 moment tensor focal mechanisms. The study employs various methodologies, including focal mechanism classification, stress inversion techniques, and Slip Model analyses to assess contemporary tectonic deformations and stress distributions. In conclusion, this updated review enhances knowledge about seismo-tectonics in Iberia by providing detailed insights into active stress regimes and their implications for seismic risk assessment in this geologically complex region.

This is an up to date seismic contribution to stress distribution in the Iberian peninsula and worth to be published.

I have annotated the manuscript by pen, hopefully the author can decipher my hand writing. Of course, there are some flaws, typos and use of "poor" english. Interestingly the quality varies during the manuscript, seems like different chapter have been writing by individual authors, and nor "stream-lining" has been carried out. This is a pity! As the quality of data and illustrations are excellent and clear.

Dear Dr. Reicherter,

We greatly appreciate your valuable review and your kind words. Your suggestions have improved significantly the quality of the original manuscript. We have implemented your comments in the original document.

Please, find below our responses in red

Best regards,

Antonio Olaiz

Some more general comments:

"Greater Iberia" - is not existing. It´s short Iberia or Iberian Peninsula. Avoid that term. Iberia has 4 nations since more than 300 years, and nobody refers to Spain as "Smaller Iberia".

Rock units are upper and lower, time (ages) is Early and Late. Correct this throughout the ms.

Sorry, but according to the international chronostratigraphic chart, the correct terms for age are lower, middle, and upper. (www.stratigraphy.org)

The introduction and objective chapter (1) needs to be re-written in terms of plate tectonics the review and status are sometimes not correct. Partly Iberia is considered as an individual plate (line 38), sometimes African Plate is Nubian Plate and v.v.  It should be consistent. Done (yellow labelled). If I wrote "ref" a reference is missing. The introduction needs a clear separation of plate tectonics and stresses induced by different sources.

Done. The new Introduction separates Iberian Peninsula from the Iberian microplate and plate-related stresses from those with a more local origin:

1 Introduction and objectives

[revised manuscript text omitted]

Line 61: seismic institutions? Better Seismic Obsersatories or Geophysical Institutes…

Done, we changed to Geophysical Institutes

Line 69: SHmax should be written consistently in the ms.

Done. We changed to $S_{hmax}$

Fig.1. Some structures are missing, Gafarillos Fault? Palomares Fault is cutting the Carboneras (better: the Carboneras Fault is ending at the Palomares). Why difference between Post-Orogenic and Late Miocene extension? It is basically in the Betics identical?

We have added the names of the cited faults in the manuscript according to Reviewer 1

Post-orogenic extension is related to the Pyrenees, whereas Late Miocene extension is due to different causes, discussed in the text

Line 99: 30 km focal depth means in the oceanic realm  --> it is in the mantle lithosphere?

Yes. It is written: The events are shallower than the Moho proposed by Diaz et al. (2016), except for some events located in oceanic crust (depth < 30 km), where the rheology of the upper mantle may be assumed to be similar to that of the crust.

Line 116: please sort like the description before…

Done. We have changed to: Conversely, when we deal with stresses, we use thrusting stress regime, strike-slip stress regime, and normal faulting stress regime.

Figure 3: what about the gaps, in Mallorca or central Spain.

We have implemented a search radius of 150 km, which may determine lacks in the final interpolation.

We have included "Search radius 150 km" in the figure caption.

4 Tectonic zonation chapter: this can be organized better, some descriptions are missing

We have reviewed the text and including Granada Basin that was missing.

Table 4 and 5: descriptions are varying? Why?

Tables 4 and 5 summarize the results of different approaches. We have homogenized including Betics > 20 km

Popultaion Betics >20 km was missing in the Supplementary material. Now is included.

Line 394: In 5.5 El Camp Fault was already mentioned in 5.3? Reduce redudancy.

The Camp fault can be considered as belonging to both the western Valencia trough and the northern Catalan coastal range. Moreover, it has the importance of having been one of the first faults studied with palaeoseismological methods in Iberia. We believe that it is not redundant to mention it in both cases.

Line 420: sentence incomplete

Completed

Line 484 and others: N070°E-N090°E is rather bulky, why not N070-090°E much simpler and easy, and please consistent throughout the manuscript.

Done

Line 499: I find the SVT here a bit displaced in your listing? It does not fit here.

We have listed the areas roughly from N to S, so SVT is explained here.

Line 589: confusion 5.14 was already called WAA? Should be Granada Basin?

It is true. Corrected

Chapter 5.15: I was wondering if the authors ignore geological work done? My own (sorry again, but this was the reason I reviewed the ms) work from 2005 (Reicherter and Peters Tectonophysics) already describes radial extension and a recent stress field including active faults. What about the Arenas de Rey earthquake 1884? This paragraph can be improved significantly. The intrabasinal deformation is compared with the margins of the GB different. Also the Jabaloy et al paper has not been considered. I know the paper are "old" but according to your new data, they already mentioned several facts.

We have added: extensional basin within the orogen, which is dominated by the presence of NW-SE normal faults related to radial extension (Reicherter and Peters, 2005).

Reicherter, K.R. and Peters, G. Neotectonic evolution of the Central Betic Cordilleras (Southern Spain). Tectonophysics 405,191-212. https://doi.org/10.1016/j.tecto.2005.05.022, 2005

Line 739: remnant effect of the slab? This should be expleaind better and reference is missing? Is this really mechanically possible? Why 20 km depth? The earthquakes there (Malaga region) are usually much deeper? Is there mid-crustal detachment?

It is written: likely influenced by the remnant effect of the slab (Gea et al., 2023). In this paper, the remnant effect of the slab is explained.

the 20 km division is for comparison with the results of Ruiz-Constán et al. (2012)

Line 832: "As it can been seen in Fig....." this is really poor English, and degrades the quality of the manuscript. This refers to the entire chapter 7 Discussion, please consider a re-writing, as the quality does not meet international standards.

Rewritten: As shown in Fig. Discussion. All the text has been reviewed.

Line 966 It should be chapter 8 Conclusions, not 7...

Sorry, yes.

Fig. 15 - this is not an Alpine tectonic map of Iberia.... It is a map of recent stress in Iberia, please change text (Line 945) accordingly.

Done

I didn´t check the references for completeness, this is editors work. Supplementary maps are very nice, but directly outline the problems: where there is no earthquake .... and especially for the GB I have major doubts, as marginal faults do not appear as seismically active.

I hope this review helps improving the manuscript, if you cannot decipher my hand-writing in the ms, let me know. Good luck.

Aachen, 25/3/2025 Prof. Klaus Reicherter

---

## Author Comment (AC3)

**Reviewer 3**

The manuscript by Olaiz et al. offers valuable insights into the seismotectonics of greater Iberia. The authors have compiled and analysed over 500 focal mechanism solutions to enhance the understanding of the region's active tectonics. While there are some typos and inconsistencies in acronym usage (e.g., SHmax) and figure references (e.g., "Fig." vs. "Figure"), these are minor issues likely to be addressed during the production stage. Overall, the manuscript is well-organized but would benefit from an additional round of English editing. Therefore, I recommend publication pending minor revision.

Dear Reviewer,

We sincerely value your feedback. Your suggestions will greatly enhance the manuscript.

Please, find below our comments written in red.

Best regards,

Antonio Olaiz

Some general comments

A clearer explanation of the database would be helpful, particularly regarding how the authors identified and handled duplicate entries. While the highest %DC is mentioned, additional detail on the process would improve clarity.

During the compilation and merge of different databases, the date and origin time format was standardized. When two or more events share the same date, the time is reviewed. If the time also matches, the coordinates are analysed to confirm that it is the same event. Finally, the %DC is compared, and only the higher value is retained.

I feel the depth parameter was not thoroughly discussed across the different sections. For example, were there any observed changes in stress regime with depth? Also, I wasn't sure how depth was accounted for in Figure 3; was the map created using focal mechanism data? If so, how did you generate a map considering that focal mechanisms come from varying depths?

In this approach, we assume that the type of stress is uniform throughout the entire seismogenic crust. With two exceptions: The focal mechanisms in the Atlantic offshore, which, being only five in number, we have included mechanisms at mantle depths. The solution is very congruent. In the Western Betics population, where evidence of a vertical slab is present, we have considered two subpopulations: those above and below 20 km.

A clearer explanation on your 're-evaluation' for other stress data (inferred from WSM) would be great. Also, I think WSM has lots of stress inferred from focal mechanics solutions for this region. So, how did you deal with it as you also have a new comprehensive database of FMS.

We will update and expand the text accordingly including a minor update of the quality assignment of the Iberia data set which does not affect any of the results. The latter is only a technical issue to be consistent with the new release of the World Stress Map (WSM) database 2025 (Heidbach et al., 2025) and the WSM technical report TR 25-01 where the latest update of the WSM quality ranking has been published very recently (Rajabi et al., 2025). The Iberia dataset has been integrated into the WSM database release 2025 with these slight changes that we will also adopt in the manuscript.

Technically we started with the compilation of stress data records from the WSM database release 2016 in the area between 15°W – 5°E and 34°N – 45°N and re-evaluated each data record. For the sub-dataset of single focal mechanisms (FMS data records), we compiled a completely new dataset (see chapter 2 of the manuscript). This was necessary as the WSM cannot look into regional details. This is an agreement with the WSM policy encouraging regional studies (special study areas) that are more precise in the data assessment. If such a special study area is reported the dataset is replaced in the global WSM compilation. This has been done e.g. for Iceland (Ziegler et al., 2016) and more recently for Taiwan (Heidbach et al., 2022). These special study areas are also explained in the WSM TR 25-01 (Rajabi et al., 2025) and our study is one of these.

The completely new compilation resulted in 542 data records with robust focal mechanisms. These were used in two ways: First, determined from the nodal plane of each focal mechanism the P-, T-, and B-axes and applied the WSM guidelines to derive from these the $S_{Hmax}$ orientation and the stress regime and assigned the data quality following the WSM quality ranking scheme (see WSM TR 25-01 of Rajabi et al., 2025). Secondly, we use these focal mechanisms for a formal stress inversion (FMF) that resulted in 24 FMF data records (see Tab. 5 of the manuscript).

For all other stress indicator types in the WSM from borehole data (BO, DIF, HF), overcoring measurements (OC) and geological fault slip analysis (GFI), we checked for each data record if the information needed to assign a data quality is provided and correctly taken from the original literature. We then re-assigned the data quality according to the latest WSM quality ranking scheme 2025 (now published in the aforementioned WSM TR 25-01). We also checked the literature for new data records in the regional of interest and added these to the compilation.

This new compilation of FMS data records from earthquake focal mechanism (n=542), new FMF data records (n=24, this study) and the new assessment of all old data records according to the up-to-date quality assignment resulted in average in a decrease of data records with higher quality, but we now have a consistent and robust dataset. This decrease is a typical result of other special study areas since lots of data records haven't been touched partly for 30 years when the first major WSM database was released in 1992, but progress in knowledge how to interpret data more robust results typically in a downgrading the quality following the up-to-date WSM quality ranking scheme.

**References**

Heidbach, O., Liang, W.-T., Morawietz, S., von Specht, S., and Ma, K.-F.: Stress Map of Taiwan 2022, GFZ German Research Centre for Geosciences, Potsdam, 10.5880/wsm.Taiwan2022, 2022.

Heidbach, O., Rajabi, M., Di Giacomo, D., Harris, J., Lammers, S., Morawietz, S., Pierdominici, S., Reiter, K., von Specht, S., Storchak, D., and Ziegler, M. O.: World Stress Map Database Release 2025, GFZ Data Services [dataset], 10.5880/WSM.2025.001, 2025.

Rajabi, M., Lammers, S., and Heidbach, O.: WSM database description and guidelines for analysis of horizontal stress orientation from borehole logging, GFZ Helmholtz Centre for Geosciences, Potsdam, WSM TR 25-01, 118, 10.48440/WSM.2025.001, 2025.

Ziegler, M. O., Rajabi, M., Heidbach, O., Hersir, G. P., Ágústsson, K., Árnadóttir, S., and Zang, A.: The stress pattern of Iceland, Tectonophys., 674, 101-113, 10.1016/j.tecto.2016.02.008, 2016.

A supplement (or appendix) consisting of the details of 542 focal mechanics solutions would be great.

Following journal guidelines, a Zenodo repository has been created, including the complete and referenced database at https://doi.org/10.5281/zenodo.14326528.

The text in the data availability chapter has been modified to be more comprehensive.

Supplementary material includes the focal mechanism compiled and the calculated for this study is available at a Zenodo repository ( https://doi.org/10.5281/zenodo.14326528). A database encompassing both the results of this study and vintage data from World Stress Map, is standardized in accordance with World Stress Map guidelines and accessible at https://doi.org/10.5281/zenodo.14326528.

Detailed comments

Line 90: use Geofon (GFZ-Potsdam) instead of GFZ-Potsdam.
Done
Line 99: Maybe show the Moho depth of the study area as a map?
It is drawn in the cited reference: Diaz et al., 2016.
Line 113: It needs a sentence or two to explain what the Reches (1992) methos is known for and why did you prefer this method?
Done
This approach enables iterative testing of various friction coefficients, validated by angular criteria established by SLIP and PAM, as detailed in the subsequent section. The methodology has been recently revised and implemented in MATLAB (Busetti et al., 2014; Wetzler et al., 2021).

Line 115: thrust or thrusting?
Thrusting. Done
Line 115-117: What about stress orientation?
We have added "and stress-strain orientations"
Why both 3.1 (line 118) and 3.2 (185) have the same title (i.e., kinematic analysis)?
Sorry. We changed to 3.1 Kinematic analysis. Composite focal mechanism
And 3.2 Kinematic analysis. Slip model
Line 185 to 190 needs at least reference as you are providing some info from the literature.
The provided references (Reches, 1983; de Vicente et al., 1988) give this information.
Figure 3: What particular depth this map has been prepared for?
We answered that question in the second comment.
Line 763: change 'inver-sions' with 'inversions'.
Done

Figure 9: I see lots of SHmax orientations inferred from FMSs on this map. It would be great to clarify if there are new FMSs (based on your database) or if they were in the WSM database?

The map includes both orientations, the new obtained in this study and the previous included in the WSM database. It is hard to represent both using different symbols. However, the database is available at https://doi.org/10.5281/zenodo.14326528.

Figure 15: it would be great to add a background (e.g., topography) and some names on the map for those who are not familiar with the area.

This is what we initially did, with the map in Fig. 1 blurred. Other reviewers suggested that we better make the background white. We believe that simultaneous viewing of the two figures yields better results.

---

## Author Response (AR2)

Dear Dr von Hagke and Dr. Reicherter,

Thank you for your thorough and detailed review of our manuscript. We greatly appreciate your insightful suggestions, which we have incorporated to enhance the quality of the original submission.

We have prepared two versions of the revised manuscript:

- A tracked-changes version, showing all modifications made in response to your comments, along with in-line annotations for clarity.
- A clean version, with all changes accepted and comments removed, representing the final submission.

Additionally, please find below our point-by-point responses to each of your comments.

Madrid, 29 June 2025

10 Jun 2025

**Associate editor decision: Reconsider after major revisions**

by Christoph von Hagke

**Public justification (visible to the public if the article is accepted and published)**:
Dear Dr. Olaiz and co-authors,

I have now received a re-evaluation of your manuscript. After reading your article as well as the revised version, I am happy to see that it has made a good step forward. However, even though this article may become an important contribution, still significant changes are required before it can be accepted.

- While the text is much better, there are still a lot of sentences that require rewording. Please proof read English language.
- Referencing is OK, but often statements lack a respective reference
- Still issues from review #1 are pending, which must be fixed or rebutted before the article can be accepted. Please include all points provided in the annotated pdf by reviewer Klaus Reicherter.
- The discussion, as also noted by K.R., is brief and requires rewording.

Minor comments:
Lines 42 and 54: delete "significant" done
Line 58: Add also older references - this has been known before 2014 done, Facenna et al., 202
Line 58: "Normal faulting stress regime also occurs in the Pyrenees, as they are subject to post-orogenic collapse (ref)" Post orogenic collapse was seuggested by Asensio et al. (2012) reference already included in the text
Line 59-62: rewording required done
Line 65: move reference to end of sentence done
Line 66-69 - rewording required. ... within projects such as TopoIberia have increadsed our knowledge on...done
Line 74: We also dervie SHmax from ...done

Figure 1: numbers hardly readable; some structures missing, what is the difference between Late Miocene to Quarternary extension and post-orogenic extension?; typos in legend We have implemented the changes. The size of the fonts is difficult to adjust due to the amount of information

Line 111: Specify that these are man-made earthquakes done "most likely associated to induced seismicity "

Line 115: reword: "might assume as similar to" done

Line 125: analyses done

Line 126: combined done

Line 146: Fik ==> subscript done

Line 158: reword "by means of the computation of the..." done

Line 173: delete "also" done

Line 203: capitalize or don't capitalize consistently done

Line 210: specify "mechanical point of view"; add sentence why it is helpful to analyze areas under similar strain conditions done "The Slip Model (Reches, 1983; de Vicente et al., 1988) identifies which of the nodal planes is more prone to slip from a mechanical point of view, as it requires less energy to mobilize (the neoformed plane, not a reactivated one)."

Line 218, 219: ez, ey: subscript done

Line 222: Where ==> where done

Line 228: reword "striation-fault pair orientations done ". Therefore, the selected plane can be effectively utilised in stress inversion methods that rely  on fault planes orientations and their associated slip directions  (Angelier and Mechler, 1977).  "

Line 231: inversion done

Line 265: explain that this search radius may lead to lack in interpolation done

Line 275: reword sentence done

Line 288: African Plate done

Line 335: specify - what do you mean by "mostly"? done

Line 350: glacial isostatic adjustment We would prefer the term isostatic, as it encompasses both glacial and erosional process

Line 420: Valencia Trough done

Line 436: add sentence that Camp Fault can be considered to belong to western Valecnia & N CCR done "The El Camp Fault can be considered as part of the NVT. "

Line 447: 8 ==> eight done

Line 582: reference missing I apologize but I am not able to identify where the reference is missed

Section 5.15: reviewer K.R. has raised important points concerning the GB - please accommodate them We have improved the text adding additional information and key references suggested by Dr. Reicherter

Line 660: please discuss whther the Carboneras Fault is indeed a continuation of the Palomares Fault done

Line 815: inver-sions == inversions done

Line 817: Zo-back ==> Zoback done

Line 883: what distinct characteristics?

Line 935 rela-tionship ==> relationship done

Line 936 bounda-ry done

Figure 15: Add plate boundaries, Valencia Trough, Granada Basin, Iberia Basin. Explain stippled lines in caption done. We reviewed all the populations. However, we do not have

added the labels for clarity.

In your revised version, please provide a point by point reply to all comments made by the reviewers, including the ones you decide not to accommodate. I look forward to receiving an updated verison.

Kind regards,
Christoph von Hagke

**Suggestions for revision or reasons for rejection**

(visible to the public if the article is accepted and published)

The manuscript has improved considerably, and in my opinion, after some minor issues it should be ready for publication. So, good work, congratulations to the authors.

Q: Rock units are upper and lower, time (ages) is Early and Late. Correct this throughout the ms.
A: Sorry, but according to the international chronostratigraphic chart, the correct terms for age are lower, middle, and upper. (www.stratigraphy.org)
R2 answer: again, your answer is not correct, please check the website yourself, and you will see L/E and U/L. Please keep logic rules, like I said above. And by the way, it the same in Spanish language…. Rocks can not be late… time can not be upper…. ; ). These are my last words to it. And sorry for being precise.

Done, we have modified accordingly

Line 58: shortenings and extensions? Extension is singular. done

Line 59: neotectonic complex deformation setting? I recommended complex neotectonic deformational setting, your sequence of words is confusing. done

Figure 1 has not been changed, why? There are typos inside, as remarked earlier. Thin-skinned. Done, we have modified accordingly. We have corrected the typos, changes the proposed labels and cleaned the patterns.

Q: The events are shallower than the Moho proposed by Diaz et al. (2016), with exception of some events located in oceanic crust (depth < 30 km).
R2: As I asked correctly, these events should be BELOW the oceanic crust, in the lithosphere? Your reply is not correct, and has obviously not been understood? To clarify this issue: a) give the thickness of the oceanic crust with reference and all are happy. "assume" is not the right word here, as it is reflexive "to assume something". So: "where the rheology might be assumed to be similar….." Done, we have improved the wording including your suggestion " The events are located at depths shallower  than the Moho proposed by Diaz et al. (2016), except for some events located in oceanic crust, where the proposed crustal thickness is estimated at approximately 18 km and hypocentral depths are less than 30 km, within a domain in which the rheology of the upper mantle might be assumed to be similar to the crust. "

Line 125 still I am not happy with "rupture" as this is a failure or earthquake, and might lead to confusion. Done

Headline of Chapter 6, Caption of Fig. 9: GREATER IBERIA… I thought this battle is fought? Done, corrected

Line 829: why didn´t you add thrusts (there is in Fig.10b clearly a thrust at Goringe Bank)? done

Finally, the English is still not as it could be?
In the present stage, I would suggest proof-reading by a professional or literate co-author.
I will not re-iterate all my comments/corrections/suggestions here, it is in version 1 of my review.
The entire DISCUSSION chapter has to be reformulated and made readable.

We have carried out an exhaustive linguistic review to ensure the highest standard of English throughout the text, especially focused in discussion and conclusions.

We have extended the conclusions chapter.

Regarding fig 11 comments, included in first review comments, we have included this comment: " The line shown in figure 11d is the representation of the diagonal line of the Flinn diagram, which shows the plane-strain tensor, or double couple in seismic tensor terminology. The k value is a measure of the distance to this line, so this value can be considered a measure of the deviation of the seismic strain tensor from the pure double couple (similar to the fclvd value). An improved explanation has been included in the figure caption."

Klaus Reicherter, 06/07/2025

The manuscript "Seismo-tectonics of Greater Iberia: An updated review" provides a comprehensive analysis of the seismo-tectonic characteristics and stress regimes in the Iberian Peninsula, based on an extensive compilation of 542 moment tensor focal mechanisms. The study employs various methodologies, including focal mechanism classification, stress inversion techniques, and Slip Model analyses to assess contemporary tectonic deformations and stress distributions. In conclusion, this updated review enhances knowledge about seismo-tectonics in Iberia by providing detailed insights into active stress regimes and their implications for seismic risk assessment in this geologically complex region.

This is an up to date seismic contribution to stress distribution in the Iberian peninsula and worth to be published.

I have annotated the manuscript by pen, hopefully the author can decipher my hand writing. Of course, there are some flaws, typos and use of "poor" english. Interestingly the quality varies during the manuscript, seems like different chapter have been writing by individual authors, and nor "stream-lining" has been carried out. This is a pity! As the quality of data and illustrations are excellent and clear.

Some more general comments:

"Greater Iberia" - is not existing. It´s short Iberia or Iberian Peninsula. Avoid that term. Iberia has 4 nations since more than 300 years, and nobody refers to Spain as "Smaller Iberia". Done

Rock units are upper and lower, time (ages) is Early and Late. Correct this throughout the ms. Done

The introduction and objective chapter (1) needs to be re-written in terms of plate tectonics the review and status are sometimes not correct. Partly Iberia is considered as an individual plate (line 38), sometimes African Plate is Nubian Plate and v.v.  It should be consistent. Done (yellow labelled). If I wrote "ref" a reference is missing. The introduction needs a clear separation of plate tectonics and stresses induced by different sources.

Done. The new Introduction separates Iberian Peninsula from the Iberian microplate and plate-related stresses from those with a more local origin:

1 Introduction and objectives

[revised manuscript text omitted]

Line 61: seismic institutions? Better Seismic Obsersatories or Geophysical Institutes…

Done, we changed to Geophysical Institutes

Line 69: SHmax should be written consistently in the ms.

Done. We changed to $S_{hmax}$

Fig.1. Some structures are missing, Gafarillos Fault? Palomares Fault is cutting the Carboneras (better: the Carboneras Fault is ending at the Palomares). Why difference between Post-Orogenic and Late Miocene extension? It is basically in the Betics identical?

We have added the names of the cited faults in the manuscript according to Reviewer 1

Post-orogenic extension is related to the Pyrenees, whereas Late Miocene extension is due to different causes, discussed in the text

Line 99: 30 km focal depth means in the oceanic realm --> it is in the mantle lithosphere?

Yes. It is written: The events are shallower than the Moho proposed by Diaz et al. (2016), except for some events located in oceanic crust (depth < 30 km), where the rheology of the upper mantle may be assumed to be similar to that of the crust.

Line 116: please sort like the description before…

Done. We have changed to: Conversely, when we deal with stresses, we use thrusting stress regime, strike-slip stress regime, and normal faulting stress regime.

Figure 3: what about the gaps, in Mallorca or central Spain.

We have implemented a search radius of 150 km, which may determine lacks in the final interpolation.

We have included " Search radius 150 km" in the figure caption.

4 Tectonic zonation chapter: this can be organized better, some descriptions are missing

We have solved some issues. Hopefully now it is more clear.

Table 4 and 5: descriptions are varying? Why?

Tables 4 and 5 summarize the results of different approaches. We have homogenized including Betics > 20 km

Popultaion Betics >20 km was missing in th Supplementary material. Now is included.

Line 394: In 5.5 El Camp Fault was already mentioned in 5.3? Reduce redudancy.

The El Camp Fault can be considered as belonging to both the western Valencia trough and the northern Catalan coastal range. Moreover, it has the importance of having been one of the first faults studied with palaeoseismological methods in Iberia. We believe that it is not redundant to mention it in both cases.

Line 420: sentence incomplete

Completed

Line 484 and others: N070°E-N090°E is rather bulky, why not N070-090°E much simpler and easy, and please consistent throughout the manuscript.

Done

Line 499: I find the SVT here a bit displaced in your listing? It does not fit here.

We have listed the areas roughly from N to S, so SVT is explained here.

Line 589: confusion 5.14 was already called WAA? Should be Granada Basin?

It is true. Corrected

Chapter 5.15: I was wondering if the authors ignore geological work done? My own (sorry again, but this was the reason I reviewed the ms) work from 2005 (Reicherter and Peters Tectonophysics) already describes radial extension and a recent stress field including active faults. What about the Arenas de Rey earthquake 1884? This paragraph can be improved significantly. The intrabasinal deformation is compared with the margins of the GB different. Also the Jabaloy et al paper has not been considered. I know the paper are "old" but according to your new data, they already mentioned several facts.

We have added: extensional basin within the orogen, which is dominated by the presence of NW-SE normal faults related to radial extension (Reicherter and Peters, 2005).

Reicherter, K.R. and Peters, G. Neotectonic evolution of the Central Betic Cordilleras (Southern Spain). Tectonophysics 405,191-212. https://doi.org/10.1016/j.tecto.2005.05.022, 2005

Line 739: remnant effect of the slab? This should be expleaind better and reference is missing? Is this really mechanically possible? Why 20 km depth? The earthquakes there (Malaga region) are usually much deeper? Is there mid-crustal detachment?

It is written: likely influenced by the remnant effect of the slab (Gea et al., 2023). In this paper, the remnant effect of the slab is explained.

the 20 km division is for comparison with the results of Ruiz-Constán et al. (2012)

Line 832: "As it can been seen in Fig....." this is really poor English, and degrades the quality of the manuscript. This refers to the entire chapter 7 Discussion, please consider a re-writing, as the quality does not meet international standards.

Rewritten: As shown in Fig. Discussion. All the text has been reviewed.

Line 966 It should be chapter 8 Conclusions, not 7...

Sorry, yes.

Fig. 15 - this is not an Alpine tectonic map of Iberia.... It is a map of recent stress in Iberia, please change text (Line 945) accordingly.

Done

I didn´t check the references for completeness, this is editors work. Supplementary maps are very nice, but directly outline the problems: where there is no earthquake .... and especially for the GB I have major doubts, as marginal faults do not appear as seismically active.

I hope this review helps improving the manuscript, if you cannot decipher my hand-writing in the ms, let me know. Good luck.

Aachen, 25/3/2025 Prof. Klaus Reicherter

**Reviewer 3**

The manuscript by Olaiz et al. offers valuable insights into the seismotectonics of greater Iberia. The authors have compiled and analysed over 500 focal mechanism solutions to enhance the understanding of the region's active tectonics. While there are some typos and inconsistencies in acronym usage (e.g., SHmax) and figure references (e.g., "Fig." vs. "Figure"), these are minor issues likely to be addressed during the production stage. Overall, the manuscript is well-organized but would benefit from an additional round of English editing. Therefore, I recommend publication pending minor revision.

Some general comments

A clearer explanation of the database would be helpful, particularly regarding how the authors identified and handled duplicate entries. While the highest %DC is mentioned, additional detail on the process would improve clarity.

During the compilation and merge of different databases, the date and origin time format was standardized. When two or more events share the same date, the time is reviewed. If the time also matches, the coordinates are analysed to confirm that it is the same event. Finally, the %DC is compared, and only the higher value is retained.

I feel the depth parameter was not thoroughly discussed across the different sections. For example, were there any observed changes in stress regime with depth? Also, I wasn't sure how depth was accounted for in Figure 3; was the map created using focal mechanism data? If so, how did you generate a map considering that focal mechanisms come from varying depths?

In this approach, we assume that the type of stress is uniform throughout the entire seismogenic crust. With two exceptions: The focal mechanisms in the Atlantic offshore, which, being only five in number, we have included mechanisms at mantle depths. The solution is very congruent. In the Western Betics population, where evidence of a vertical slab is present, we have considered two subpopulations: those above and below 20 km.

A clearer explanation on your 're-evaluation' for other stress data (inferred from WSM) would be great. Also, I think WSM has lots of stress inferred from focal mechanics solutions for this region. So, how did you deal with it as you also have a new comprehensive database of FMS.

We have updated the text accordingly including a minor update of the quality assignment of the Iberia data set which does not affect any of the results. The latter is only a technical issue to be consistent with the new release of the World Stress Map (WSM) database 2025 (Heidbach et al., 2025) and the WSM technical report TR 25-01 where the latest update of the WSM quality ranking has been published very recently (Rajabi et al., 2025). The Iberia dataset has been integrated into the WSM database release 2025 with these slight changes that we will also adopt in the manuscript.

Technically we started with the compilation of stress data records from the WSM database release 2016 in the area between 15°W – 5°E and 34°N – 45°N and re-evaluated each data record. For the sub-dataset of single focal mechanisms (FMS data records), we compiled a completely new dataset (see chapter 2 of the manuscript). This was necessary as the WSM cannot look into regional details. This is an agreement with the WSM policy

encouraging regional studies (special study areas) that are more precise in the data assessment. If such a special study area is reported the dataset is replaced in the global WSM compilation. This has been done e.g. for Iceland (Ziegler et al., 2016) and more recently for Taiwan (Heidbach et al., 2022). These special study areas are also explained in the WSM TR 25-01 (Rajabi et al., 2025) and our study is one of these.

The completely new compilation resulted in 542 data records with robust focal mechanisms. These were used in two ways: First, determined from the nodal plane of each focal mechanism the P-, T-, and B-axes and applied the WSM guidelines to derive from these the $S_{Hmax}$ orientation and the stress regime and assigned the data quality following the WSM quality ranking scheme (see WSM TR 25-01 of Rajabi et al., 2025). Secondly, we use these focal mechanisms for a formal stress inversion (FMF) that resulted in 24 FMF data records (see Tab. 5 of the manuscript).

For all other stress indicator types in the WSM from borehole data (BO, DIF, HF), overcoring measurements (OC) and geological fault slip analysis (GFI), we checked for each data record if the information needed to assign a data quality is provided and correctly taken from the original literature. We then re-assigned the data quality according to the latest WSM quality ranking scheme 2025 (now published in the aforementioned WSM TR 25-01). We also checked the literature for new data records in the regional of interest and added these to the compilation.

This new compilation of FMS data records from earthquake focal mechanism (n=542), new FMF data records (n=24, this study) and the new assessment of all old data records according to the up-to-date quality assignment resulted in average in a decrease of data records with higher quality, but we now have a consistent and robust dataset. This decrease is a typical result of other special study areas since lots of data records haven't been touched partly for 30 years when the first major WSM database was released in 1992, but progress in knowledge how to interpret data more robust results typically in a downgrading the quality following the up-to-date WSM quality ranking scheme.

A supplement (or appendix) consisting of the details of 542 focal mechanics solutions would be great.

Following journal guidelines, a Zenodo repository has been created, including the complete and referenced database at https://doi.org/10.5281/zenodo.14326528.

The text in the data availability chapter has been modified to be more comprehensive.

 Supplementary material includes the focal mechanism compiled and the calculated for this study is available at a Zenodo repository ( https://doi.org/10.5281/zenodo.14326528). A database encompassing both the results of this study and vintage data from World Stress Map, is standardized in accordance with World Stress Map guidelines and accessible at https://doi.org/10.5281/zenodo.14326528.

Detailed comments

Line 90: use Geofon (GFZ-Potsdam) instead of GFZ-Potsdam.

Done

Line 99: Maybe show the Moho depth of the study area as a map?

It is drawn in the cited reference: Diaz et al., 2016.

Line 113: It needs a sentence or two to explain what the Reches (1992) methos is known for and why did you prefer this method?

Done

This approach enables iterative testing of various friction coefficients, validated by angular criteria established by SLIP and PAM, as detailed in the subsequent section. The methodology has been recently revised and implemented in MATLAB (Busetti et al., 2014; Wetzler et al., 2021).

Line 115: thrust or thrusting?

Thrusting. Done

Line 115-117: What about stress orientation?

We have added "and stress-strain orientations"

Why both 3.1 (line 118) and 3.2 (185) have the same title (i.e., kinematic analysis)?

Sorry. We changed to 3.1 Kinematic analysis. Composite focal mechanism

And 3.2 Kinematic analysis. Slip model

Line 185 to 190 needs at least reference as you are providing some info from the literature.

The provided references (Reches, 1983; de Vicente et al., 1988) give this information.

Figure 3: What particular depth this map has been prepared for?

We answered that question in the second comment.

Line 763: change 'inver-sions' with 'inversions'.

Done

Figure 9: I see lots of SHmax orientations inferred from FMSs on this map. It would be great to clarify if there are new FMSs (based on your database) or if they were in the WSM database?

The map includes both orientations, the new obtained in this study and the previous included in the WSM database. It is hard to represent both using different symbols. However, the database is available at https://doi.org/10.5281/zenodo.14326528.

Figure 15: it would be great to add a background (e.g., topography) and some names on the map for those who are not familiar with the area.

This is what we initially did, with the map in Fig. 1 blurred. Other reviewers suggested that we better make the background white. We believe that simultaneous viewing of the two figures yields better results.

---

## Author Response (AR3)

Dear Dr. Fusseis and Dr von Hagke

Thank you for accepting our manuscript. We believe it makes a valuable contribution to the knowledge of the seismotectonics in the area, and we are honoured to have our work published in Solid Earth.

We greatly appreciate your thorough review and invaluable support throughout the submission process. Your insightful suggestions have been incorporated to enhance the quality of our manuscript.

All minor corrections suggested on July 8th have been addressed in the final version. Please find the details below.

Sincerely,

Antonio Olaiz

Madrid, 15 July 2025

**13 Jul 2025**

**Executive editor decision: Publish subject to technical corrections**

**by Florian Fusseis**

**Comments to the author:**
Dear Authors,

Following the recommendation and assessment of our Associate Editor, I am pleased to accept your paper for publication, subject to the minor technical corrections outlined by Prof von Hagke.

Thanks for submitting your work to SE!

Witth best regards,
Florian Fusseis

**08 Jul 2025**

**Associate editor decision: Publish subject to technical corrections**

**by Christoph von Hagke**

**Public justification (visible to the public if the article is accepted and published):**
Dear authors,

Thank you for the revised version of the manuscript. It is now almost ready for publication, after some minor corrections.

- in all table, "." should be used instead of "," done
Line 809: delete "very" done

Line 873: strikking ==> striking done
Line 876: delete "." done
Line 889: oare ==> are done
Line 897: delete " done
Line 902: wherefocal ==> where focal done
Line 902: have determined ==> have been determined done
Line 948: analyzed ==> analysed done
Line 961: Alboran Slab done
Line 966: delete "." done
Line 968: delete " , " done
Line 980: trasntensional ==> transtensional done
Line 982: Iberia.. ==> Iberia. done

Congratulations on this nice article.
Christoph von Hagke